# Robust large-margin learning in hyperbolic space

**Melanie Weber**[*]
Princeton University
mw25@math.princeton.edu

**Manzil Zaheer**
Google Research
manzilzaheer@google.com

**Ankit Singh Rawat**
Google Research
ankitsrawat@google.com

**Aditya Menon**
Google Research
adityakmenon@google.com

**Sanjiv Kumar**
Google Research
sanjivk@google.com

## Abstract

Recently, there has been a surge of interest in representation learning in hyperbolic spaces, driven by their ability to represent hierarchical data with significantly fewer dimensions than standard Euclidean spaces. However, the viability and benefits of hyperbolic spaces for downstream machine learning tasks have received less attention. In this paper, we present, to our knowledge, the first theoretical guarantees for learning a classifier in hyperbolic rather than Euclidean space. Specifically, we consider the problem of learning a *large-margin* classifier for data possessing a hierarchical structure. Our first contribution is a *hyperbolic perceptron* algorithm, which provably converges to a separating hyperplane. We then provide an algorithm to efficiently learn a large-margin hyperplane, relying on the careful injection of *adversarial examples*. Finally, we prove that for hierarchical data that embeds well into hyperbolic space, the low embedding dimension ensures superior guarantees when learning the classifier directly in hyperbolic space.

## 1 Introduction

Hyperbolic spaces have received sustained interest in recent years, owing to their ability to compactly represent data possessing hierarchical structure (e.g., trees and graphs). In terms of *representation learning*, hyperbolic spaces offer a provable advantage over Euclidean spaces for such data: objects requiring an *exponential* number of dimensions in Euclidean space can be represented in a *polynomial* number of dimensions in hyperbolic space [28]. This has motivated research into efficiently learning a suitable hyperbolic embedding for large-scale datasets [22, 4, 31].

Despite this impressive representation power, little is known about the benefits of hyperbolic spaces for downstream tasks. For example, suppose we wish to perform classification on data that is intrinsically hierarchical. One may naïvely ignore this structure, and use a standard Euclidean embedding and corresponding classifier (e.g., SVM). However, can we design classification algorithms that exploit the structure of hyperbolic space, and offer *provable* benefits in terms of *performance*? This fundamental question has received surprisingly limited attention. While some prior work has proposed specific algorithms for learning classifiers in hyperbolic space [6, 21], these have been primarily empirical in nature, and do not come equipped with theoretical guarantees on convergence and generalization.

In this paper, we take a first step towards addressing this question for the problem of learning a *large-margin* classifier. We provide a series of algorithms to *provably* learn such classifiers in hyperbolic space, and establish their superiority over the classifiers naïvely learned in Euclidean space. This shows that by using a hyperbolic space that better reflects the intrinsic geometry of the data, one can see gains in both representation size and performance. Specifically, our contributions are:

---

[*]Work performed while intern at Google Research, New York.

(i) we provide a hyperbolic version of the classic perceptron algorithm and establish its convergence for data that is separable with a margin (Theorem 3.1).

(ii) we establish how suitable injection of *adversarial examples* to *gradient-based* loss minimization yields an algorithm which can *efficiently* learn a *large-margin* classifier (Theorem 4.3). We further establish that simply performing gradient descent or using adversarial examples alone does not suffice to efficiently yield such a classifier.

(iii) we compare the Euclidean and hyperbolic approaches for hierarchical data and analyze the trade-off between low embedding dimensions and low distortion (*dimension-distortion trade-off*) when learning robust classifiers on embedded data. For hierarchical data that embeds well into hyperbolic space, it suffices to use smaller embedding dimension while ensuring superior guarantees when we learn a classifier in hyperbolic space.

Contribution (i) establishes that it is possible to design classification algorithms that exploit the structure of hyperbolic space, while provably converging to *some* admissible (not necessarily large-margin) separator. Contribution (ii) establishes that it is further possible to design classification algorithms that provably converge to a *large-margin* separator, by suitably injecting adversarial examples. Contribution (iii) shows that the adaptation of algorithms to the intrinsic geometry of the data can enable efficient utilization of the embedding space without affecting the performance.

**Related work**. Our results can be seen as hyperbolic analogue of classic results for Euclidean spaces. The large-margin learning problem is well studied in the Euclidean setting. Classic algorithms for learning classifiers include the perceptron [25, 24, 12] and support vector machines [8]. Robust margin-learning has been widely studied; notably by Lanckriet et al. [16], El Ghaoui et al. [10], Kim et al. [15] and, recently, via adversarial approaches by Charles et al. [5], Ji and Telgarsky [14], Li et al. [18] and Soudry et al. [30]. Adversarial learning has recently gained interest through efforts to train more robust deep learning systems (see, e.g., [20, 11]).

Recently, the representation of (hierarchical) data in hyperbolic space has gained a surge of interest. The literature focuses mostly on learning representations in the Poincare [22, 4, 31] and Lorentz [23] models of hyperbolic space, as well as on analyzing representation trade-offs in hyperbolic embeddings [27, 32] . The body of work on performing downstream ML tasks in hyperbolic space is much smaller and mostly without theoretical guarantees. Monath et al. [21] study hierarchical clustering in hyperbolic space. Cho et al. [6] introduce a hyperbolic version of support vector machines for binary classification in hyperbolic space, albeit without theoretical guarantees. Ganea et al. [13] introduce a hyperbolic version of neural networks that shows empirical promise on downstream tasks, but likewise without theoretical guarantees. To the best of our knowledge, neither robust large-margin learning nor adversarial learning in hyperbolic spaces has been studied in the literature, including [6]. Furthermore, we are not aware of any theoretical analysis of dimension-distortion trade-offs in the related literature.

## 2 Background and notation

We begin by reviewing some background material on hyperbolic spaces, as well as embedding into and learning in these spaces.

### 2.1 Hyperbolic space

Hyperbolic spaces are smooth Riemannian manifolds $\mathcal{M} = \mathbb{H}^d$ with constant negative curvature $\kappa$ and are as such locally Euclidean spaces. There are several equivalent models of hyperbolic space, each highlighting a different geometric aspect. In this work, we mostly consider the *Lorentz model* (aka *hyperboloid model*), which we briefly introduce below, with more details provided Appendix A (see [3] for a comprehensive overview).

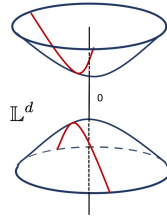

Figure 1: Lorentz model (geodesics in red).

For $\boldsymbol{x}, \boldsymbol{x}' \in \mathbb{R}^{d+1}$, let $\boldsymbol{x} * \boldsymbol{x}' = x_0 x_0' - \sum_{i=1}^{d} x_i x_i'$ denote their *Minkowski product*. The Lorentz model is defined as $\mathbb{L}^d = \{\boldsymbol{x} \in \mathbb{R}^{d+1} : \boldsymbol{x} * \boldsymbol{x} = 1\}$ with distance measure $d_{\mathbb{L}}(\boldsymbol{x}, \boldsymbol{x}') = \mathrm{acosh}(\boldsymbol{x} * \boldsymbol{x}')$. Note that the distance $d_{\mathbb{L}}(\boldsymbol{x}, \boldsymbol{x}')$ corresponds to the length of the shortest line (*geodesic*) along the manifold connecting $\boldsymbol{x}$ and $\boldsymbol{x}'$ (cf. Fig. 1). We also point out that $(\mathbb{L}, d_{\mathbb{L}})$ forms a metric space.

### 2.2 Embeddability of hierarchical data

A map $\phi : X_1 \to X_2$ between metric spaces $(X_1, d_1)$ and $(X_2, d_2)$ is called an *embedding*. The *multiplicative* distortion of $\phi$ is defined to be the smallest constant $c_M \geq 1$ such that, $\forall \boldsymbol{x}, \boldsymbol{x}' \in X_1$,

$d_2\big(\phi(\boldsymbol{x}), \phi(\boldsymbol{x}'))\big) \leq d_1(\boldsymbol{x}, \boldsymbol{x}') \leq c_M \cdot d_2\big(\phi(\boldsymbol{x}), \phi(\boldsymbol{x}')\big)$. When $c_M = 1$, $\phi$ is termed an *isometric embedding*. Since hierarchical data is tree-like, we can use classic embeddability results for trees as a reference point. Bourgain [2], Linial et al. [19] showed that an $N$-point metric $\mathcal{X}$ (i.e., $|\mathcal{X}| = N$) embeds into Euclidean space $\mathbb{R}^{O(\log^2 N)}$ with $c_M = O(\log N)$. This bound is tight for trees in the sense that embedding them in a Euclidean space (of any dimension) must have $c_m = \Omega(\log N)$ [19]. In contrast, Sarkar [28] showed that trees embed quasi-isometrically with $c_M = O(1 + \epsilon)$ into hyperbolic space $\mathbb{H}^d$, even in the low-dimensional regime with the dimension as small as $d = 2$.

## 2.3 Classification in hyperbolic space

We consider classification problems of the following form: $\mathcal{X} \subset \mathbb{L}^d$ denotes the feature space, $\mathcal{Y} = \{\pm 1\}$ the binary label space, and $\mathcal{W} \subset \mathbb{R}^{d+1}$ the model space. In the following, we denote the training set as $\mathcal{S} \subset \mathcal{X} \times \mathcal{Y}$.

We begin by defining *geodesic decision boundaries*. Consider the Lorentz space $\mathbb{L}^d$ with ambient space $\mathbb{R}^{d+1}$. Then every geodesic decision boundary is a hyperplane in $\mathbb{R}^d$ intersecting $\mathbb{L}^d$ and $\mathbb{R}^{d+1}$. Further, consider the set of linear separators or *decision functions* of the form

$$\mathcal{H} = \{h_{\boldsymbol{w}} \colon \boldsymbol{w} \in \mathbb{R}^{d+1}, \boldsymbol{w} * \boldsymbol{w} < 0\}, \quad \text{where} \quad h_{\boldsymbol{w}}(\boldsymbol{x}) = \begin{cases} 1, & \boldsymbol{w} * \boldsymbol{x} > 0 \\ -1, & \text{otherwise.} \end{cases} \quad (2.1)$$

Note that the requirement $\boldsymbol{w} * \boldsymbol{w} < 0$ in (2.1) ensures that the intersection of $\mathbb{L}^d$ and the decision hyperplane $h_{\boldsymbol{w}}$ is not empty. The geodesic decision boundary corresponding to the decision function $h_{\boldsymbol{w}}$ is then given by $\partial \mathcal{H}_{\boldsymbol{w}} = \{\boldsymbol{z} \in \mathbb{L}^d : \boldsymbol{w} * \boldsymbol{z} = 0\}$. The distance of a point $\boldsymbol{x} \in \mathbb{L}^d$ from the decision boundary $\partial \mathcal{H}_{\boldsymbol{w}}$ can be computed as $d\big(\boldsymbol{x}, \partial \mathcal{H}_{\boldsymbol{w}}\big) = \big| \operatorname{asinh} \big(\boldsymbol{w} * \boldsymbol{x}/\sqrt{-\boldsymbol{w} * \boldsymbol{w}}\big)\big|$ [6].

## 2.4 Large-margin classification in hyperbolic space

In this paper, we are interested in learning a *large margin* classifier in a hyperbolic space. Analogous to the Euclidean setting, the natural notion of margin is the minimal distance to the decision boundary over all training samples:

$$\operatorname{margin}_{\mathcal{S}}(w) = \inf_{(\boldsymbol{x}, y) \in \mathcal{S}} y h_{\boldsymbol{w}}(\boldsymbol{x}) \cdot d(\boldsymbol{x}, \partial \mathcal{H}_w) = \inf_{(\boldsymbol{x}, y) \in \mathcal{S}} \operatorname{asinh}\left( y(\boldsymbol{w} * \boldsymbol{x})/\sqrt{-\boldsymbol{w} * \boldsymbol{w}}\right). \quad (2.2)$$

For *large-margin classifier learning*, we aim to find $h_{\boldsymbol{w}^*}$ defined by $\boldsymbol{w}^* = \operatorname{argmax}_{\boldsymbol{w} \in \mathcal{C}} \operatorname{margin}_{\mathcal{S}}(\boldsymbol{w})$, where $\mathcal{C} = \{\boldsymbol{w} \in \mathbb{R}^{d+1} : \boldsymbol{w} * \boldsymbol{w} < 0\}$ imposes a *nonconvex* constraint. This makes the problem computationally intractable using classical methods, unlike its Euclidean counterpart.

# 3 Hyperbolic linear separator learning

The first step towards the goal of learning a large-margin classifier is to establish that we can *provably* learn *some* separator. To this end, we present a hyperbolic version of the classic perceptron algorithm and establish that it will converge on data that is separable with a margin.

## 3.1 The hyperbolic perceptron algorithm

The hyperbolic perceptron (cf. Alg. 1) learns a binary classifier $\boldsymbol{w}$ with respect to the Minkowski product. This is implemented in the update rule $\boldsymbol{v}_t \leftarrow \boldsymbol{w}_t + y\boldsymbol{x}$, similar to the Euclidean case. In contrast to the Euclidean case, the hyperbolic perceptron requires an additional normalization step $\boldsymbol{w}_{t+1} \leftarrow \boldsymbol{v}_t/\sqrt{-\boldsymbol{v}_t * \boldsymbol{v}_t}$. This ensures that $\boldsymbol{w}_{t+1} * \boldsymbol{w}_{t+1} < 0$; as a result $\boldsymbol{w}_{t+1}$ defines a meaningful classifier, i.e., $\mathbb{L}^d \cap \partial \mathcal{H}_{\boldsymbol{w}_{t+1}} \neq \emptyset$. For more details, see Appendix B.

While intuitive, it remains to establish that this algorithm converges, i.e., finds a solution which correctly classifies all the training samples. To this end, consider the following notion of *hyperbolic linear separability with a margin*: for $X, X' \subseteq \mathbb{L}^d$, we say that $X$ and $X'$ are linearly separable with (hyperbolic) margin $\gamma_H$, if there exists a $\boldsymbol{w} \in \mathbb{R}^{d+1}$ with $\sqrt{-\boldsymbol{w} * \boldsymbol{w}} = 1$ such that $\boldsymbol{w} * \boldsymbol{x} > \sinh(\gamma_H) \ \forall \ \boldsymbol{x} \in X$ and $\boldsymbol{w} * \boldsymbol{x}' < -\sinh(\gamma_H) \ \forall \ \boldsymbol{x}' \in X'$. Assuming our training set is separable with a margin, the hyperbolic perceptron has the following convergence guarantee:

**Theorem 3.1.** *Assume that there is some $\bar{\boldsymbol{w}} \in \mathbb{R}^{d+1}$ with $\sqrt{-\bar{\boldsymbol{w}} * \bar{\boldsymbol{w}}} = 1$ and $\boldsymbol{w}_0 * \bar{\boldsymbol{w}} \leq 0$, and some $\gamma_H > 0$, such that $y_j(\bar{\boldsymbol{w}} * \boldsymbol{x}_j) \geq \sinh(\gamma_H)$ for $j = 1, \ldots, |\mathcal{S}|$. Then, Alg. 1 converges in $O\left(1/\sinh(\gamma_H)\right)$ steps and returns a solution with margin $\gamma_H$.*

| **Algorithm 1** Hyperbolic perceptron | **Algorithm 2** Adversarial Training |
|---|---|
| 1: Initialize $\boldsymbol{w}_0 \in \mathbb{R}^{d+1}$. | 1: Initialize $\boldsymbol{w}_0 = 0$, $\mathcal{S}' \leftarrow \emptyset$. |
| 2: **for** $t = 0, 1, \ldots, T-1$ **do** | 2: **for** $t = 0, 1, \ldots, T-1$ **do** |
| 3:     **for** $j = 1, \ldots, n$ **do** | 3:     $\mathcal{S}_t \sim \mathcal{S}$ iid with $|\mathcal{S}_t| = m$; $\mathcal{S}'_t \leftarrow \emptyset$. |
| 4:         **if** $\mathrm{sgn}(\boldsymbol{x}_j * \boldsymbol{w}_t) \neq y_j$ **then** | 4:     **for** $i = 0, 1, \ldots, m$ **do** |
| 5:             $\boldsymbol{v}_t \leftarrow \boldsymbol{w}_t + y_j \boldsymbol{x}_j$ | 5:         $\tilde{\boldsymbol{x}}_i \leftarrow \mathrm{argmax}_{d_{\mathbb{L}}(\boldsymbol{x}_i, \boldsymbol{z}) \leq \alpha}\, l(\boldsymbol{z}, y_i; \boldsymbol{w}_t)$ |
| 6:             $\boldsymbol{w}_{t+1} \leftarrow \boldsymbol{v}_t / \min\{1, \sqrt{-\boldsymbol{v}_t * \boldsymbol{v}_t}\}$ | 6:     **end for** |
| 7:             **break** | 7:     $\mathcal{S}'_t \leftarrow \{(\tilde{\boldsymbol{x}}_i, y_i)\}_{i=1}^m$ |
| 8:         **end if** | 8:     $\mathcal{S}' \leftarrow \mathcal{S}' \cup \mathcal{S}'_t$ |
| 9:     **end for** | 9:     $\boldsymbol{w}_{t+1} \leftarrow \mathcal{A}(\boldsymbol{w}_t, \mathcal{S}, \mathcal{S}')$ |
| 10: **end for** | 10: **end for** |
| 11: Output: $\boldsymbol{w}_T$ | 11: Output: $\boldsymbol{w}_T$ |

The proof of Thm. 3.1 (provided in Appendix B) follows the standard proof of the Euclidean perceptron and utilizes the Cauchy-Schwartz inequality for the Minkowski product. To verify that Algorithm 1 always converges to a valid hyperplane, i.e., such that $\mathbb{L}^d \cap \mathcal{H}_{\boldsymbol{v}_t} \neq \emptyset$, which happens iff $\boldsymbol{v}_t * \boldsymbol{v}_t < 0$, consider the following argument:

$$\boldsymbol{v}_t * \boldsymbol{v}_t = (\boldsymbol{w}_t + y_j \boldsymbol{x}_j) * (\boldsymbol{w}_t + y_j \boldsymbol{x}_j) = \underbrace{\boldsymbol{w}_t * \boldsymbol{w}_t}_{\substack{(i)\\ \leq -1}} + 2 \underbrace{y_j(\boldsymbol{x}_j * \boldsymbol{w}_t)}_{\substack{(ii)\\ <0}} + \underbrace{y^2}_{=1} \underbrace{(\boldsymbol{x}_j * \boldsymbol{x}_j)}_{\substack{(iii)\\ =1}} < 0 \ .$$

Here, $(i)$ is a consequence of the normalization step in Algorithm 1 and $(iii)$ follows as $\boldsymbol{x}_j * \boldsymbol{x}_j = 1$, since $\boldsymbol{x}_j \in \mathbb{L}^d$. As for $(ii)$, note that we perform the update $\boldsymbol{v}_t \leftarrow \boldsymbol{w}_t + y_j \boldsymbol{x}_j$ only when $y_j \neq \mathrm{sign}(\boldsymbol{x}_j * \boldsymbol{w})$ (cf. Algorithm 1).

**Remark 3.2.** Note that the perceptron algorithm in Euclidean space exhibits a $O(1/\gamma_E^2)$ convergence rate [24], where $\gamma_E$ denotes the Euclidean margin. When $\gamma_E \sim 0$, the $1/\sinh(\gamma_H)$ convergence rate for hyperbolic spaces can be significantly faster than $1/\gamma_E^2$, indicating that exploiting the structure of hyperbolic space can be beneficial.

## 3.2 The challenge of large-margin learning

Thm. 3.1 establishes that the hyperbolic perceptron converges to *some* linear separator. However, for the purposes of generalization, one would ideally like to converge to a *large-margin* separator. As with the classic Euclidean perceptron, no such guarantee is possible for the hyperbolic perceptron; this motivates us to ask whether a suitable modification can rectify this.

Drawing inspiration from the Euclidean setting, a natural way to proceed is to consider the use of *margin losses*, such as the logistic or hinge loss. Formally, let $l \colon \mathcal{X} \times \{\pm 1\} \to \mathbb{R}_+$ be a loss function:

$$l(\boldsymbol{x}, y; \boldsymbol{w}) = f(y \cdot (\boldsymbol{w} * \boldsymbol{x})), \tag{3.1}$$

where $f \colon \mathbb{R} \to \mathbb{R}_+$ is some convex, non-increasing function, e.g., the hinge loss. The empirical risk of the classifier parameterized by $\boldsymbol{w}$ on the training set $\mathcal{S} \subset \mathcal{X} \times \{\pm 1\}$ is $L(\boldsymbol{w}; \mathcal{S}) = \sum_{(\boldsymbol{x}, y) \in \mathcal{S}} l(\boldsymbol{x}, y; \boldsymbol{w}) / |\mathcal{S}|$. Commonly, we learn a classifier by minimizing $L(\boldsymbol{w}; \mathcal{S})$ via gradient descent with iterates

$$\boldsymbol{w}_{t+1} \leftarrow \boldsymbol{w}_t - \eta \sum_{(\boldsymbol{x}, y) \in \mathcal{S}} \nabla l(\boldsymbol{x}, y; \boldsymbol{w}_t) / |\mathcal{S}| \ , \tag{3.2}$$

where $\eta > 0$ denotes the learning rate. Unfortunately, while this will yield a large-margin solution, the following result demonstrates that the number of iterations required may be prohibitively large.

**Theorem 3.3.** *Let $\boldsymbol{e}_i \in \mathbb{R}^{d+1}$ be the $i$-th standard basis vector. Consider the training set $\mathcal{S} = \{(\boldsymbol{e}_1, 1), (-\boldsymbol{e}_1, -1)\}$ and the initialization $\boldsymbol{w}_0 = \boldsymbol{e}_2$. Suppose $\{\boldsymbol{w}_t\}_{t \geq 0}$ is a sequence of iterates in (3.2). Then, the number of iterations needed to achieve margin $\gamma_H$ is $\overline{\Omega}(\exp(\gamma_H))$.*

While this result is disheartening, fortunately, we now present a simple resolution: by suitably adding *adversarial examples*, the gradient descent converges to a large-margin solution in *polynomial time*.

## 4 Hyperbolic large-margin separator learning via adversarial examples

Thm. 3.3 reveals that gradient descent on a margin loss is insufficient to efficiently obtain a large-margin classifier. Adapting the approach proposed in [5] for the Euclidean setting, we show how to

| Space | Perceptron | Adversarial margin | Adversarial ERM | Adversarial GD |
|-------|-----------|--------------------|-----------------|----------------|
| Euclidean (prior work) | $O\left(\frac{1}{\gamma_E^2}\right)$ | $\gamma_E - \alpha$ | $\Omega\left(\exp(d)\right)$ | $\Omega\left(\text{poly}\left(\frac{1}{\gamma_E - \alpha}\right)\right)$ |
| Hyperbolic (this paper) | $O\left(\frac{1}{\sinh(\gamma_H)}\right)$ | $\frac{\gamma_H}{\cosh(\alpha)}$ | $\Omega\left(\exp(d)\right)$ | $\Omega\left(\text{poly}\left(\frac{\cosh(\alpha)}{\sinh(\gamma_H)}\right)\right)$ |

Table 1: Comparison between Euclidean and hyperbolic spaces for Perceptron (cf. Alg. 1) and adversarial training (cf. Alg. 2). Recall that $\gamma_{E/H}$, $\alpha$, and $d$ denote the (Euclidean/ hyperbolic) margin of the training data, the adversarial perturbation budget, and the underlying dimension, respectively.

alleviate this problem by enriching the training set with *adversarial examples* before updating the classifier (cf. Alg. 2). In particular, we minimize a robust loss

$$\min_{\boldsymbol{w}\in\mathbb{R}^{d+1}} \ L_{\text{rob}}(\boldsymbol{w};\mathcal{S}) := \frac{1}{|\mathcal{S}|}\sum_{(\boldsymbol{x},y)\in\mathcal{S}} l_{\text{rob}}(\boldsymbol{x},y;\boldsymbol{w}) \tag{4.1}$$

$$l_{\text{rob}}(\boldsymbol{x},y;\boldsymbol{w}) := \max_{\substack{\boldsymbol{z}\in\mathbb{L}^d: \\ d_{\mathbb{L}}(\boldsymbol{x},\boldsymbol{z})\leq\alpha}} l(\boldsymbol{z},y;\boldsymbol{w}) \ . \tag{4.2}$$

The problem has a minimax structure, where the outer optimization minimizes the training error over $\mathcal{S}$. The inner optimization generates an adversarial example by perturbing a given input feature $\boldsymbol{x}$ on the hyperbolic manifold. Note that the magnitude of the perturbation added to the original example is bounded by $\alpha$, which we refer to as the *adversarial budget*. In particular, we want to construct a perturbation that maximizes the loss $l$, i.e., $\tilde{\boldsymbol{x}} \leftarrow \text{argmax}_{d_{\mathbb{L}}(\boldsymbol{x},\boldsymbol{z})\leq\alpha} l(\boldsymbol{z},y;\boldsymbol{w})$. In this paper, we restrict ourselves to only those adversarial examples that lead to misclassification with respect to the current classifier, i.e., $h_{\boldsymbol{w}}(\boldsymbol{x}) \neq h_{\boldsymbol{w}}(\tilde{\boldsymbol{x}})$ (cf. Remark 4.2).

Adversarial example $\tilde{\boldsymbol{x}}$ can be generated efficiently by reducing the problem to an (Euclidean) linear program with a spherical constraint:

$$\text{(CERT)} \ \max_{\boldsymbol{z}\in\mathbb{R}^d} \ -w_0 z_0 + \sum_{i=1}^{d} w_i z_i \quad \text{s.t.} \ \sum_{i=1}^{d} -x_i z_i \leq \cosh(\alpha) - x_0 z_0, \ \|\boldsymbol{z}_{\backslash 0}\|^2 = z_0^2 - 1 \ . \tag{4.3}$$

Importantly, as detailed in Appendix C.2 and summarized next, (CERT) can be solved in closed-form.

**Theorem 4.1.** *Given the input example $(\boldsymbol{x},y)$, let $\boldsymbol{x}_{\backslash 0} = (x_1,\ldots,x_d)$. We can efficiently compute a solution to* CERT *or decide that no solution exists. If a solution exists, then based on a guess of $z_0$, the solution has the form $\tilde{\boldsymbol{x}} = \left(z_0, \sqrt{z_0^2-1}\left(b_\alpha\check{\boldsymbol{x}} + \sqrt{1-b_\alpha^2}\check{\boldsymbol{x}}^\perp\right)\right)$. Here, $b_\alpha$ depends on the adversarial budget $\alpha$, and $\check{\boldsymbol{x}}^\perp$ is a unit vector orthogonal to $\check{\boldsymbol{x}} = -\boldsymbol{x}_{\backslash 0}/\|\boldsymbol{x}_{\backslash 0}\|$ along $\boldsymbol{w}$.*

**Remark 4.2.** Note that according to Thm. 4.1, it is possible that, for a particular guess of $z_0$, we may not be able to find an adversarial example $\tilde{\boldsymbol{x}}$ that leads to a prediction that is inconsistent with $\boldsymbol{x}$, i.e., $h_{\boldsymbol{w}}(\boldsymbol{x}) \neq h_{\boldsymbol{w}}(\tilde{\boldsymbol{x}})$. Thus, for some $t$, we may have $|\mathcal{S}_t'| < m$ in Alg. 2.

The minimization with respect to $\boldsymbol{w}$ in (4.1) can be performed by an iterative optimization procedure, which generates a sequence of classifiers $\{\boldsymbol{w}_t\}$. We update the classifier $\boldsymbol{w}_t$ according to an update rule $\mathcal{A}$, which accepts as input the current estimate of the weight vector, the original training set, and an adversarial perturbation of the training set. The update rule produces as output a weight vector which approximately minimizes the robust loss $L_{\text{rob}}$ in (4.1).

We now establish that for a gradient based update rule, the above adversarial training procedure will efficiently converge to a large-margin solution. Table 1 summarizes the results of this section and compares with the corresponding results in the Euclidean setting [5].

## 4.1 Fast convergence via gradient-based update

Consider Alg. 2 with $\mathcal{A}(\boldsymbol{w}_t, \mathcal{S}, \mathcal{S}_t')$ being a gradient-based update with learning rate $\eta_t > 0$:

$$\boldsymbol{w}_{t+1} \leftarrow \boldsymbol{w}_t - \eta_t/|\mathcal{S}_t'| \cdot \sum_{(\tilde{\boldsymbol{x}},y)\in\mathcal{S}_t'} \nabla_{\boldsymbol{w}_t} l(\tilde{\boldsymbol{x}},y;\boldsymbol{w}_t) \ ; \quad \boldsymbol{w}_{t+1} \leftarrow \boldsymbol{w}_{t+1}/\sqrt{-\boldsymbol{w}_{t+1}*\boldsymbol{w}_{t+1}} \ , \tag{4.4}$$

where the normalization is performed to ensure that the update remains valid, i.e., $\mathbb{L}^d \cap \partial\mathcal{H}_{\boldsymbol{w}} \neq \emptyset$.

To compute the update, we need to compute gradients of the outer minimization problem, i.e., $\nabla_{\boldsymbol{w}} l_{\text{rob}}$ over $\mathcal{S}_t'$ (cf. (4.1)). However, this function is itself a maximization problem. We therefore compute

the gradient at the maximizer of this inner maximization problem. Danskin's Theorem [9, 1] ensures that this gives a valid descent direction. Given the closed form expression for the adversarial example $\tilde{\boldsymbol{x}}$ as per Thm. 4.1, the gradient of the loss is

$$\nabla_{\boldsymbol{w}} \, l(\tilde{\boldsymbol{x}}, y; \boldsymbol{w}) = f'(y(\boldsymbol{w} * \tilde{\boldsymbol{x}})) \cdot \nabla_{\boldsymbol{w}} \, y(\boldsymbol{w} * \tilde{\boldsymbol{x}}) = f'(y(\boldsymbol{w} * \tilde{\boldsymbol{x}})) \cdot y\widehat{\tilde{\boldsymbol{x}}}^{\mathrm{T}},$$

where $y\widehat{\tilde{\boldsymbol{x}}}^{\mathrm{T}} = y(\tilde{x}_0, -\tilde{x}_1, \ldots, -\tilde{x}_n)^{\mathrm{T}}$. With Danskin's theorem, $\nabla l(\tilde{\boldsymbol{x}}, y; \boldsymbol{w}) \in \partial l_{\mathrm{rob}}(\boldsymbol{x}, y; \boldsymbol{w})$, so we can compute the descent direction and perform the step in (4.4). We defer details to Appendix C.4.

### 4.1.1 Convergence analysis

We now establish that the above gradient-based update converges to a large-margin solution in polynomial time. For this analysis, we need the following assumptions:

**Assumption 1.** 1. The training set $\mathcal{S}$ is linearly separable with margin at least $\gamma_H$, i.e., there exists a $\bar{\boldsymbol{w}} \in \mathbb{R}^{d+1}$, such that $y(\bar{\boldsymbol{w}} * \boldsymbol{x}) \geq \sinh(\gamma_H)$ for all $(\boldsymbol{x}, y) \in \mathcal{S}$.

2. There exists constants $R_x, R_w \geq 0$, such that (i) $\|\boldsymbol{x}\| \leq R_x$; (ii) all possible adversarial perturbations remain within this constraint, i.e., $\|\tilde{\boldsymbol{x}}\| \leq R_x$; and (iii) $\|\boldsymbol{w}\| \leq R_w$. Let $R_\alpha := R_x R_w$.

3. the function $f(s)$, underlying the loss (cf. (3.1)), has the following properties: (i) $f(s) > 0 \, \forall \, s$; (ii) $f'(s) < 0 \, \forall \, s$; (iii) $f$ is differentiable, and (iv) $f$ is $\beta$-smooth.

In the rest of the section, we work with the following hyperbolic equivalent of the *logistic regression loss* that fulfills Assumption 1:

$$l(\boldsymbol{x}, y; \boldsymbol{w}) = \ln\left(1 + \exp\left(-\operatorname{asinh}\left(y(\boldsymbol{w} * \boldsymbol{x})/2R_\alpha\right)\right)\right) \, , \tag{4.5}$$

where $R_\alpha$ is as defined in Assumption 1. Other loss functions as well as the derivation of the hyperbolic logistic regression loss are discussed in Appendix C.1.

We first show that Alg. 2 with a gradient update is guaranteed to converge to a large-margin classifier.

**Theorem 4.3.** *With constant step size and $\mathcal{A}$ being the GD update with an initialization $\boldsymbol{w}_0$ with $\boldsymbol{w}_0 * \boldsymbol{w}_0 < 0$, $\lim_{t \to \infty} L(\boldsymbol{w}_t; \mathcal{S} \cup \mathcal{S}'_t) = 0$.*

The proof can be found in Appendix C.4. While this result guarantees convergence, it does not guarantee efficiency (e.g., by showing a polynomial convergence rate). The following result shows that Alg. 2 with a gradient based update obtains a max-margin classifier in polynomial time.

**Theorem 4.4.** *For a fixed constant $c \in (0, 1)$, let $\eta_t = \eta := c \cdot \frac{2\sinh^2(\gamma_H)}{\beta\sigma_{\max}^2\cosh^2(\alpha)R_\alpha^2}$ with $\sigma_{\max}$ denoting an upper bound on the maximum singular value of the data matrix $\sum_{\boldsymbol{x} \in \mathcal{S}'} \boldsymbol{x}\boldsymbol{x}^T$, and $\mathcal{A}$ the GD update as defined in (4.4). Then, Alg. 2 achieves the margin $\gamma_H/\cosh(\alpha)$ in $\Omega\big(\operatorname{poly}\left(\cosh(\alpha)/\sinh(\gamma_H)\right)\big)$ steps.*

Below, we briefly sketch some of the proof ideas, but defer a detailed exposition to Appendix C.4 (cf. Thm. C.13 and C.14). To prove the gradient-based convergence result, we first analyze the convergence of an "adversarial perceptron", that resembles the adversarial GD in that it performs updates of the form $\boldsymbol{w}_{t+1} \leftarrow \boldsymbol{w}_t + y\tilde{\boldsymbol{x}}$. We then extend the analysis to the adversarial GD, where the convergence analysis builds on classical ideas from convex optimization.

The following auxiliary lemma relates the adversarial margin to the max-margin classifier.

**Lemma 4.5.** *Let $\bar{\boldsymbol{w}}$ be the max-margin classifier of $\mathcal{S}$ with margin $\gamma_H$. At each iteration of Algorithm 2, $\bar{\boldsymbol{w}}$ linearly separates $\mathcal{S} \cup \mathcal{S}'$ with margin at least $\frac{\gamma_H}{\cosh(\alpha)}$.*

The result follows from geometric arguments, as discussed in Section C.3. With the help of this lemma, we can show the following bound on the sample complexity of the adversarial perceptron:

**Theorem 4.6.** *Assume that there is some $\bar{\boldsymbol{w}} \in \mathbb{R}^{d+1}$ with $\sqrt{-\bar{\boldsymbol{w}} * \bar{\boldsymbol{w}}} = 1$ and $\boldsymbol{w}_0 * \bar{\boldsymbol{w}} \leq 0$, and some $\gamma_H > 0$, such that $y_j(\bar{\boldsymbol{w}} * \boldsymbol{x}_j) \geq \sinh(\gamma_H)$ for $j = 1, \ldots, |\mathcal{S}|$. Then, the adversarial perceptron (with adversarial budget $\alpha$) converges after $O\left(\frac{\cosh(\alpha)}{\sinh(\gamma_H)}\right)$ steps, at which it has margin of at least $\frac{\gamma_H}{\cosh(\alpha)}$.*

The technical proof can be found in Section C.3.

## 4.2 On the necessity of combining gradient descent and adversarial training

We remark here that the enrichment of the training set with adversarial examples is critical for the polynomial-time convergence. Recall first that by Thm. 3.3, without adversarial training, we can construct a simple max-margin problem that cannot be solved in polynomial time. Interestingly, merely using adversarial examples by themselves does not suffice for fast convergence either.

Consider Alg. 2 with an ERM as the update rule $\mathcal{A}(\boldsymbol{w}_t, \mathcal{S}, \mathcal{S}')$. In this case, the iterate $\boldsymbol{w}_{t+1}$ corresponds to an ERM solution for $\mathcal{S} \cup \mathcal{S}'$, i.e.,

$$\boldsymbol{w}_{t+1} \leftarrow \mathrm{argmin}_{\boldsymbol{w}} \sum\nolimits_{(\boldsymbol{x},y) \in \mathcal{S} \cup \mathcal{S}'} l(\boldsymbol{x}, y; \boldsymbol{w}) . \qquad (4.6)$$

Let $\mathcal{S}_t = \mathcal{S}$, i.e., we utilize the full power of the adversarial training in each step. The following result reveals that even under this optimistic setting, Alg. 2 may not converge to a solution with a non-trivial margin in polynomial time:

**Theorem 4.7.** *Suppose Alg. 2 (with an ERM update) outputs a linear separator of $\mathcal{S} \cup \mathcal{S}'$. In the worst case, the number of iteration required to achieve a margin at least $\epsilon$ is $\Omega\left(\exp(d)\right)$.*

We note that a similar result in the Euclidean setting appears in [5]. We establish Thm. 4.7 by extending the proof strategy of [5, Thm. 4] to hyperbolic spaces. In particular, given a spherical code in $\mathbb{R}^d$ with $T$ codewords and $\theta \sim \sinh(\epsilon)\cosh(\alpha)$ minimum separation, we construct a training set $\mathcal{S}$ and subsequently the adversarial examples $\{\mathcal{S}'_t\}$ such that there exists a sequence of ERM solutions $\{\boldsymbol{w}_t\}_{t \leq T}$ on $\mathcal{S} \cup \mathcal{S}'$ (cf. (4.6)) that has margin less than $\epsilon$. Now the result in Thm. 4.7 follows by utilizing a lower bound [7] on the size of the spherical code with $T = \Omega\left(\exp(d)\right)$ codewords and $\theta \sim \sinh(\epsilon)\cosh(\alpha)$ minimum separation. The proof of Thm. 4.7 is in Appendix C.5.

## 5 Dimension-distortion trade-off

So far we have focused on classifying data that is given in either Euclidean spaces $\mathbb{R}^d$ or Lorentz space $\mathbb{L}^{d'}$. Now, consider data $(\mathcal{X}, d_{\mathcal{X}})$ with similarity metric $d_{\mathcal{X}}$ that was embedded into the respective spaces. We assume access to maps $\phi_E : \mathcal{X} \to \mathbb{R}^d$ and $\phi_H : \mathcal{X} \to \mathbb{L}^{d'}_+$ that embed $\mathcal{X}$ into the Euclidean space $\mathbb{R}^d$ and the upper sheet of the Lorentz space $\mathbb{L}^{d'}_+$, respectively (cf. Remark A.1). Let $c_E$ and $c_H$ denote the multiplicative distortion induced by $\phi_E$ and $\phi_H$, respectively (cf. § 2.2). Upper bounds on $c_E$ and $c_H$ can be estimated based on the structure of $\mathcal{X}$ and the embedding dimensions.

In this section, we address the natural question: *How does the distortion $c_E, c_H$ impact our guarantees on the margin?* In the previous sections, we noticed that some of the guarantees scale with the dimension of the embedding space. Therefore, we want to analyze the trade-off between the higher distortion resulting from working with smaller embedding dimensions and the higher cost of training robust models due to working with larger embedding dimensions.

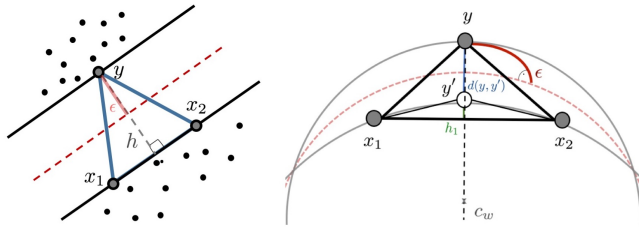

Figure 2: Margin as distance between support vectors. **Left:** Euclidean. **Right:** Hyperbolic.

We often encounter data sets in ML applications that are intrinsically hierarchical. Theoretical results on the embeddability of trees (cf. § 2.2) suggest that hyperbolic spaces are especially suitable to represent hierarchical data. We therefore restrict our analysis to such data. Further, we make the following assumptions on the underlying data $\mathcal{X}$ and the embedding maps, respectively.

**Assumption 2.** (1) Both $\phi_H(\mathcal{X})$ and $\phi_E(\mathcal{X})$ are linearly separable in the respective spaces, and (2) $\mathcal{X}$ is hierarchical, i.e., has a partial order relation.

**Assumption 3.** The maps $\phi_H, \phi_E$ preserve the partial order relation in $\mathcal{X}$ and the root is mapped onto the origin of the embedding space.

Towards understanding the impact of the distortion of the embedding maps $\phi_H$ and $\phi_E$ on margin, we relate the distance between the support vectors to the size of the margin. The distortion of these distances via embedding then gives us the desired bounds on the margin.

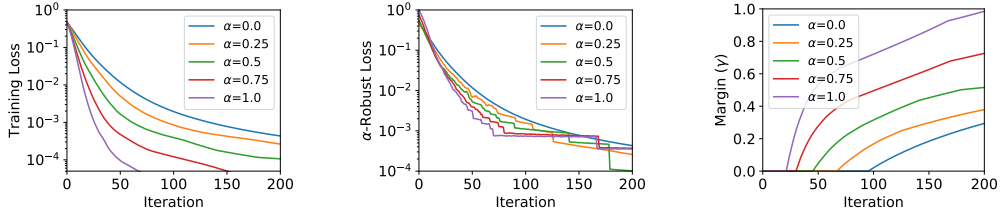

Figure 3: Performance of Adversarial GD. **Left:** Loss $L(\boldsymbol{w})$ on the original data. **Middle:** $\alpha$-robust loss $L_\alpha(\boldsymbol{w})$. **Right:** Hyperbolic margin $\gamma_H$. We vary the adversarial budget $\alpha$ over $\{0, 0.25, 0.5, 0.75, 1.0\}$. Note that $\alpha = 0$ corresponds to the state of the art [6].

## 5.1 Euclidean case

In the Euclidean case, we relate the distance of the support vectors to the size of the margin via triangle relations. Let $\boldsymbol{x}, \boldsymbol{y} \in \mathbb{R}^d$ denote support vectors, such that $\langle \boldsymbol{x}, \boldsymbol{w} \rangle > 0$ and $\langle \boldsymbol{y}, \boldsymbol{w} \rangle < 0$ and $\mathrm{margin}(\boldsymbol{w}) = \epsilon$. Note that we can rotate the decision boundary, such that the support vectors are not unique. So, without loss of generality, assume that $\boldsymbol{x}_1, \boldsymbol{x}_2$ are equidistant from the decision boundary and $\|\boldsymbol{w}\| = 1$ (cf. Fig. 2(left)). In this setting, we show the following relation between the margin with and without the influence of distortion:

**Theorem 5.1.** *Let $\epsilon'$ and $\epsilon$ denote the margin with and without distortion, respectively. If $\mathcal{X}$ is a tree embedded into $\mathbb{R}^{O(\log^2 |\mathcal{X}|)}$, then $\epsilon' = O\left(\epsilon / \log^3 |\mathcal{X}|\right)$.*

The proof of Thm. 5.1 follows from a simple side length-altitude relations in the Euclidean triangle between support vectors (cf. Fig. 2(left)) and a simple application of Bourgain's result on embedding trees into $\mathbb{R}^d$. For more details see Appendix D.1.

## 5.2 Hyperbolic case

As in the Euclidean case, we want to relate the margin to the pairwise distances of the support vectors. Such a relation can be constructed both in the original and in the embedding space, which allows us to study the influence of distortion on the margin in terms of $c_H$. In the following, we will work with the half-space model $\mathbb{P}^{d'}$ (cf. Appendix A.1). However, since the theoretical guarantees in the rest of the paper consider the Lorentz model $\mathbb{L}_+^{d'}$, we have to map between the two spaces. We show in Appendix D.2 that such a mapping exists and preserves the Minkowski product, following [6].

The hyperbolic embedding $\phi_H$ has two sources of distortion: (1) the multiplicative distortion of pairwise distances, measured by the factor $1/c_H$; and (2) the distortion of order relations, in most embedding models captured by the alignment of ranks with the Euclidean norm. Under Assumption 3, order relationships are preserved and the root is mapped to the origin. Therefore, for $x \in \mathcal{X}$, the distortion on the Euclidean norms is given as $\|\phi_H(x)\| = d_E(\phi_H(x), \phi_H(0)) = d_\mathcal{X}(x, 0)/c_H$, i.e., the distortion on both pairwise distances and norms is given by a factor $1/c_H$.

In $\mathbb{P}^{d'}$, the decision hyperplane corresponds to a hypercircle $\mathcal{K}_w$. We express its radius $r_w$ in terms of the hyperbolic distance between a point on the decision boundary and one of the hypercircle's ideal points [6]. The support vectors $\boldsymbol{x}, \boldsymbol{y}$ lie on hypercircles $\mathcal{K}_x$ and $\mathcal{K}_y$, which correspond to the set of points of hyperbolic distance $\epsilon$ (i.e., the margin) from the decision boundary. We again assume, without loss of generality, that at least one support vector is not unique and let $\boldsymbol{x}_1, \boldsymbol{x}_2 \in \mathcal{K}_x$ and $\boldsymbol{y} \in \mathcal{K}_y$ (cf. Fig. 2(right)). We now show that the distortion introduced by $\phi_H$ has a negligible effect on the margin.

**Theorem 5.2.** *Let $\epsilon'$ and $\epsilon$ denote the margin with and without distortion, respectively. If $\mathcal{X}$ is a tree embedded into $\mathbb{L}_+^2$, then $\epsilon' \approx \epsilon$.*

The technical proof relies on a construction that reduces the problem to Euclidean geometry via circle inversion on the decision hypercircle. We defer all details to Appendix D.2.

## 6 Experiments

We now present empirical studies for hyperbolic linear separator learning to corroborate our theory. In particular, we evaluate our proposed Adversarial GD algorithm (§4) on data that is linearly separable in hyperbolic space and compare with the state of the art [6]. Furthermore, we analyze dimension-

distortion trade-offs (§5). As in our theory, we train hyperbolic linear classifiers $\boldsymbol{w}$ whose prediction on $\boldsymbol{x}$ is $y = \text{sgn}(\boldsymbol{w} * \boldsymbol{x})$. Additional experimental results can be found in Appendix F.

We emphasise that our focus in this work is in theoretically understanding the benefits of hyperbolic spaces for classification. The above experiments serve to illustrate our theoretical results, which as a starting point were derived for linear models and separable data. While extensions to non-separable data and non-linear models are of practical interest, a detailed study is left for future work.

**Data.** We use the ImageNet ILSVRC 2012 dataset [26] along with its label hierarchy from wordnet. Hyperbolic embeddings in Lorentz space are obtained for the internal label nodes and leaf image nodes using Sarkar's construction [28]. For the first experiment, we verify the effectiveness of adversarial learning by picking two classes (n09246464 and n07831146), which allows for a data set that is linearly separable in hyperbolic space. In this set, there were 1,648 positive and 1,287 negative examples. For the second experiment, to showcase better representational power of hyperbolic spaces for hierarchical data, we pick two disjoint subtrees (n00021939 and n00015388) from the hierarchy.

**Adversarial GD.** In the following, we utilize the hyperbolic hinge loss (C.2), (see Appendix F for other loss functions). To verify the effectiveness of adversarial training, we compute three quantities: (i) loss on original data $L(\boldsymbol{w})$, (ii) $\alpha$-robust loss $L_\alpha(\boldsymbol{w})$, and (iii) the hyperbolic margin $\gamma$. We vary the budget $\alpha$ over $\{0, 0.25, 0.5, 0.75, 1.0\}$, where $\alpha = 0$ corresponds to the setup in [6]. For a given budget, we obtain adversarial examples by solving the CERT problem (4.3), which is feasible for $z_0 \in (x_0 \cosh(\alpha) - \Delta, \ x_0 \cosh(\alpha) + \Delta)$, where $\Delta = \sqrt{(x_0^2 - 1)(\cosh^2(\alpha) - 1)}$. We do a binary search in this range for $z_0$, solve CERT and check if we can obtain an adversarial example. We utilize the adversarial examples we find, and ignore other data points. In all experiments, we use a constant step-size $\eta_t = 0.01 \ \forall t$. The results are shown in Fig. 3. As $\alpha$ increases the problem becomes harder to solve (higher training robust loss) but we achieve a better margin. Notably, we observe strong performance gains over the training procedures without adversarial examples [6].

**Dimensional efficiency.** In this experiment, we illustrate the benefit of using hyperbolic space when the underlying data is truly hierarchical. To be more favourable to Euclidean setting, we subsample images from each subtree, such that in total we have 1000 vectors. We obtain Euclidean embeddings following the setup and code from Nickel and Kiela [22]. The Euclidean embeddings in 16 dimensions reach a mean rank (MR) $\leq 2$, which indicates reasonable quality in preserving distance to few-hop neighbors. We observe superior classification performance at much lower dimensions by leveraging hyperbolic space (see Table 2 in Appendix F.3). In particular, our hyperbolic classifier achieves zero test error on 8-dimensional embeddings, whereas Euclidean logistic regression struggles even with 16-dimensional embeddings. This is consistent with our theoretical results (§5): Due to high distortion, lower-dimensional Euclidean embeddings struggle to capture the global structure among the data points that makes the data points easily separable.

## 7 Conclusion and future work

We studied the problem of learning robust classifiers with large margins in hyperbolic space. We introduced and analyzed a hyperbolic perceptron algorithm. Moreover, we explored multiple adversarial approaches to robust large-margin learning. The second part of the paper analyzed the role of geometry in learning robust classifiers. We compared Euclidean and hyperbolic approaches with respect to the intrinsic geometry of the data. For hierarchical data that embeds well into hyperbolic space, the lower embedding dimension ensures superior guarantees when learning the classifier in hyperbolic space. This result suggests that it can be highly beneficial to perform downstream machine learning and optimization tasks in a space that naturally reflects the intrinsic geometry of the data. Promising avenues for future research include (i) exploring the practicality of these results in broader machine learning and data science applications; and (ii) studying other related methods in non-Euclidean spaces, together with an evaluation of dimension-distortion trade-offs.

## Acknowledgements

We thank Pranjal Awasthi for helpful discussions on computing adversarial examples and Eli Chien for helpful comments on an earlier version of the paper.

## Broader Impact

The paper proposes novel algorithms for large-margin learning in hyperbolic space. The paper's scope is theoretical and does not discuss specific applications with societal consequences. Therefore, a discussion of the broader impact of this work is not applicable.

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
