[Supplementary Material 1 · complete_manuscript.pdf]

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

$ and as such locally Euclidean spaces. In the following we introduce basic notation for three popular models of hyperbolic spaces. For a comprehensive overview see Bridson and Haefliger [3].

## A.1 Models of hyperbolic spaces

Figure 4: Models of hyperbolic space: The Lorentz model $\mathbb{L}^d$, the Poincare ball $\mathbb{B}^d$, and the Poincare half-plane $\mathbb{P}^d$.

The Poincare ball defines a hyperbolic space within the Euclidean unit ball, i.e.

$$\mathbb{B}^d = \{\boldsymbol{x} \in \mathbb{R}^d : \|\boldsymbol{x}\| < 1\}$$
$$d_{\mathbb{B}}(\boldsymbol{x}, \boldsymbol{x}') = \mathrm{acosh}\left(1 + 2\frac{\|\boldsymbol{x} - \boldsymbol{x}'\|^2}{(1 - \|\boldsymbol{x}\|^2)(1 - \|\boldsymbol{x}'\|^2)}\right) .$$

Here, $\|\cdot\|$ is the usual Euclidean norm.

The closely related Poincare half-plane model is defined as

$$\mathbb{P}^2 = \{\boldsymbol{x} \in \mathbb{R}^2 : x_1 > 0\}$$
$$d_{\mathbb{P}}(\boldsymbol{x}, \boldsymbol{x}') = \mathrm{acosh}\left(1 + \frac{(x_0' - x_0)^2 + (x_1' - x_1)^2}{2x_1 x_1'}\right) .$$

Note that if $x_0 = x_0'$, the metric simplifies as

$$d_{\mathbb{P}}(\boldsymbol{x}, \boldsymbol{x}') = d_{\mathbb{P}}((x_0, x_1), (x_0, x_1')) = \left| \ln \frac{x_1'}{x_1} \right| .$$

The model can be generalized to higher dimensions with

$$\mathbb{P}^d = \{(x_0, \ldots, x_{d-1}) \in \mathbb{R}^d \mid x_{d-1} > 0\} ,$$

however, we will only use the two-dimensional model $\mathbb{P}^2$ here. We further define the hyperboloid as

$$\mathbb{L}^d = \{\boldsymbol{x} \in \mathbb{R}^{d+1} : \boldsymbol{x} * \boldsymbol{x} = 1\}$$
$$d_{\mathbb{L}}(\boldsymbol{x}, \boldsymbol{x}') = \mathrm{acosh}(\boldsymbol{x} * \boldsymbol{x}') ,$$

where $*$ denotes the Minkowski product $\boldsymbol{x} * \boldsymbol{x}' = x_0 x_0' - \sum_{i=1}^d x_i x_i'$.

**Remark A.1.** The Lorentz model

$$\mathbb{L}^d = \{x \in \mathbb{R}^{d+1} : \boldsymbol{x} * \boldsymbol{x} = 1\} .$$

is also called double-sheet model. We use this more general setting in sections 2-4. For simplicity, we restrict ourselves to the upper sheet

$$\mathbb{L}_+^d = \{\boldsymbol{x} \in \mathbb{R}^{d+1} : \boldsymbol{x} * \boldsymbol{x} = 1, \ x_0 > 0\} ,$$

in section 5. All constructions of mappings between the different models of hyperbolic space can be extended to the double-sheet $\mathbb{L}^d$.

## A.2 Equivalence of different models of hyperbolic spaces

The Poincare ball $\mathbb{B}^d$ and the Lorentz model $\mathbb{L}_+^d$ are equivalent models of hyperbolic space. A mapping is given by

$$\pi_{\mathrm{LB}} : \mathbb{L}_+^d \to \mathbb{B}^d$$
$$\boldsymbol{x} = (x_0, \dots, x_d) \mapsto \left( \frac{x_1}{1 + x_0}, \dots, \frac{x_d}{1 + x_0} \right) .$$

We can further construct a mapping from $\mathbb{B}^d$ to $\mathbb{P}^d$ by inversion on a circle centered at $(-1, 0, \dots, 0)$:

$$\pi_{\mathrm{BP}} : \mathbb{B}^d \to \mathbb{P}^d$$
$$\boldsymbol{x} = (x_0, \dots, x_{d-1}) \mapsto \frac{(2x_1, \dots, 2x_{d-1}, 1 - \|\boldsymbol{x}\|^2)}{1 + 2x_0 + \|\boldsymbol{x}\|^2} .$$

## A.3 Embeddability

When analyzing the dimension-distortion trade-off, we make use of two key results on the embeddability (cf. §2.2) of trees into Euclidean and hyperbolic spaces. We state them below for reference.

**Theorem A.2** ([2]). *An $N$-point metric $\mathcal{X}$ (i.e., $|\mathcal{X}| = N$) embeds into Euclidean space $\mathbb{R}^{O(\log^2 N)}$ with the distortion $c_M = O(\log N)$.*

This bound in Theorem A.2 is tight for trees in the sense that embedding them in a Euclidean space (of any dimension) must incur the distortion $c_m = \Omega(\log N)$ [19].

**Theorem A.3** ([28]). *Tree metrics embed quasi-isometrically with $c_M = O(1 + \epsilon)$ into $\mathbb{H}^d$.*

## A.4 Spherical codes in hyperbolic space

Consider the unit sphere $\mathbb{S}^{d-1} \subseteq \mathbb{R}^d$. A *spherical code* is a subset of $\mathbb{S}^{d-1}$, such that any two distinct elements $\boldsymbol{x}, \boldsymbol{x}'$ are separated by at least an angle $\theta$, i.e. $\langle \boldsymbol{x}, \boldsymbol{x}' \rangle \leq \cos\theta$. We denote the size of the largest code as $A(d, \theta)$.

A similar construction of such "spherical caps" can be obtained in $\mathbb{H}^d$. Note that the induced geometry of these caps is spherical, hence they inherit a spherical geometric structure. This allows in particular the transfer of bounds on $A(d, \theta)$ to hyperbolic space [7]:

**Theorem A.4** (Chabauty, Shannon, Wyner (see, e.g., [29])). $A(d, \theta) \geq (1 + o(1))\sqrt{2\pi d}\frac{\cos\theta}{\sin^{d-1}\theta}$.

# B Hyperbolic Perceptron

In this section we analyze the convergence and generalization properties of the hyperbolic perceptron (cf. Algorithm 1). Note that the update $\boldsymbol{v}_t \leftarrow \boldsymbol{w}_t + y_j \boldsymbol{x}_j$ in Algorithm 1 always leads to a valid hyperplane, i.e., $\mathbb{L}^d \cap \mathcal{H}_{\boldsymbol{v}_t} \neq \emptyset$, which happens iff $\boldsymbol{v}_t * \boldsymbol{v}_t < 0$. This can be verified as follows:

$$\boldsymbol{v}_t * \boldsymbol{v}_t = (\boldsymbol{w}_t + y_j \boldsymbol{x}_j) * (\boldsymbol{w}_t + y_j \boldsymbol{x}_j) = \underbrace{\boldsymbol{w}_t * \boldsymbol{w}_t}_{\substack{(i) \\ \leq -1}} + 2\underbrace{y_j(\boldsymbol{x}_j * \boldsymbol{w}_t)}_{\substack{(ii) \\ < 0}} + \underbrace{y^2}_{=1}\underbrace{(\boldsymbol{x}_j * \boldsymbol{x}_j)}_{\substack{(iii) \\ = 1}} < 0 ,$$

where $(i)$ is a consequence of the normalization step in Algorithm 1 and $(iii)$ follows as $\boldsymbol{x} * \boldsymbol{x} = 1$, since $\boldsymbol{x} \in \mathbb{L}^d$. As for $(ii)$, note that we perform the update $\boldsymbol{v}_t \leftarrow \boldsymbol{w}_t + y_j \boldsymbol{x}_j$ only when $y_j \neq \mathrm{sign}(\boldsymbol{x}_j * \boldsymbol{w})$ (cf. Algorithm 1).

We now restate Theorem 3.1 and present a detailed proof of the result.

**Theorem B.1** (Convergence hyperbolic Perceptron in Algorithm 1 (Theorem 3.1)). *Assume that there is some $\bar{\boldsymbol{w}} \in \mathbb{R}^{d+1}$ with $\sqrt{-\bar{\boldsymbol{w}} * \bar{\boldsymbol{w}}} = 1$ and $\boldsymbol{w}_0 * \bar{\boldsymbol{w}} \leq 0$, and some $\gamma_H > 0$, such that $y_j(\bar{\boldsymbol{w}} * \boldsymbol{x}_j) \geq \sinh(\gamma_H)$ for $j = 1, \dots, |\mathcal{S}|$. Then, Algorithm 1 converges in $O\left(\frac{1}{\sinh(\gamma_H)}\right)$ steps and returns a solution with margin $\gamma_H$.*

*Proof.* Assume wlog $\boldsymbol{w}_0 = (0, 1, 0, \ldots, 0) \in \mathbb{R}^{d+1}$. Then $\boldsymbol{w}_0 * \boldsymbol{w}_0 = -1$, i.e., $\mathbb{L}^d \cap \mathcal{H}_{\boldsymbol{w}_0} \neq \emptyset$. Hence, $\boldsymbol{w}_0$ is a valid initialization. Furthermore, assume that the $t$th error is made at the $j$th sample, i.e. update $\boldsymbol{v}_t \leftarrow \boldsymbol{w}_t + y_j \boldsymbol{x}_j$. For $\boldsymbol{u} \in \mathbb{R}^{d+1}$, let $|\boldsymbol{u}| = \sqrt{-\boldsymbol{u} * \boldsymbol{u}}$.

Now let us consider two cases:

- **Case 1**. In this case, we assume that the normalization is not performed in $t$th step, i.e.,

$$\boldsymbol{w}_{t+1} = \boldsymbol{w}_t + y_j \boldsymbol{x}_j.$$

  Therefore,

$$\boldsymbol{w}_{t+1} * \bar{\boldsymbol{w}} = (\boldsymbol{w}_t + y_j \boldsymbol{x}_j) * \bar{\boldsymbol{w}} = \boldsymbol{w}_t * \bar{\boldsymbol{w}} + y_j (\boldsymbol{x}_j * \bar{\boldsymbol{w}}) \geq \boldsymbol{w}_t * \bar{\boldsymbol{w}} + \underbrace{\gamma'_H}_{:=\sinh(\gamma_H)} . \quad \text{(B.1)}$$

- **Case 2**. In this case, the normalization is performed in the $t$th step of Algorithm 1, i.e.,

$$\boldsymbol{w}_{t+1} = \frac{\boldsymbol{w}_t + y_j \boldsymbol{x}_j}{|\boldsymbol{w}_t + y_j \boldsymbol{x}_j|}.$$

  Thus,

$$\boldsymbol{w}_{t+1} * \bar{\boldsymbol{w}} = \frac{\boldsymbol{w}_t + y_j \boldsymbol{x}_j}{|\boldsymbol{w}_t + y_j \boldsymbol{x}_j|} * \bar{\boldsymbol{w}} \overset{(i)}{\geq} (\boldsymbol{w}_t + y_j \boldsymbol{x}_j) * \bar{\boldsymbol{w}}$$

$$\geq \boldsymbol{w}_t * \bar{\boldsymbol{w}} + \underbrace{y_j (\boldsymbol{x}_j * \bar{\boldsymbol{w}})}_{\geq \gamma'_H} \geq \boldsymbol{w}_t * \bar{\boldsymbol{w}} + \gamma'_H , \quad \text{(B.2)}$$

where $(i)$ follows as the normalization is performed only if $|\boldsymbol{w}_t + y_j \boldsymbol{x}_j| < 1$ and numerator is positive by induction.

By utilizing (B.1) and (B.2), we obtain the following telescoping sum

$$\sum_{k=0}^{T-1} (-\boldsymbol{w}_{k+1} + \boldsymbol{w}_k) * \bar{\boldsymbol{w}} \leq \sum_{k=0}^{T-1} -\gamma'_H$$

$$\Rightarrow -\boldsymbol{w}_T * \bar{\boldsymbol{w}} \leq -\boldsymbol{w}_0 * \bar{\boldsymbol{w}} - T\gamma'_H . \quad \text{(B.3)}$$

Recall that, for the Minkowski product, we have

$$\cosh(\angle(\boldsymbol{u}, \boldsymbol{u}')) = -\frac{\boldsymbol{u} * \boldsymbol{u}'}{\sqrt{-\boldsymbol{u} * \boldsymbol{u}} \sqrt{-\boldsymbol{u}' * \boldsymbol{u}'}} = \frac{-\boldsymbol{u} * \boldsymbol{u}'}{|\boldsymbol{u}| |\boldsymbol{u}'|} . \quad \text{(B.4)}$$

By utilizing (B.4) with $(\boldsymbol{u}, \boldsymbol{u}') = (\boldsymbol{w}_T, \bar{\boldsymbol{w}})$ and $(\boldsymbol{u}, \boldsymbol{u}') = (\boldsymbol{w}_0, \bar{\boldsymbol{w}})$ in (B.3), we obtain that

$$|\boldsymbol{w}_T| |\bar{\boldsymbol{w}}| \cosh(\angle(\boldsymbol{w}_T, \bar{\boldsymbol{w}})) \leq |\boldsymbol{w}_0| |\bar{\boldsymbol{w}}| \cosh(\angle(\boldsymbol{w}_0, \bar{\boldsymbol{w}})) - T\gamma'_H. \quad \text{(B.5)}$$

Since, we have $|\bar{\boldsymbol{w}}| = |\boldsymbol{w}_0| = 1$, it follows from (B.5) that

$$|\boldsymbol{w}_T| \cosh(\angle(\boldsymbol{w}_T, \bar{\boldsymbol{w}})) \leq \cosh(\angle(\boldsymbol{w}_0, \bar{\boldsymbol{w}})) - T\gamma'_H. \quad \text{(B.6)}$$

Further, using the facts that, due to normalization in Algorithm 1, $|\boldsymbol{w}_T| \geq 1$ and $\cosh(\cdot) \geq 1$, it follows from (B.6) that

$$1 \leq \cosh(\angle(\boldsymbol{w}_0, \bar{\boldsymbol{w}})) - T\gamma'_H \overset{(ii)}{\leq} C - T\gamma'_H, \quad \text{(B.7)}$$

where $(ii)$ follows as $\angle(\boldsymbol{w}_i, \bar{\boldsymbol{w}}) < \pi$, since the orientation is fixed by the requirement that $\mathbb{L}^d \cap \mathcal{H}_{\boldsymbol{w}_i} \neq \emptyset$; as a result, we can find an upper bound $\cosh(\angle(\boldsymbol{w}_0, \bar{\boldsymbol{w}})) < \cosh(\pi) = C$. Now, it follows from (B.7) that

$$T \leq \frac{C-1}{\gamma'_H}, \quad \text{(B.8)}$$

which completes the proof of the convergence guarantee. The margin is given by

$$\text{margin}_{\mathcal{S}}(w) = \inf_{(x,y) \in \mathcal{S}} \text{asinh}\left( \frac{y(\boldsymbol{w} * \boldsymbol{x})}{\sqrt{-\boldsymbol{w} * \boldsymbol{w}}} \right) = \text{asinh}(\gamma'_H) = \text{asinh}\left( \sinh(\gamma_H) \right) = \gamma_H ,$$

which implies that a margin of $\gamma_H$ is achieved in $O\left( \frac{1}{\sinh(\gamma_H)} \right)$ steps. □

# C   Adversarial Learning

## C.1   Loss functions

For training the classifier, we consider the margin losses that have the following form

$$l(\boldsymbol{x}, y; \boldsymbol{w}) = f(y \cdot (\boldsymbol{w} * \boldsymbol{x})), \tag{C.1}$$

where $f \colon \mathbb{R} \to \mathbb{R}_+$ is some convex, non-increasing function. Cho et al. [6] introduce the *hinge loss* in the hyperbolic setting which is defined by the (hyperbolic) hinge function $f(s) = \max\{0, \operatorname{asinh}(1) - \operatorname{asinh}(s)\}$, i.e.,

$$l(\boldsymbol{x}, y; \boldsymbol{w}) = \max\{0, \operatorname{asinh}(1) - \operatorname{asinh}(y(\boldsymbol{w} * \boldsymbol{x}))\} . \tag{C.2}$$

A significant shortcoming of this notion is its non-smoothness and non-convexity. Therefore, we additionally consider a smoothed *least squares loss*:

$$l(\boldsymbol{x}_i, y_i; \boldsymbol{w}) = \begin{cases} \frac{1}{2}\left(\operatorname{asinh}(1) - \operatorname{asinh}(y_i(\boldsymbol{w} * \boldsymbol{x}_i))\right)^2, & y_i(\boldsymbol{w} * \boldsymbol{x}_i) \le 1 \\ 0, & \text{else} \end{cases}, \tag{C.3}$$

We present experimental results for both losses.

The majority of the paper employs a hyperbolic version of the logistic loss to introduce the logistic regression problem in hyperbolic space. First, recall the logistic regression problem in the Euclidean setting. Given an input $\boldsymbol{x}$ and a linear classifier defined by $\boldsymbol{w}$, the prediction of the classifier is defined as

$$p(y|\boldsymbol{x}; \boldsymbol{w}) = 1/\left(1 + \exp(-y\langle \boldsymbol{x}, \boldsymbol{w}\rangle)\right) \tag{C.4}$$

Thus the log-loss takes the following form

$$\begin{aligned} l(\boldsymbol{x}, y; \boldsymbol{w}) = -\log p(y|\boldsymbol{x}; \boldsymbol{w}) &= \log\left(1 + \exp(-y\langle \boldsymbol{x}, \boldsymbol{w}\rangle)\right) \\ &= \log\left(1 + \exp(-y\|\boldsymbol{w}\|\langle \boldsymbol{x}, \bar{\boldsymbol{w}}\rangle)\right) \\ &= \log\left(1 + \exp(-y\operatorname{sgn}(\langle \boldsymbol{x}, \bar{\boldsymbol{w}}\rangle)\|\boldsymbol{w}\|d(\boldsymbol{x}, \partial H_{\bar{\boldsymbol{w}}}))\right) \end{aligned} \tag{C.5}$$

where $\bar{\boldsymbol{w}} = \boldsymbol{w}/\|\boldsymbol{w}\|$ and $d(\boldsymbol{x}, \partial H_{\bar{\boldsymbol{w}}})$ is the distance of $\boldsymbol{x}$ from the decision boundary $\partial H_{\bar{\boldsymbol{w}}} := \{\boldsymbol{z} \in \mathbb{R}^{d+1} : \langle \boldsymbol{z}, \bar{\boldsymbol{w}}\rangle = 0\}$. Note that $y\operatorname{sgn}(\langle \boldsymbol{x}, \bar{\boldsymbol{w}}\rangle)d(\boldsymbol{x}, \partial H_{\bar{\boldsymbol{w}}}))$ denotes the Euclidean margin of the $(\boldsymbol{x}, y)$ with respect to the decision boundary defined by $\bar{\boldsymbol{w}}$.

We can define a hyperbolic version of the logistic regression problem, where we replace the Euclidean margin with the hyperbolic margin with respect to the linear classifier $\boldsymbol{w}$. Recall that the hyperbolic margin has the following form (cf.. (2.2)):

$$y\operatorname{sgn}(\boldsymbol{x} * \boldsymbol{w})d(\boldsymbol{x}, \partial \mathcal{H}_{\boldsymbol{w}}) = y\operatorname{sgn}(\boldsymbol{x} * \boldsymbol{w})\left|\operatorname{asinh}\left(\frac{\boldsymbol{w} * \boldsymbol{x}}{\sqrt{-\boldsymbol{w} * \boldsymbol{w}}}\right)\right| = \operatorname{asinh}\left(\frac{y(\boldsymbol{w} * \boldsymbol{x})}{\sqrt{-\boldsymbol{w} * \boldsymbol{w}}}\right) \tag{C.6}$$

Therefore, by combining (C.5) and (C.6), the hyperbolic logistic regression problem with a linear classifier corresponds to minimizing the following loss:

$$l(\boldsymbol{x}, y; \boldsymbol{w}) = \ln\left(1 + \exp\left(-\operatorname{asinh}\left(\frac{y(\boldsymbol{w} * \boldsymbol{x})}{\sqrt{-\boldsymbol{w} * \boldsymbol{w}}}\right)\right)\right) . \tag{C.7}$$

Note that the hyperbolic logistic loss and the Euclidean logistic loss differ in the scaling factor $\|\boldsymbol{w}\|$. In order to ensure that the hyperbolic logistic loss satisfies Assumption 1, we introduce additional explicit scaling to obtain the following form of the loss.

$$l(\boldsymbol{x}, y; \boldsymbol{w}) = \ln\left(1 + \exp\left(-\operatorname{asinh}\left(\frac{y(\boldsymbol{w} * \boldsymbol{x})}{2R}\right)\right)\right) . \tag{C.8}$$

The following result verifies that the loss in (C.8) indeed satisfies Assumption 1.

**Lemma C.1.** *For valid inputs $(\boldsymbol{x}, y; \boldsymbol{w})$, the hyperbolic logistic loss in* (C.8) *fulfills Assumption 1.*

*Proof.* The robust loss (Eq. 4.1) is evaluated over inputs $(\boldsymbol{x}, y; \boldsymbol{w})$ only if $y(\boldsymbol{w} * \boldsymbol{x}) < 0$. A simple calculation shows, that Assumption 1.3 holds iff $\frac{|\boldsymbol{w} * \boldsymbol{x}|}{R_\alpha} \le 1$, where $R_\alpha$ is as given in Assumption 1.2.

As a results, we want to show $|\boldsymbol{w} * \boldsymbol{x}| \leq R_\alpha$ for all allowable inputs $(\boldsymbol{x}, y; \boldsymbol{w})$. Recall that

$$\boldsymbol{w} * \boldsymbol{x} = w_0 x_0 - \sum_{i=1}^{d} w_i x_i$$

$$\boldsymbol{w} \cdot \boldsymbol{x} = w_0 x_0 + \sum_{i=1}^{d} w_i x_i \,.$$

We consider the following cases:

1. $w_0 x_0 > 0$ *and* $\sum_{i=1}^{d} w_i x_i < 0$: $|\boldsymbol{w} * \boldsymbol{x}| \geq |\boldsymbol{w} \cdot \boldsymbol{x}|$;

2. $w_0 x_0 > 0$ *and* $\sum_{i=1}^{d} w_i x_i > 0$: $|\boldsymbol{w} \cdot \boldsymbol{x}| \geq |\boldsymbol{w} * \boldsymbol{x}|$;

3. $w_0 x_0 < 0$ *and* $\sum_{i=1}^{d} w_i x_i > 0$: $|\boldsymbol{w} * \boldsymbol{x}| \geq |\boldsymbol{w} \cdot \boldsymbol{x}|$;

4. $w_0 x_0 < 0$ *and* $\sum_{i=1}^{d} w_i x_i < 0$: $|\boldsymbol{w} \cdot \boldsymbol{x}| \geq |\boldsymbol{w} * \boldsymbol{x}|$.

In case (2) and (4) we have

$$|\boldsymbol{w} * \boldsymbol{x}| \leq |\boldsymbol{w} \cdot \boldsymbol{x}| \overset{(i)}{\leq} \|\boldsymbol{w}\| \, \|\boldsymbol{x}\| \overset{(ii)}{\leq} R_x R_w = R_\alpha \,,$$

where $(i)$ follows from the Cauchy-Schwartz inequality and $(ii)$ follows from Assumption 1.2. In case (1) and (3), we have

$$|\boldsymbol{w} * \boldsymbol{x}| = |\boldsymbol{w} \cdot \hat{\boldsymbol{x}}| \overset{(i)}{\leq} \|\boldsymbol{w}\| \, \|\hat{\boldsymbol{x}}\| \overset{(ii)}{\leq} R_x R_w = R_\alpha \,, \tag{C.9}$$

where $\hat{\boldsymbol{x}} = (x_0, -x_1, \ldots, -x_n)$ and $(i)$ and $(ii)$ again follow from the Cauchy-Schwartz inequality, respectively. This completes the proof. $\qquad\square$

**Remark C.2.** A conceptually similar logistic loss is introduced in [17] for multinomial manifold. Max-margin learning with the above hyperbolic hinge loss was studied in [6].

## C.2 Generating adversarial examples (Certification problem)

Recall that to train a classifier with large margin, we enrich the training set with adversarial examples (cf. Algorithm 2). For a classifier $\boldsymbol{w}$, an adversarial example $\tilde{\boldsymbol{x}}$ for a given $(\boldsymbol{x}, y)$ is generated by perturbing $\boldsymbol{x}$ in the hyperbolic space up to the maximum allowed perturbation budget $\alpha$ such that

$$\tilde{\boldsymbol{x}} \leftarrow \operatorname*{argmax}_{\substack{\boldsymbol{z} \in \mathbb{L}^d \\ d_{\mathbb{L}}(\boldsymbol{x}, \boldsymbol{z}) \leq \alpha}} l(\boldsymbol{z}, y; \boldsymbol{w}) \,.$$

For the underlying loss function (cf. Section C.1), due to the monotonicity of $\mathrm{asinh}$, the above problem can be equivalently expressed as

$$\tilde{\boldsymbol{x}} \leftarrow \operatorname*{argmin}_{\substack{\boldsymbol{z} \in \mathbb{L}^d \\ d_{\mathbb{L}}(\boldsymbol{x}, \boldsymbol{z}) \leq \alpha}} y \cdot (\boldsymbol{w} * \boldsymbol{z}) = \operatorname*{argmax}_{\substack{\boldsymbol{z} \in \mathbb{L}^d \\ d_{\mathbb{L}}(\boldsymbol{x}, \boldsymbol{z}) \leq \alpha}} -\boldsymbol{w}' * \boldsymbol{z}$$

$$= \operatorname*{argmax}_{\substack{\boldsymbol{z} \in \mathbb{L}^d \\ d_{\mathbb{L}}(\boldsymbol{x}, \boldsymbol{z}) \leq \alpha}} -w_0' z_0 + \sum_{i} w_i' z_i \tag{C.10}$$

where $\boldsymbol{w}' = -y\boldsymbol{w}$. Since $\boldsymbol{w}', \boldsymbol{z} \in \mathbb{R}^{d+1}$, we can rewrite (C.10) as a constraint optimization task in the ambient Euclidean space:

$$\max_{\boldsymbol{z} \in \mathbb{R}^{d+1}} -w_0 z_0 + \sum_{i} w_i z_i \tag{C.11}$$

$$\text{s.t.} \quad d_{\mathbb{L}}(\boldsymbol{x}, \boldsymbol{z}) \leq \alpha$$

$$z_0^2 - \sum_{i=1}^{d} z_i^2 = 1 \,.$$

Assuming that we guess $z_0$ based on $x_0$, the constraint $z_0^2 - \sum_{i=1}^d z_i^2 = 1$ confines the solution space onto a $d$-dimensional sphere of radius $r = \sqrt{z_0^2 - 1}$, which also implies that $z_0 \geq 1$. On the other hand the constraint $d_\mathbb{L}(\boldsymbol{x}, \boldsymbol{z}) \leq \alpha$ is equivalent to

$$d_\mathbb{L}(\boldsymbol{x}, \boldsymbol{z}) = \operatorname{acosh}(\boldsymbol{x} * \boldsymbol{z}) = \operatorname{acosh}(x_0 z_0 - \sum_{i=1}^d x_i z_i) < \alpha \quad \text{or} \quad \sum_i -x_i z_i \leq \cosh(\alpha) - x_0 z_0 \; .$$

Thus, the problem in (C.11) reduces to the following linear program with a spherical constraint.

$$\text{(CERT)} \quad \max_{\boldsymbol{z}_{\backslash 0} \in \mathbb{R}^d} \; - w_0 z_0 + \sum_i w_i z_i \tag{C.12}$$

$$\text{s.t.} \quad \sum_{i=1}^d -x_i z_i \leq \cosh(\alpha) - x_0 z_0$$

$$\|\boldsymbol{z}_{\backslash 0}\|^2 = z_0^2 - 1 \; ,$$

where $\boldsymbol{z}_{\backslash 0} = (z_1, \ldots, z_d)$. We now present a proof of Theorem 4.1 which characterizes a solution of the program in (C.12). For the sake of readability, we first restate the result from the main text:

**Theorem C.3** (Theorem 4.1). *Given the input example $(\boldsymbol{x}, y)$, let $\boldsymbol{x}_{\backslash 0} = (x_1, \ldots, x_d)$. We can efficiently compute a solution to* (CERT) *or decide that no solution exists. If a solution exists, then based on a guess of $z_0$ a maximizing adversarial example has the form $\tilde{\boldsymbol{x}} = \left( z_0, \sqrt{z_0^2 - 1} \left( b\check{\boldsymbol{x}} + \sqrt{1 - b^2}\check{\boldsymbol{x}}^\perp \right) \right)$. Here, $b = \frac{(\cosh(\alpha) - x_0 z_0)}{(\|\boldsymbol{x}_{\backslash 0}\| \sqrt{z_0^2 - 1})}$ depends on the adversarial budget $\alpha$, and $\check{\boldsymbol{x}}_{\backslash 0}^\perp$ is a unit vector orthogonal to $\check{\boldsymbol{x}} = -\boldsymbol{x}_{\backslash 0}/\|\boldsymbol{x}_{\backslash 0}\|$ along $\boldsymbol{w}$.*

*Proof.* First, note that (CERT) can be rewritten as

$$\text{(CĚRT)} \quad \max \; \langle \check{\boldsymbol{w}}, \check{\boldsymbol{z}} \rangle$$

$$\text{s.t.} \; \langle \check{\boldsymbol{x}}, \check{\boldsymbol{z}} \rangle \leq b$$

$$\|\check{\boldsymbol{z}}\| = 1 \; ,$$

where $\check{\boldsymbol{w}} = \boldsymbol{w}_{\backslash 0}/\|\boldsymbol{w}_{\backslash 0}\|$, $\check{\boldsymbol{x}} = -\boldsymbol{x}_{\backslash 0}/\|\boldsymbol{x}_{\backslash 0}\|$, and $b = (\cosh(\alpha) - x_0 z_0)/(\|\boldsymbol{x}_{\backslash 0}\|\|\boldsymbol{z}_{\backslash 0}\|)$. We further set $\check{\boldsymbol{z}} = \boldsymbol{z}_{\backslash 0}/\|\boldsymbol{z}_{\backslash 0}\|$ so that the norm constraint confines the solution to the unit sphere to simplify the derivation. We can later rescale the solution to have the norm $\sqrt{z_0^2 - 1}$.

The solution of CĚRT lies on the cone $\langle \check{\boldsymbol{x}}, \check{\boldsymbol{z}} \rangle = b$. We decompose $\check{\boldsymbol{w}}$ along $\check{\boldsymbol{x}}$ and its orthogonal complement $\check{\boldsymbol{x}}^\perp$, i.e.

$$\check{\boldsymbol{w}} = \xi \check{\boldsymbol{x}} + \zeta \check{\boldsymbol{x}}^\perp \; .$$

with $\zeta \geq 0$ and $\|\check{\boldsymbol{x}}^\perp\| = 1$. Without loss of generality, such a decomposition always exists. Note that

$$\langle \check{\boldsymbol{w}}, \check{\boldsymbol{z}}^* \rangle = \xi \langle \check{\boldsymbol{x}}, \check{\boldsymbol{z}}^* \rangle + \zeta \langle \check{\boldsymbol{x}}^\perp, \check{\boldsymbol{z}}^* \rangle = \xi b + \zeta \langle \check{\boldsymbol{x}}^\perp, \check{\boldsymbol{z}}^* \rangle \; ,$$

where the second equality follows from $\langle \check{\boldsymbol{x}}, \check{\boldsymbol{z}}^* \rangle = b$. This implies that for the objective $\langle \check{\boldsymbol{w}}, \check{\boldsymbol{z}} \rangle$ to be maximized, $\check{\boldsymbol{z}}^*$ has to have all of its remaining mass along $\check{\boldsymbol{x}}^\perp$, i.e.,

$$\check{\boldsymbol{z}}^* = b\check{\boldsymbol{x}} + \sqrt{1 - b^2}\check{\boldsymbol{x}}^\perp.$$

After rescaling to satisfy the original norm constraint in CERT, the maximizing adversarial example (for a given $z_0$) is given as

$$\tilde{\boldsymbol{x}} = \left( z_0, \sqrt{z_0^2 - 1} \cdot \check{\boldsymbol{z}}^* \right) = \left( z_0, \sqrt{z_0^2 - 1} \left( b\check{\boldsymbol{x}} + \sqrt{1 - b^2}\check{\boldsymbol{x}}^\perp \right) \right).$$

$\square$

## C.3 Adversarial Perceptron

For the convergence analysis of the gradient-based update, we first need to analyze the convergence of the adversarial perceptron. We first state the following lemma that relates the adversarial margin to the max-margin classifier.

**Lemma C.4.** *Let $\bar{w}$ be the max-margin classifier of $\mathcal{S}$ with margin $\gamma_H$. At each iteration of Algorithm 2, $\bar{w}$ linearly separates $\mathcal{S} \cup \mathcal{S}'$ with margin at least $\frac{\gamma_H}{\cosh(\alpha)}$.*

**Remark C.5.** Note that this "adversarial Perceptron" corresponds to a gradient update of the form $w_{t+1} \leftarrow w_t + y\tilde{x}$, which resembles the adversarial SGD.

*Proof.* The proof reduces the problem to Euclidean geometry in the Poincare half plane. We defer the proof until Section E, since the respective geometric tools are introduced only in Section D.2. $\square$

With this result, we can show the following bound on the sample complexity of the adversarial perceptron:

**Theorem C.6.** *Assume that there is some $\bar{w} \in \mathbb{R}^{d+1}$ with $\sqrt{-\bar{w} * \bar{w}} = 1$ and $w_0 * \bar{w} \leq 0$, and some $\gamma_H > 0$, such that $y_j(\bar{w} * x_j) \geq \sinh(\gamma_H)$ for $j = 1, \ldots, |\mathcal{S}|$. Then, adversarial perceptron (with adversarial budget $\alpha$) converges after $O\left(\frac{\cosh(\alpha)}{\sinh(\gamma_H)}\right)$ steps, at which it has margin of at least $\frac{\gamma_H}{\cosh(\alpha)}$.*

*Proof.* Without loss of generality, we initialize the classifier as $w_0 = (0, 1, 0, \ldots, 0)$. Furthermore, assume that the $t$th error is made at the $j$th sample. For the ease of exposition, we assume that the normalization step is not performed at this update. (The case with normalization after the update can be handled as in the Proof of Theorem B.1.) Thus,

$$w_{t+1} \leftarrow w_t + y_j\tilde{x}_i \,,$$

which implies that

$$(w_{t+1} - w_t) * \bar{w} = (y_j\tilde{x}_j) * \bar{w} = y_j(\tilde{x}_j * \bar{w}) \geq \frac{\gamma_H'}{\cosh(\alpha)} \,,$$

where $\gamma_H' = \sinh(\gamma_H)$ and the last inequality follows from Lemma C.4. By summing and telescoping, we obtain that

$$\sum_{k=0}^{t}(w_{k+1} - w_k) * \bar{w} \geq \sum_{k=0}^{t}\frac{\gamma_H'}{\cosh(\alpha)}$$

$$\Rightarrow \quad (w_{t+1} - w_0) * \bar{w} \geq \frac{t\gamma_H'}{\cosh(\alpha)} \,.$$

Now, by multiplying both sides by $-1$ and rewriting the Minkowski product gives us that

$$-w_{t+1} * \bar{w} \leq -w_0 * \bar{w} - \frac{t\gamma_H'}{\cosh(\alpha)}$$

$$\leq \underbrace{|w_0|}_{=1} \underbrace{|\bar{w}|}_{=1} \underbrace{\cosh(\angle(w_0, \bar{w}))}_{\leq \cosh(\pi) =: C} - \frac{t\gamma_H'}{\cosh(\alpha)}$$

$$\leq C - \frac{t\gamma_H'}{\cosh(\alpha)} \,. \tag{C.13}$$

Now, note that

$$1 \leq \cosh(\angle(w_{t+1}, \bar{w})) \leq \frac{-w_{t+1} * \bar{w}}{\underbrace{|w_{t+1}|}_{\geq 1} \underbrace{|\bar{w}|}_{=1}} \leq -w_{t+1} * \bar{w} \stackrel{(i)}{\leq} C - \frac{t\gamma_H'}{\cosh(\alpha)} \,,$$

where $(i)$ utilizes (C.13). Now, solving for $t$ gives us that

$$t \leq (C - 1) \cdot \frac{\cosh(\alpha)}{\gamma_H'}.$$

Further, it follows from (2.2) that an adversarial hyperbolic margin of $\frac{\gamma_H}{\cosh(\alpha)}$ is then achieved after $O\left(\frac{\cosh(\alpha)}{\sinh(\gamma_H)}\right)$ steps. $\square$

## C.4 Gradient-based update

Recall that, our objective in Algorithm 2 consists of an inner optimization (that computes the adversarial example) and an outer optimization (that updates the classifier). In particular, we consider

$$\min_{\boldsymbol{w} \in \mathbb{R}^{d+1}} L_{\mathrm{rob}}(\boldsymbol{w}; \mathcal{S}) := \frac{1}{|\mathcal{S}|} \sum_{(x,y) \in \mathcal{S}} l_{\mathrm{rob}}(\boldsymbol{x}, y; \boldsymbol{w}) \;,$$

where the robust loss is given by

$$l_{\mathrm{rob}}(\boldsymbol{x}, y; \boldsymbol{w}) := \max_{\boldsymbol{z} \in \mathbb{L}^d, d_{\mathbb{L}}(\boldsymbol{x}, \boldsymbol{z}) \leq \alpha} l(\boldsymbol{x}, y; \boldsymbol{w}) = l(\tilde{\boldsymbol{x}}, y; \boldsymbol{w}) \;,$$

where $\tilde{\boldsymbol{x}} \in \mathrm{argmax}_{\boldsymbol{z} \in \mathbb{L}^d, d_{\mathbb{L}}(\boldsymbol{x}, \boldsymbol{z}) \leq \alpha} l(\boldsymbol{x}, y; \boldsymbol{w})$.

Recall that, to compute the update, we need to compute gradients of the outer minimization problem, i.e., $\nabla_w \, l_{\mathrm{rob}}$ over $\mathcal{S}$. However, the function $l_{\mathrm{rob}}$ is itself a maximization problem (referred to as the inner maximization problem above). Therefore, we compute the gradient at the maximizer of the inner problem. Danskin's theorem ensures that this gives a valid decent direction. For the sake of completeness, we recall the Danskin's theorem here.

**Theorem C.7** ( Danskin [9], Bertsekas [1]). *Suppose $X$ is a non-empty compact topological space and $g : \mathbb{R}^d \times X \to \mathbb{R}$ is a continuous function such that $g(\cdot, \delta)$ is differentiable for every $\delta \in X$. Let $\delta_{\boldsymbol{w}}^* = \mathrm{argmax}_{\delta \in X} \, g(\boldsymbol{w}, \delta)$. Then, the function $\psi(\boldsymbol{w}) = \max_{\delta \in X} \, g(\boldsymbol{w}, \delta)$ is subdifferentiable and the subdifferential is given by*

$$\partial \psi(\boldsymbol{w}) = \mathrm{conv} \left( \{ \nabla_{\boldsymbol{w}} \, g(\boldsymbol{w}, \delta) | \; \delta \in \delta_{\boldsymbol{w}}^* \} \right) \;.$$

This approach has been previously used in Madry et al. [20] and Charles et al. [5]. Note that when we find an adversarial example in Algorithm 2, we can write it in a closed form (cf. Theorem C.3). In particular,

$$l_{\mathrm{rob}}(\boldsymbol{x}, y; \boldsymbol{w}) = \max_{d_{\mathbb{L}}(\boldsymbol{x}, \boldsymbol{z}) \leq \alpha} l(\boldsymbol{z}, y; \boldsymbol{w}) = l(\tilde{\boldsymbol{x}}, y; \boldsymbol{w}) \quad \text{with } \tilde{\boldsymbol{x}} = \left( \tilde{x}_0, \sqrt{\tilde{x}_0^2 - 1} \left( b \, \check{\boldsymbol{x}} + \sqrt{1 - b^2} \tilde{\boldsymbol{x}}^\perp \right) \right) \;.$$

Note that

$$\nabla_{\boldsymbol{w}} \, l(\tilde{\boldsymbol{x}}, y; \boldsymbol{w}) = f'(y(\boldsymbol{w} * \tilde{\boldsymbol{x}})) \cdot \nabla_{\boldsymbol{w}} \, y(\boldsymbol{w} * \tilde{\boldsymbol{x}}) \; = f'(y(\boldsymbol{w} * \tilde{\boldsymbol{x}})) \cdot y\widehat{(\tilde{\boldsymbol{x}})}^T \;,$$

where we have used the fact that $\nabla_{\boldsymbol{w}} \, y(\boldsymbol{w} * \tilde{\boldsymbol{x}}) = y\widehat{(\tilde{\boldsymbol{x}})}^T = y(\tilde{x}_0, -\tilde{x}_1, \ldots, -\tilde{x}_n)^T$. From Danskin's theorem, we have $\nabla_{\boldsymbol{w}} l(\tilde{\boldsymbol{x}}, y; \boldsymbol{w}) \in \partial \, l_{\mathrm{rob}}(\boldsymbol{x}, y; \boldsymbol{w})$. This enables us to compute the decent direction and perform the update step with

$$\nabla \, L(\boldsymbol{w}; \mathcal{S}') = \frac{1}{|\mathcal{S}'|} \sum_{(\tilde{\boldsymbol{x}}, y) \in \mathcal{S}'} \nabla \, l(\tilde{\boldsymbol{x}}, y; \boldsymbol{w}) \; \in \partial L_{\mathrm{rob}}(\boldsymbol{w}; \mathcal{S}) \;,$$

Furthermore, we have

$$\nabla_{\boldsymbol{w}}^2 \, l(\tilde{\boldsymbol{x}}, y; \boldsymbol{w}) = f''(y(\boldsymbol{w} * \tilde{\boldsymbol{x}})) \tilde{\boldsymbol{x}} \tilde{\boldsymbol{x}}^T \in \partial^2 l_{\mathrm{rob}}(\boldsymbol{x}, y; \boldsymbol{w}) \;, \tag{C.14}$$

which enable the computation of the Hessian of $L(\boldsymbol{w}; \mathcal{S}')$.

The convergence results in this section build on hyperbolic analogues of comparable Euclidean results in [30, 14].

We first show a bound on the Hessian of the loss:

**Lemma C.8.**

$$\nabla^2 L(\boldsymbol{w}_t; \mathcal{S}'_t) \preceq \beta \sigma_{\max}^2 \cdot I \;,$$

*where $\sigma_{\max}$ is an upper bound on the maximum singular value of the data matrix $\frac{1}{|\mathcal{S}'_t|} \sum_{(\tilde{\boldsymbol{x}}, y) \in \mathcal{S}'_t} \tilde{\boldsymbol{x}} \tilde{\boldsymbol{x}}^T$.*

*Proof.*

$$\nabla^2 L(\boldsymbol{w}_t; \mathcal{S}'_t) = \frac{1}{|\mathcal{S}'_t|} \sum_{(\tilde{\boldsymbol{x}}, y) \in \mathcal{S}'_t} \nabla^2 l(\tilde{\boldsymbol{x}}, y; \boldsymbol{w}_t) \overset{(i)}{=} \frac{1}{|\mathcal{S}'_t|} \sum_{(\tilde{\boldsymbol{x}}, y) \in \mathcal{S}'_t} f''(y(\tilde{\boldsymbol{x}} * \boldsymbol{w}_t)) \tilde{\boldsymbol{x}} \tilde{\boldsymbol{x}}^T$$

$$\overset{(ii)}{\preceq} \beta \cdot \frac{1}{|\mathcal{S}'_t|} \sum_{(\tilde{\boldsymbol{x}}, y) \in \mathcal{S}'_t} \tilde{\boldsymbol{x}} \tilde{\boldsymbol{x}}^T \preceq \beta \sigma_{\max}^2 \cdot I \;,$$

where $(i)$ and $(ii)$ follow from (C.14) and the assumption that $f$ is $\beta$-smooth. $\qquad \square$

With the help of Lemma C.8, we can show the following result (a restatement of Theorem 4.4), which establishes that the gradient updates are guaranteed to converge to a large-margin classifier:

**Theorem C.9** (Theorem 4.3). *Let $\{w_t\}$ be the GD iterates*

$$w_{t+1} \leftarrow w_t - \frac{\eta}{|\mathcal{S}_t'|} \sum_{(\tilde{x},y) \in \mathcal{S}_t'} \nabla l(\tilde{x}, y; w)$$

$$w_{t+1} \leftarrow \frac{w_{t+1}}{\sqrt{-w_{t+1} * w_{t+1}}}$$

*with constant step size $\eta < \frac{2}{\beta \sigma_{\max}^2}$ and an initialization $w_0$ with $w_0 * w_0 < 0$. Then, we have $\lim_{t \to \infty} L(w_t; \mathcal{S} \cup \mathcal{S}_t') = 0$.*

*Proof.* By Assumption 1.1 we can find a $\bar{w}$ that linearly separates $\mathcal{S}$. Then, we have

$$\langle \bar{w}, \nabla L(w; \mathcal{S}_t') \rangle = \langle \bar{w}, \frac{1}{|\mathcal{S}_t'|} \sum_{(\tilde{x},y) \in \mathcal{S}_t'} f'(y(\tilde{x} * w_t)) \, y\widehat{\tilde{x}} \rangle$$

$$= \underbrace{\left( \frac{1}{|\mathcal{S}_t'|} \sum_{(x,y) \in \mathcal{S}_t'} f'(y(\tilde{x} * w_t)) \right)}_{<0} \underbrace{y\langle \bar{w}, \widehat{\tilde{x}} \rangle}_{=y(\bar{w}*\tilde{x})<0} ,$$

where the negativity of the first term follows from the assumptions on $f$ (cf. Assumption 1.3) and the upper bound on the second term from the separability assumption. This implies that $\langle \bar{w}, \nabla L(w; \mathcal{S}_t') \rangle \neq 0$ for any finite $w$. Therefore, there are no finite critical points $w$ for which $\nabla L(w; \mathcal{S}_t') = 0$. However, GD is guaranteed to converge to a critical point for smooth objectives with an appropriate step size. Therefore, $\|w_t\| \to \infty$ and $y(w_t * x) > 0 \, \forall \, (x,y) \in \mathcal{S} \cup \mathcal{S}_t'$ and large enough $t$. Then, we have $l(x, y; w_t) \to 0$, for all $(x,y) \in \mathcal{S} \cup \mathcal{S}_t'$. This further implies that $L(w_t; \mathcal{S} \cup \mathcal{S}_t') = \frac{1}{|\mathcal{S} \cup \mathcal{S}_t'|} \sum_{(x,y) \in \mathcal{S} \cup \mathcal{S}_t'} l(x, y; w_t) \to 0$. $\square$

We further show that the enrichment of the training set with adversarial examples is critical for polynomial-time convergence: Without adversarial training, we can construct a simple max-margin problem, that cannot be solved in polynomial time.

**Theorem C.10** (Theorem 3.3). *Consider $\mathcal{S} = \{(e_1, 1), (-e_1, -1)\} \subset \mathbb{R}^{d+1} \times \{+1, -1\}$ and a typical initialization $w_0 = e_2 \in \mathbb{R}^{d+1}$ (with the standard basis vectors $e_1, e_2 \in \mathbb{R}^{d+1}$). Let $\{w_t\}_t$ is a sequence of classifiers generated by the GD updates (with fixed step size $\eta$)*

$$w_{t+1} \leftarrow w_t - \frac{\eta}{|\mathcal{S}|} \sum_{(x,y) \in \mathcal{S}} \nabla l(x, y; w)$$

$$w_{t+1} \leftarrow \frac{w_{t+1}}{\sqrt{-w_{t+1} * w_{t+1}}} .$$

*Then, the number of iterations needed to achieve margin $\gamma_H$ is $\Omega(\exp(\gamma_H))$.*

*Proof.* First, note that the initialization $w_0$ is valid as $w_0 * w_0 = -1 < 0$. The gradient of the loss can be computed as

$$\nabla l(x_i, y_i; w_t) = f'(y_i(x_i * w_t))y_i\widehat{x}_i$$

where

$$f'(s) = -\frac{\exp\left(-\operatorname{asinh}\left(\frac{s}{2R}\right)\right)}{R\sqrt{\frac{s^2}{4R^2} + 1}\left(\exp\left(-\operatorname{asinh}\left(\frac{s}{2R}\right)\right) + 1\right)}$$

is the derivative of the hyperbolic logistic regression loss (cf. (4.5)). Note that due to the structure of $\mathcal{S}$ and $w_0$, the GD update will produce the following iteration sequence

$$a_{t+1} = a_t - f'(a_t)$$

$$w_t = (a_t, \sqrt{a_t^2 + 1}, 0, \ldots, 0) ,$$

where the first coordinate is determined through the GD update and the second through normalization to ensure the validaty of the classifier, i.e., $\boldsymbol{w}_t * \boldsymbol{w}_t < 0$. In order to see this, note that $\boldsymbol{w}_t * \boldsymbol{w}_t = a_t^2 - (\sqrt{a_t^2 + 1})^2 = -1 < 0$. We now want to show that

$$a_t \leq \sinh(\ln(t+1)) .$$

For the induction, note that $a_0 = 0 = \ln(1) = \sinh(\ln(1))$. Assume, that $a_t \leq \sinh(\ln(t+1))$. We want to show

$$a_{t+1} \leq \sinh(\ln(t+2)) .$$

Note, that

$$a_{t+1} = \underbrace{a_t}_{\text{①}} + \underbrace{\frac{\exp\left(-\operatorname{asinh}\left(\frac{a_t}{2R}\right)\right)}{R\sqrt{\frac{a_t^2}{4R^2} + 1}\left(\exp\left(-\operatorname{asinh}\left(\frac{a_t}{2R}\right)\right) + 1\right)}}_{\text{②}} .$$

Since $\exp\left(-\operatorname{asinh}\left(-\frac{a_t}{2R}\right)\right) \leq \exp\left(-\operatorname{asinh}\left(-\frac{a_t}{2R}\right)\right) + 1$ and $R\sqrt{\frac{a_t^2}{4R^2} + 1} \geq 1$, clearly ② is bounded by 1. Inserting this above and replacing ① with the induction assumption, we have

$$a_{t+1} \leq \sinh(\ln(t+1)) + 1 .$$

Note that, by definition, $\sinh(z) = \frac{1}{2}\left(e^x - e^{-x}\right)$. Thus,

$$\sinh(\ln(t+1)) = \frac{1}{2}\left(t + 1 - (-(t+1))\right) = t + 1 ,$$

which further implies that

$$a_{t+1} = \sinh(\ln(t+1)) + 1 \leq t + 2 = \sinh(\ln(t+2)) . \tag{C.15}$$

This finishes the induction proof. Assuming a margin of at least $\gamma_H$, we have

$$\gamma_H \leq \operatorname{margin}_{\mathcal{S}}(\boldsymbol{w}_t) = \operatorname{asinh}\left(\frac{y(\boldsymbol{x} * \boldsymbol{w}_t)}{\sqrt{-\boldsymbol{w}_t * \boldsymbol{w}_t}}\right) \overset{(i)}{=} \operatorname{asinh}(a_{t+1}) \overset{(ii)}{\leq} \operatorname{asinh}(\sinh(\ln(t+2))) \leq \ln(t+2) ,$$

where $(i)$ follows $\boldsymbol{w}_t * \boldsymbol{w}_t = -1$ after normalization and $(ii)$ from the upper bound in (C.15). Now, by solving for $t$, we obtain that $t = \Omega(\exp(\gamma_H))$. □

Next, we quantify the convergence rate of adversarial training with GD updates (cf. (4.4)). We start by presenting some auxiliary results.

**Lemma C.11** (Smoothness bound). *Let $\eta_t =: \eta < \frac{2\sinh^2(\gamma_H)}{\beta\sigma_{\max}^2 \cosh^2(\alpha)R_\alpha^2}$ be the fixed step size and $\boldsymbol{w}_0$ a valid initialization, i.e. $\boldsymbol{w}_0 * \boldsymbol{w}_0 < 0$. Then, for the GD update (with fixed step size $\eta_t =: \eta$)*

$$\boldsymbol{w}_{t+1} \leftarrow \boldsymbol{w}_t - \eta_t \underbrace{\nabla L(\boldsymbol{w}_t; \mathcal{S}_t')}_{\in \partial L_{\mathrm{rob}}(\boldsymbol{w}_t; \mathcal{S}_t)}$$

$$\boldsymbol{w}_{t+1} \leftarrow \frac{\boldsymbol{w}_{t+1}}{\sqrt{-\boldsymbol{w}_{t+1} * \boldsymbol{w}_{t+1}}} .$$

*we have*

1. $L_{\mathrm{rob}}(\boldsymbol{w}_{t+1}; \mathcal{S}) \leq L_{\mathrm{rob}}(\boldsymbol{w}_t; \mathcal{S}) - \eta\left(\frac{\sinh(\gamma_H)^2}{\cosh^2(\alpha)R_\alpha^2} - \frac{\beta\sigma_{\max}^2\eta}{2}\right) \|\underbrace{\nabla L(\boldsymbol{w}_t; \mathcal{S}_t')}_{\in \partial L_{\mathrm{rob}}(\boldsymbol{w}_t; \mathcal{S}_t)}\|^2;$

2. $\sum_{k=0}^{\infty} \|\nabla L(\boldsymbol{w}_k; \mathcal{S}_k')\|^2 < \infty$; *as a result,* $\lim_{t\to\infty} \|\nabla L(\boldsymbol{w}_t); \mathcal{S}_t'\|^2 = 0$.

*Proof.* In Algorithm 2 with gradient update rule, we have

$$\begin{aligned} \boldsymbol{w}_{t+1} &= \boldsymbol{w}_t - \eta\nabla L(\boldsymbol{w}_t; \mathcal{S}_t') \\ &= \boldsymbol{w}_t - \frac{\eta}{|\mathcal{S}_t'|}\sum_{(\tilde{\boldsymbol{x}}, y)\in\mathcal{S}_t'} l(\tilde{\boldsymbol{x}}, y; \boldsymbol{w}_t) \\ &= \boldsymbol{w}_t - \frac{\eta}{|\mathcal{S}_t'|}\sum_{(\tilde{\boldsymbol{x}}, y)\in\mathcal{S}_t'} f'(y(\tilde{\boldsymbol{x}} * \boldsymbol{w}_t))y\widehat{\tilde{\boldsymbol{x}}} . \end{aligned}$$

Now, consider the inner product $\langle \boldsymbol{w}_{t+1}, \bar{\boldsymbol{w}} \rangle$, where $\bar{\boldsymbol{w}}$ is the optimal classifier. With out loss of generality, we assume $\|\bar{\boldsymbol{w}}\| = 1$.

$$\langle \boldsymbol{w}_{t+1}, \bar{\boldsymbol{w}} \rangle = \langle \boldsymbol{w}_t, \bar{\boldsymbol{w}} \rangle - \frac{\eta}{|\mathcal{S}'|} \sum_{(\tilde{\boldsymbol{x}},y)\in\mathcal{S}'} f'(y(\tilde{\boldsymbol{x}}*\boldsymbol{w}_t))y\langle \widehat{\tilde{\boldsymbol{x}}}, \bar{\boldsymbol{w}} \rangle$$

$$\overset{(i)}{=} \langle \boldsymbol{w}_t, \bar{\boldsymbol{w}} \rangle - \frac{\eta}{|\mathcal{S}'_t|} \sum_{(\tilde{\boldsymbol{x}},y)\in\mathcal{S}'_t} f'(y(\tilde{\boldsymbol{x}}*\boldsymbol{w}_t))y(\tilde{\boldsymbol{x}}*\bar{\boldsymbol{w}})$$

$$\overset{(ii)}{\geq} \langle \boldsymbol{w}_t, \bar{\boldsymbol{w}} \rangle - \frac{\eta\gamma'_H}{|\mathcal{S}'_t|\cosh(\alpha)} \sum_{(\tilde{\boldsymbol{x}},y)\in\mathcal{S}'_t} f'(y(\tilde{\boldsymbol{x}}*\boldsymbol{w}_t)) ,$$

where $(i)$ and $(ii)$ follow from $\langle \widehat{\tilde{\boldsymbol{x}}}, \bar{\boldsymbol{w}} \rangle = \tilde{\boldsymbol{x}}*\bar{\boldsymbol{w}}$ and $y(\tilde{\boldsymbol{x}}*\bar{\boldsymbol{w}}) \geq \frac{\gamma'_H}{\cosh(\alpha)}$ (cf. Lemma C.4), respectively. We use the shorthand $\gamma'_H = \sinh(\gamma_H)$. With the linearity of the inner product, we get

$$\langle \boldsymbol{w}_{t+1} - \boldsymbol{w}_t, \bar{\boldsymbol{w}} \rangle \geq - \frac{\eta\gamma'_H}{|\mathcal{S}'_t|\cosh(\alpha)} \sum_{(\tilde{\boldsymbol{x}},y)\in\mathcal{S}'_t} f'(y(\tilde{\boldsymbol{x}}*\boldsymbol{w}_t)) .$$

Since $f'$ is negative (cf. Assumption 1.3), we can replace $-f'(y(\tilde{\boldsymbol{x}}*\boldsymbol{w}_t))$ with $|f'(y(\tilde{\boldsymbol{x}}*\boldsymbol{w}_t))|$ to get

$$\langle \boldsymbol{w}_t - \boldsymbol{w}_{t+1}, \bar{\boldsymbol{w}} \rangle \geq \frac{\eta\gamma'_H}{|\mathcal{S}'_t|\cosh(\alpha)} \sum_{(\tilde{\boldsymbol{x}},y)\in\mathcal{S}'_t} |f'(y(\tilde{\boldsymbol{x}}*\boldsymbol{w}_t))|$$

$$\overset{(i)}{=} \frac{\eta\gamma'_H}{\cosh(\alpha)R_\alpha}\|\nabla L(\boldsymbol{w}_t;\mathcal{S}'_t)\| , \tag{C.16}$$

where $(i)$ holds as follows: Recall, that $\|\nabla l(\tilde{\boldsymbol{x}},y;\boldsymbol{w}_t)\| \leq |f'(y(\tilde{\boldsymbol{x}}*\boldsymbol{w}_t))|\|\widehat{\tilde{\boldsymbol{x}}}\|$. Thus,

$$\|\nabla L(\boldsymbol{w}_t;\mathcal{S}'_t)\| = \|\frac{1}{|\mathcal{S}'_t|} \sum_{(\tilde{\boldsymbol{x}},y)\in\mathcal{S}'_t} l(\tilde{\boldsymbol{x}},y;\boldsymbol{w}_t)\| \leq \frac{1}{|\mathcal{S}'_t|} \sum_{(\tilde{\boldsymbol{x}},y)\in\mathcal{S}'_t} \|l(\tilde{\boldsymbol{x}},y;\boldsymbol{w}_t)\|$$

$$\leq \frac{1}{|\mathcal{S}'_t|} \sum_{(\tilde{\boldsymbol{x}},y)\in\mathcal{S}'_t} |f'(y(\tilde{\boldsymbol{x}}*\boldsymbol{w}_t))|\|\widehat{\tilde{\boldsymbol{x}}}\| \leq \frac{R_\alpha}{|\mathcal{S}'_t|} \sum_{(\tilde{\boldsymbol{x}},y)\in\mathcal{S}'_t} |f'(y(\tilde{\boldsymbol{x}}*\boldsymbol{w}_t))| .$$

This implies that

$$\frac{1}{|\mathcal{S}'_t|} \sum_{(\tilde{\boldsymbol{x}},y)\in\mathcal{S}'_t} |f'(y(\tilde{\boldsymbol{x}}*\boldsymbol{w}_t))| = \frac{1}{R_\alpha}\|\nabla L(\boldsymbol{w}_t;\mathcal{S}'_t)\| .$$

Applying Cauchy-Schwarz to the left hand side of (C.16) gives us that

$$\|\boldsymbol{w}_t - \boldsymbol{w}_{t+1}\| \, \|\bar{\boldsymbol{w}}\| \geq \langle \boldsymbol{w}_t - \boldsymbol{w}_{t+1}, \bar{\boldsymbol{w}} \rangle \geq \frac{\eta\gamma'_H}{\cosh(\alpha)R_\alpha}\|\nabla L(\boldsymbol{w}_t;\mathcal{S}'_t)\| . \tag{C.17}$$

Now, using the fact that $\|\bar{\boldsymbol{w}}\| = 1$ in (C.17), we get

$$\|\boldsymbol{w}_{t+1} - \boldsymbol{w}_t\| \geq \frac{\eta\gamma'_H}{\cosh(\alpha)R_\alpha}\|\nabla L(\boldsymbol{w}_t;\mathcal{S}'_t)\| . \tag{C.18}$$

Now, consider the following Taylor approximation:

$$L_{\mathrm{rob}}(\boldsymbol{w}_{t+1};\mathcal{S}) = L_{\mathrm{rob}}(\boldsymbol{w}_t;\mathcal{S}) + \langle \underbrace{\nabla L(\boldsymbol{w}_t;\mathcal{S}'_t)}_{\in \partial L_{\mathrm{rob}}(\boldsymbol{w}_t;\mathcal{S})}, \boldsymbol{w}_{t+1} - \boldsymbol{w}_t \rangle +$$

$$(\boldsymbol{w}_{t+1} - \boldsymbol{w}_t)^T \underbrace{\nabla^2 L(\boldsymbol{v};\mathcal{S}'_t)}_{\in \partial^2 L_{\mathrm{rob}}(\boldsymbol{v};\mathcal{S})} (\boldsymbol{w}_{t+1} - \boldsymbol{w}_t)/2, \tag{C.19}$$

where $\boldsymbol{v} \in \mathrm{conv}(\boldsymbol{w}_{t+1}, \boldsymbol{w}_t)$. By utilizing Lemma C.8 in (C.19), we get that

$$L_{\mathrm{rob}}(\boldsymbol{w}_{t+1};\mathcal{S}) \leq L_{\mathrm{rob}}(\boldsymbol{w}_t;\mathcal{S}) + \langle \nabla L(\boldsymbol{w}_t;\mathcal{S}'_t), \boldsymbol{w}_{t+1} - \boldsymbol{w}_t \rangle + \frac{\beta\sigma^2_{\max}}{2}\|\boldsymbol{w}_{t+1} - \boldsymbol{w}_t\|^2 . \tag{C.20}$$

Recall the update rule

$$\boldsymbol{w}_{t+1} = \boldsymbol{w}_t - \eta \nabla L(\boldsymbol{w}_t; \mathcal{S}_t') \tag{C.21}$$

$$\Rightarrow \boldsymbol{w}_{t+1} - \boldsymbol{w}_t = -\eta \nabla L(\boldsymbol{w}_t; \mathcal{S}_t') . \tag{C.22}$$

Inserting this in (C.20), we get

$$L_{\text{rob}}(\boldsymbol{w}_{t+1}; \mathcal{S}) = L_{\text{rob}}(\boldsymbol{w}_t; \mathcal{S}) + \langle -\eta^{-1}(\boldsymbol{w}_{t+1} - \boldsymbol{w}_t), \boldsymbol{w}_{t+1} - \boldsymbol{w}_t \rangle + \frac{\beta \sigma_{\max}^2}{2} \|\boldsymbol{w}_{t+1} - \boldsymbol{w}_t\|^2$$

$$= L_{\text{rob}}(\boldsymbol{w}_t; \mathcal{S}) - \eta^{-1} \|\boldsymbol{w}_{t+1} - \boldsymbol{w}_t\|^2 + \frac{\beta \sigma_{\max}^2}{2} \|\boldsymbol{w}_{t+1} - \boldsymbol{w}_t\|^2 . \tag{C.23}$$

By combining (C.18) and (C.23), we obtain that

$$L_{\text{rob}}(\boldsymbol{w}_{t+1}; \mathcal{S}) \leq L_{\text{rob}}(\boldsymbol{w}_t; \mathcal{S}) - \frac{\eta \gamma_H'^2}{\cosh^2(\alpha) R_\alpha^2} \|\nabla L(\boldsymbol{w}_t; \mathcal{S}_t')\|^2 + \frac{\beta \sigma_{\max}^2}{2} \|\boldsymbol{w}_{t+1} - \boldsymbol{w}_t\|^2 . \tag{C.24}$$

Again, utilizing (C.21), it follows from (C.24) that

$$L_{\text{rob}}(\boldsymbol{w}_{t+1}; \mathcal{S}) \leq L_{\text{rob}}(\boldsymbol{w}_t; \mathcal{S}) - \frac{\eta \gamma_H'^2}{\cosh^2(\alpha) R_\alpha^2} \|\nabla L(\boldsymbol{w}_t; \mathcal{S}_t')\|^2 + \frac{\beta \sigma_{\max}^2 \eta^2}{2} \|\nabla L(\boldsymbol{w}_t; \mathcal{S}_t')\|^2 \tag{C.25}$$

$$= L_{\text{rob}}(\boldsymbol{w}_t; \mathcal{S}) - \eta \left( \frac{\gamma_H'^2}{\cosh^2(\alpha) R_\alpha^2} - \frac{\beta \sigma_{\max}^2 \eta}{2} \right) \|\nabla L(\boldsymbol{w}_t; \mathcal{S}_t')\|^2 . \tag{C.26}$$

This establishes the first claim of Lemma C.11. Now, we can rewrite (C.25) to obtain the following.

$$\frac{L_{\text{rob}}(\boldsymbol{w}_t; \mathcal{S}) - L_{\text{rob}}(\boldsymbol{w}_{t+1}; \mathcal{S})}{\eta \left( \frac{\gamma_H'^2}{\cosh^2(\alpha) R_\alpha^2} - \frac{\beta \sigma_{\max}^2 \eta}{2} \right)} \geq \|\nabla L(\boldsymbol{w}_t; \mathcal{S}_t')\|^2 .$$

Note that our assumption on the step size $\eta$ ensures that the denominator in (C.25) is $\neq 0$.

Next, summing and telescoping gives us that

$$\sum_{k=0}^{t} \|\nabla L(\boldsymbol{w}_k; \mathcal{S}_k')\|^2 \leq \sum_{k=0}^{t} \frac{L_{\text{rob}}(\boldsymbol{w}_k; \mathcal{S}) - L_{\text{rob}}(\boldsymbol{w}_{k+1}; \mathcal{S})}{\eta \left( \frac{\gamma_H'^2}{\cosh^2(\alpha) R_\alpha^2} - \frac{\beta \sigma_{\max}^2 \eta}{2} \right)} = \frac{L_{\text{rob}}(\boldsymbol{w}_0; \mathcal{S}) - L_{\text{rob}}(\boldsymbol{w}_{t+1}; \mathcal{S})}{\eta \left( \frac{\gamma_H'^2}{\cosh^2(\alpha) R_\alpha^2} - \frac{\beta \sigma_{\max}^2 \eta}{2} \right)} ,$$

where the right term is bounded, since $L_{\text{rob}}(\boldsymbol{w}_0; \mathcal{S}) < \infty$ and $0 \leq L_{\text{rob}}(\boldsymbol{w}_{t+1}; \mathcal{S})$. This establishes the second claim of Lemma C.11 as

$$\sum_{k=0}^{\infty} \|\nabla L(\boldsymbol{w}_k; \mathcal{S}_k')\|^2 < \infty \quad \Rightarrow \quad \lim_{t \to \infty} \|\nabla L(\boldsymbol{w}_t; \mathcal{S}_t')\|^2 = 0 .$$

$\square$

**Lemma C.12.** *With the assumptions of Lemma C.11, Lemma C.11.1 implies for all $\boldsymbol{w} \in \mathbb{R}^{d+1}$*

$$2 \sum_{k=0}^{t-1} \eta_k \big( L_{\text{rob}}(\boldsymbol{w}_k; \mathcal{S}) - L_{\text{rob}}(\boldsymbol{w}; \mathcal{S}) \big) +$$

$$\sum_{k=0}^{t-1} \frac{\eta_k^2}{\bar{\eta}_k} \big( L_{\text{rob}}(\boldsymbol{w}_{k+1}; \mathcal{S}) - L_{\text{rob}}(\boldsymbol{w}_k; \mathcal{S}) \big) \leq \|\boldsymbol{w}_0 - \boldsymbol{w}\|^2 - \|\boldsymbol{w}_t - \boldsymbol{w}\|^2 ,$$

*where $\bar{\eta}_k = \eta_k \left( \frac{\gamma_H'^2}{\cosh(\alpha)^2 R_\alpha^2} - \frac{\beta \sigma_{\max}^2 \eta_k}{4} \right)$.*

*Proof.* First, note that the GD update

$$\boldsymbol{w}_{t+1} = \boldsymbol{w}_t - \eta_t \underbrace{\nabla L(\boldsymbol{w}_t; \mathcal{S}_t')}_{\in \partial L_{\text{rob}}(\boldsymbol{w}_t; \mathcal{S})}$$

implies that

$$\|\boldsymbol{w}_{t+1} - \boldsymbol{w}\|^2 = \|\boldsymbol{w}_t - \boldsymbol{w}\|^2 - 2\eta_t\langle\nabla L(\boldsymbol{w}_t; \mathcal{S}'_t),\ \boldsymbol{w}_t - \boldsymbol{w}\rangle + \eta_t^2\|\nabla L(\boldsymbol{w}_t; \mathcal{S}'_t)\|^2 \qquad \text{(C.27)}$$

$$= \|\boldsymbol{w}_t - \boldsymbol{w}\|^2 + 2\eta_t\langle\nabla L(\boldsymbol{w}_t; \mathcal{S}'_t),\ \boldsymbol{w} - \boldsymbol{w}_t\rangle + \eta_t^2\|\nabla L(\boldsymbol{w}_t; \mathcal{S}'_t)\|^2\ . \qquad \text{(C.28)}$$

Note that the hyperbolic logistic regression loss $f(z)$ in (4.5) is convex for $z < 0$. As a consequence, $l_{\mathrm{rob}}(\boldsymbol{x}, y; \boldsymbol{w})$ is convex for any adversarial example with $\mathrm{sgn}(\boldsymbol{w}_t * \boldsymbol{x}) \neq \mathrm{sgn}(\boldsymbol{w}_t * \tilde{\boldsymbol{x}})$. This implies that

$$l_{\mathrm{rob}}(\boldsymbol{x}, y; \boldsymbol{w}) \geq l_{\mathrm{rob}}(\boldsymbol{x}, y; \boldsymbol{w}_t) + \langle\partial l_{\mathrm{rob}}(\boldsymbol{x}, y; \boldsymbol{w}_t),\ \boldsymbol{w} - \boldsymbol{w}_t\rangle\ ,$$

for any $\boldsymbol{w} \in \mathbb{R}^{d+1}$ and any pair $(\boldsymbol{x}, y)$ for which an adversarial example exists.

Since the sum of convex function is convex, we further have

$$L_{\mathrm{rob}}(\boldsymbol{w}; \mathcal{S}) \geq L_{\mathrm{rob}}(\boldsymbol{w}_t; \mathcal{S}) + \langle\underbrace{\nabla L(\boldsymbol{w}_t; \mathcal{S}'_t)}_{\in \partial L_{\mathrm{rob}}(\boldsymbol{w}_t; \mathcal{S})},\ \boldsymbol{w} - \boldsymbol{w}_t\rangle\ . \qquad \text{(C.29)}$$

By combining (C.27) and (C.29), we obtain that

$$\|\boldsymbol{w}_{t+1} - \boldsymbol{w}\|^2 \leq \|\boldsymbol{w}_t - \boldsymbol{w}\|^2 + 2\eta_t\big(L_{\mathrm{rob}}(\boldsymbol{w}; \mathcal{S}) - L_{\mathrm{rob}}(\boldsymbol{w}_t; \mathcal{S})\big) + \eta_t^2\|\nabla L(\boldsymbol{w}_t; \mathcal{S}'_t)\|^2$$

$$\overset{(i)}{\leq} \|\boldsymbol{w}_t - \boldsymbol{w}\|^2 + 2\eta_t\big(L_{\mathrm{rob}}(\boldsymbol{w}; \mathcal{S}) - L_{\mathrm{rob}}(\boldsymbol{w}_t; \mathcal{S})\big) + \frac{\eta_t^2\big(L_{\mathrm{rob}}(\boldsymbol{w}_t; \mathcal{S}) - L_{\mathrm{rob}}(\boldsymbol{w}_{t+1}; \mathcal{S})\big)}{\bar{\eta}_t},$$

where $(i)$ follows from the first claim in Lemma C.11 and $\bar{\eta}_t := \eta_t\left(\frac{\gamma_H'^2}{\cosh^2(\alpha)R_\alpha^2} - \frac{\beta\sigma_{\max}^2\eta_t}{4}\right)$.

Next, summing and telescoping gives us that

$$\sum_{k=0}^{t-1}\|\boldsymbol{w}_{k+1} - \boldsymbol{w}\|^2 - \|\boldsymbol{w}_k - \boldsymbol{w}\|^2 \leq \sum_{k=0}^{t-1}\Big[2\eta_k\big(L_{\mathrm{rob}}(\boldsymbol{w}; \mathcal{S}) - L_{\mathrm{rob}}(\boldsymbol{w}_k; \mathcal{S})\big) +$$

$$\frac{\eta_k^2}{\bar{\eta}_k}\big(L_{\mathrm{rob}}(\boldsymbol{w}_k; \mathcal{S}) - L_{\mathrm{rob}}(\boldsymbol{w}_{k+1}; \mathcal{S})\big)\Big]$$

or

$$\|\boldsymbol{w}_t - \boldsymbol{w}\|^2 - \|\boldsymbol{w}_0 - \boldsymbol{w}\|^2 \leq 2\sum_{k=0}^{t-1}\eta_k\big(L_{\mathrm{rob}}(\boldsymbol{w}; \mathcal{S}) - L_{\mathrm{rob}}(\boldsymbol{w}_k; \mathcal{S})\big) +$$

$$\sum_{k=0}^{t-1}\frac{\eta_k^2}{\bar{\eta}_k}\big(L_{\mathrm{rob}}(\boldsymbol{w}_k; \mathcal{S}) - L_{\mathrm{rob}}(\boldsymbol{w}_{k+1}; \mathcal{S})\big).$$

Now, multiplying both sides by $-1$ completes the proof as follow.

$$2\sum_{k=0}^{t-1}\eta_k\big(L_{\mathrm{rob}}(\boldsymbol{w}_k; \mathcal{S}) - L_{\mathrm{rob}}(\boldsymbol{w}; \mathcal{S})\big) +$$

$$\sum_{k=0}^{t-1}\frac{\eta_k^2}{\bar{\eta}_k}\big(L_{\mathrm{rob}}(\boldsymbol{w}_{k+1}; \mathcal{S}) - L_{\mathrm{rob}}(\boldsymbol{w}_k; \mathcal{S})\big) \leq \|\boldsymbol{w}_0 - \boldsymbol{w}\|^2 - \|\boldsymbol{w}_t - \boldsymbol{w}\|^2\ .$$

$$\square$$

We are now in a position to present the desired convergence result.

**Theorem C.13** (Convergence GD update, Algorithm 2). *For a fixed constant $c \in (0, 1)$, let the step size $\eta_t := \eta = c \cdot \frac{2\sinh^2(\gamma_H)}{\beta\sigma_{\max}^2\cosh^2(\alpha)R_\alpha^2}$ and $\mathcal{A}$ be the GD update as defined in (4.4). Then, the iterates $\{\boldsymbol{w}_t\}$ in Algorithm 2 satisfy*

$$L_{\mathrm{rob}}(\boldsymbol{w}_t; \mathcal{S}) = O\left(\frac{\sinh^2(\ln(t))}{t} \cdot \left(\frac{\sinh(\gamma_H)}{\cosh(\alpha)}\right)^{-4}\right)\ .$$

*Proof.* Without loss of generality, assume that $\boldsymbol{w}_0 = (0, \boldsymbol{e}_i)$ where $\boldsymbol{e}_i \in \mathbb{R}^d$ is a standard basis vector whose $i$-th coordinate is 1. Note that this is a valid initialization, since $\boldsymbol{w}_0 * \boldsymbol{w}_0 < 0$; furthermore, we have $\|\boldsymbol{w}_0\| = 1$. Let $\boldsymbol{w}^* \in \mathbb{R}^{d+1}$ be a classifier that achieves the margin $\gamma_H$ on $\mathcal{S}$, i.e., $\forall (\boldsymbol{x}, y) \in \mathcal{S}$,

$$y(\boldsymbol{x} * \boldsymbol{w}^*) \geq \sinh(\gamma_H) \quad \Longleftrightarrow \quad \operatorname{asinh}\left(\frac{y(\boldsymbol{w}^* * \boldsymbol{x})}{\sqrt{-\boldsymbol{w}^* * \boldsymbol{w}^*}}\right) \geq \gamma_H.$$

Without loss of generality, assume that $\|\boldsymbol{w}^*\| = 1$. Let $\boldsymbol{u}_t := \frac{2R_\alpha \sinh(\ln(t)) \cosh(\alpha)}{\sinh(\gamma_H)} \boldsymbol{w}^*$; then $\|\boldsymbol{u}_t\| = \frac{2R_\alpha \sinh(\ln(t)) \cosh(\alpha)}{\sinh(\gamma_H)}$. We have

$$
\begin{aligned}
L_{\mathrm{rob}}(\boldsymbol{u}_t; \mathcal{S}'_t) &= \frac{1}{|\mathcal{S}'_t|} \sum_{(\boldsymbol{x}, y) \in \mathcal{S}'_t} l_{\mathrm{rob}}(\boldsymbol{x}, y; \boldsymbol{u}_t) = \frac{1}{|\mathcal{S}'_t|} \sum_{(\boldsymbol{x}, y) \in \mathcal{S}'_t} f(y(\tilde{\boldsymbol{x}} * \boldsymbol{u}_t)) \\
&\overset{(i)}{\leq} \frac{1}{|\mathcal{S}'_t|} \sum_{(\tilde{\boldsymbol{x}}, y) \in \mathcal{S}'_t} f\big(2R_\alpha \sinh(\ln(t))\big) = f\big(2R_\alpha \sinh(\ln(t))\big) \\
&\overset{(ii)}{\leq} \ln\big(1 + \exp\left(-\ln(t)\right)\big) \overset{(iii)}{\leq} \frac{1}{t},
\end{aligned}
\tag{C.30}
$$

where $(i)$ follows from

$$y(\tilde{\boldsymbol{x}} * \boldsymbol{u}_t) = \frac{2R_\alpha \sinh(\ln(t)) \cosh(\alpha)}{\sinh(\gamma_H)} \underbrace{y(\tilde{\boldsymbol{x}} * \boldsymbol{w}^*)}_{\geq \frac{\sinh(\gamma_H)}{\cosh(\alpha)}} \geq 2R_\alpha \sinh(\ln(t)),$$

$(ii)$ from $\sqrt{-\boldsymbol{u}_t * \boldsymbol{u}_t} \geq 1$ and $(iii)$ follows from the fact that $\ln(1 + x) \leq x$.

Now, consider

$$
\begin{aligned}
2\eta(t-1)\big(L_{\mathrm{rob}}(\boldsymbol{w}_t; \mathcal{S}) - L_{\mathrm{rob}}(\boldsymbol{u}_t; \mathcal{S})\big) &\overset{(i)}{=} 2\sum_{k=0}^{t-1} \eta_k \big(L_{\mathrm{rob}}(\boldsymbol{w}_t; \mathcal{S}) - L_{\mathrm{rob}}(\boldsymbol{u}_t; \mathcal{S})\big) \\
&= 2\sum_{k=0}^{t-1} \eta_k \big(L_{\mathrm{rob}}(\boldsymbol{w}_t; \mathcal{S}) - L_{\mathrm{rob}}(\boldsymbol{u}_t; \mathcal{S}) + L_{\mathrm{rob}}(\boldsymbol{w}_k; \mathcal{S}) - L_{\mathrm{rob}}(\boldsymbol{w}_k; \mathcal{S})\big) \\
&= 2\sum_{k=0}^{t-1} \eta_k \big(L_{\mathrm{rob}}(\boldsymbol{w}_k; \mathcal{S}) - L_{\mathrm{rob}}(\boldsymbol{u}_t; \mathcal{S})\big) + 2\sum_{k=0}^{t-1} \eta_k \big(L_{\mathrm{rob}}(\boldsymbol{w}_t; \mathcal{S}) - L_{\mathrm{rob}}(\boldsymbol{w}_k; \mathcal{S})\big) \\
&\overset{(ii)}{\leq} 2\sum_{k=0}^{t-1} \eta_k \big(L_{\mathrm{rob}}(\boldsymbol{w}_k; \mathcal{S}) - L_{\mathrm{rob}}(\boldsymbol{u}_t; \mathcal{S})\big) + \sum_{k=0}^{t-1} \eta_k \big(L_{\mathrm{rob}}(\boldsymbol{w}_{k+1}; \mathcal{S}) - L_{\mathrm{rob}}(\boldsymbol{w}_k; \mathcal{S})\big) \\
&\leq 2\sum_{k=0}^{t-1} \eta_k \big(L_{\mathrm{rob}}(\boldsymbol{w}_k; \mathcal{S}) - L_{\mathrm{rob}}(\boldsymbol{u}_t; \mathcal{S})\big) + \sum_{k=0}^{t-1} \frac{\eta_k^2}{\bar{\eta}_k} \big(L_{\mathrm{rob}}(\boldsymbol{w}_{k+1}; \mathcal{S}) - L_{\mathrm{rob}}(\boldsymbol{w}_k; \mathcal{S})\big) \\
&\overset{(iii)}{\leq} \|\boldsymbol{w}_0 - \boldsymbol{u}_t\|^2 - \|\boldsymbol{w}_t - \boldsymbol{u}_t\|^2,
\end{aligned}
$$

where $(i)$ holds as we have a constant step-size, i.e., $\eta_k = \eta$ and $(ii)$ follows from the fact that

$$L_{\mathrm{rob}}(\boldsymbol{w}_t; \mathcal{S}) \leq L_{\mathrm{rob}}(\boldsymbol{w}_{k+1}; \mathcal{S}) \quad \text{for } 0 \leq k \leq t-1.$$

The inequality in $(iii)$ follows from Lemma C.12 with $\boldsymbol{w} = \boldsymbol{u}_t$.

We can rewrite this as

$$
\begin{aligned}
L_{\mathrm{rob}}(\boldsymbol{w}_t; \mathcal{S}) &\leq L_{\mathrm{rob}}(\boldsymbol{u}_t; \mathcal{S}) + \frac{\|\boldsymbol{w}_0 - \boldsymbol{u}_t\|^2 - \|\boldsymbol{w}_t - \boldsymbol{u}_t\|^2}{2\sum_{k=0}^{t-1} \eta_k} \\
&\overset{(i)}{\leq} \frac{1}{t} + \frac{\|\boldsymbol{w}_0 - \boldsymbol{u}_t\|^2}{2(t-1)\eta} \\
&\overset{(ii)}{\leq} \frac{1}{t} + \frac{2\|\boldsymbol{w}_0\|^2 + 2\|\boldsymbol{u}_t\|^2}{2(t-1)\eta} = \frac{1}{t} + \frac{\|\boldsymbol{w}_0\|^2 + \|\boldsymbol{u}_t\|^2}{(t-1)\eta}
\end{aligned}
$$

where $(i)$ follows from (C.30) and $(ii)$ follows from $(a+b)^2 \leq 2a^2 + 2b^2$. Now, using the fact that $\|\boldsymbol{w}_0\| = 1$ and $\|\boldsymbol{u}_t\| = \frac{2R_\alpha \sinh(\ln(t)) \cosh(\alpha)}{\sinh(\gamma_H)}$, we obtain that

$$L_{\text{rob}}(\boldsymbol{w}_t; \mathcal{S}) \leq \frac{1}{t} + \frac{1 + 4R_\alpha^2 \left(\cosh(\alpha)/\sinh(\gamma_H)\right)^2 \cdot \sinh^2(\ln(t))}{(t-1)\eta}. \tag{C.31}$$

By substituting $\eta = c \cdot \frac{2\sinh^2(\gamma_H)}{\beta \sigma_{\max}^2 \cosh^2(\alpha) R_\alpha^2}$, we get

$$L_{\text{rob}}(\boldsymbol{w}_t; \mathcal{S}) = O\left( \frac{\sinh^2(\ln(t))}{t} \cdot \left( \frac{\sinh(\gamma_H)}{\cosh(\alpha)} \right)^{-4} \right).$$

$\square$

**Theorem C.14** (Iteration complexity). *Consider Algorithm 2 with $\eta_t := \eta = c \cdot \frac{2\sinh^2(\gamma_H)}{\beta \sigma_{\max}^2 \cosh^2(\alpha) R_\alpha^2}$ and $\mathcal{A}$ being the GD update. Then Algorithm 2 converges as $\Omega\left( \text{poly}\left( \frac{\sinh(\gamma_H)}{\cosh(\alpha)} \right) \right)$.*

*Proof.* Let $\varrho = \frac{\ln(1+1/e)}{\ln(1+e)}$. We first argue that

$$L_{\text{rob}}(\boldsymbol{w}_t; \mathcal{S}) \leq \varrho \cdot \ln\left( 1 + \exp\left( -\frac{\gamma_H}{\cosh(\alpha)} \right) \right) \tag{C.32}$$

implies that $\boldsymbol{w}_t$ achieves margin $\gamma_H/\cosh(\alpha)$ on $\mathcal{S}$. To see this, note that

$$L_{\text{rob}}(\boldsymbol{w}_t; \mathcal{S}) = \frac{1}{|\mathcal{S}|} \sum_{(\boldsymbol{x},y) \in \mathcal{S}} l_{\text{rob}}(\boldsymbol{x}, y; \boldsymbol{w}_t)$$

$$= \underbrace{\max_{(\boldsymbol{x},y) \in \mathcal{S}} l_{\text{rob}}(\boldsymbol{x}, y; \boldsymbol{w}_t)}_{:=l_{\text{rob}}^{\max}(\mathcal{S})} \cdot \frac{1}{|\mathcal{S}|} \sum_{(\boldsymbol{x},y) \in \mathcal{S}} \frac{l_{\text{rob}}(\boldsymbol{x}, y; \boldsymbol{w}_t)}{l_{\text{rob}}^{\max}(\mathcal{S})}$$

$$\geq l_{\text{rob}}^{\max} \cdot \frac{1}{|\mathcal{S}|} \sum_{(\boldsymbol{x},y) \in \mathcal{S}} \varrho = \varrho \cdot l_{\text{rob}}^{\max}. \tag{C.33}$$

The last inequality in (C.33) holds as, for each $(\boldsymbol{x}, y) \in \mathcal{S}$, we have

$$\ln(1+1/e) \overset{(i)}{\leq} \underbrace{\ln\left( 1 + \exp\left( -\text{asinh}\left( \frac{y(\tilde{\boldsymbol{x}} * \boldsymbol{w}_t)}{2R_\alpha} \right) \right) \right)}_{=l_{\text{rob}}(\boldsymbol{x},y;\boldsymbol{w}_t)} \leq \max_{(\boldsymbol{x},y) \in \mathcal{S}} l_{\text{rob}}(\boldsymbol{x}, y; \boldsymbol{w}_t) \overset{(ii)}{\leq} \ln(1+e),$$

where $(i)$ and $(ii)$ follows from Assumption 1.2. Thus, for each $(\boldsymbol{x}, y) \in \mathcal{S}$, we have

$$\frac{l_{\text{rob}}(\boldsymbol{x}, y; \boldsymbol{w}_t)}{l_{\text{rob}}^{\max}} \geq \frac{\ln(1+1/e)}{\ln(1+e)} = \varrho. \tag{C.34}$$

Now, by combining (C.32) and (C.33), we obtain that

$$l_{\text{rob}}(\boldsymbol{x}, y; \boldsymbol{w}_t) \leq \ln\left( 1 + \exp\left( -\frac{\gamma_H}{\cosh(\alpha)} \right) \right)$$

for any $(\boldsymbol{x}, y) \in \mathcal{S}$. Equivalently, for each $(\boldsymbol{x}, y) \in \mathcal{S}$,

$$l_{\text{rob}}(\boldsymbol{x}, y; \boldsymbol{w}_t) = \ln\left( 1 + \exp\left( -\text{asinh}\left( \frac{y(\tilde{\boldsymbol{x}} * \boldsymbol{w}_t)}{2R_\alpha} \right) \right) \right)$$

$$\leq \ln\left( 1 + \exp\left( -\frac{\gamma_H}{\cosh(\alpha)} \right) \right). \tag{C.35}$$

Thus, for each $(\boldsymbol{x}, y) \in \mathcal{S}$, we have

$$\text{asinh}\left( \frac{y(\tilde{\boldsymbol{x}} * \boldsymbol{w}_t)}{\sqrt{-\boldsymbol{w}_t * \boldsymbol{w}_t}} \right) \overset{(i)}{\geq} \text{asinh}\left( \frac{y(\tilde{\boldsymbol{x}} * \boldsymbol{w}_t)}{2R_\alpha} \right) \overset{(ii)}{\geq} \frac{\gamma_H}{\cosh(\alpha)}.$$

where (i) from the definition of $R_\alpha$ (cf. Assumption 1) and (ii) from Eq. C.35. Thus, $w_t$ achieves margin $\gamma_H/\cosh(\alpha)$ on $\mathcal{S}$.

Next, introduce the following constant:

$$C_q := \inf\{t \geq 2 : \ 2 + \ln(t)^2 \leq (t-1)t^{-1/q}\} \ .$$

With this, for $t \geq C_q$, we can rewrite the bound in (C.31) as follows:

$$L_{\mathrm{rob}}(\boldsymbol{w}_t; \mathcal{S}) \leq \underbrace{\frac{1}{t}}_{\leq \frac{1}{(t-1)\eta}} + \frac{1 + \sinh(\ln(t))^2 \left(\frac{\sinh(\gamma_H)}{\cosh(\alpha)}\right)^{-2}}{(t-1)\eta} \leq \frac{2 + \sinh(\ln(t))^2 \left(\frac{\sinh(\gamma_H)}{\cosh(\alpha)}\right)^{-2}}{(t-1)\eta}$$

$$\leq \frac{2 + \ln(t)^2 \left(\frac{\sinh(\gamma_H)}{\cosh(\alpha)}\right)^{-2}}{(t-1)\eta} \leq \frac{(t-1)t^{-1/q}}{\eta(t-1)} \left(\frac{\sinh(\gamma_H)}{\cosh(\alpha)}\right)^{-2} \leq \frac{t^{-1/q}}{\eta \left(\frac{\sinh(\gamma_H)}{\cosh(\alpha)}\right)^2} \ .$$

Solving for $t$ and plugging in the above bound on $L_{\mathrm{rob}}$ for which $w_t$ achieves the desire margin, as well as $\eta = c \cdot \frac{2\sinh^2(\gamma_H)}{\beta\sigma_{\max}^2 \cosh^2(\alpha)R_\alpha^2}$, we get

$$t = \max\{C_q, \Omega\left(\left((\sinh(\gamma_H)^4/\cosh(\alpha)^4)\right)^{-q}\right)\} \ ,$$

from which the claim follows directly. $\qquad\square$

### C.5 Algorithm 2 with an ERM update

Consider the unit sphere $\mathbb{S}^{d-1} \subseteq \mathbb{R}^d$. A spherical code with minimum separation $\theta$ is a subset of $\mathbb{S}^{d-1}$, such that any two distinct elements $\boldsymbol{u}, \boldsymbol{u}'$ in the subset are separated by at least an angle $\theta$, i.e. $\langle \boldsymbol{u}, \boldsymbol{u}' \rangle \leq \cos\theta$. We denote the size of the largest such code as $A(d, \theta)$. A similar construction can be made in hyperbolic space, which allows the transfer of bounds on $A(d, \theta)$ to hyperbolic space [7].

The following lemma shows that a spherical code with a suitable minimum separation $\theta$ enables a simple pathological training set such that Algorithm 2 along with an ERM update rule cannot produce a classifier with a desired margin in a small number of iteration. In particular, the lemma shows that the number of iterations required to find the desire margin is lower-bounded by the size of the underlying spherical code.

**Lemma C.15.** *Consider* $\mathcal{S} = \{(\boldsymbol{x}_1, y_1) = ((1, 0, \ldots, 0), 1), (\boldsymbol{x}_2, y_2) = ((-1, 0, \ldots, 0), -1)\}$, *where* $\boldsymbol{x}_1, \boldsymbol{x}_2 \in \mathbb{L}^d$ *and* $y_1, y_2$ *the corresponding labels. For any* $\epsilon < \alpha$, *there is an admissible sequence of classifiers* $\{\boldsymbol{w}_t\}_{1 \leq t \leq T}$, *with*

$$T = A\left(d, \arccos\left(\rho \cdot \frac{\sinh(\epsilon)\cosh(\alpha)}{\sqrt{\cosh^2(\alpha)-1}\sqrt{1+\sinh^2(\epsilon)}}\right)\right)$$

*Proof.* First, note that $\boldsymbol{x}_1 * \boldsymbol{x}_1 = \boldsymbol{x}_2 * \boldsymbol{x}_2 = 1$, i.e., $\boldsymbol{x}_1, \boldsymbol{x}_2 \in \mathbb{L}^d$ as desired. Let $\epsilon' = \sinh(\epsilon)$ and $\boldsymbol{e}_i \in \mathbb{R}^{d+1}$ denotes the standard basis vector that has its $i$-th coordinate equal to 1. Now, consider classifiers of the form

$$\boldsymbol{w}_t = \left(\frac{\epsilon'}{\sqrt{1+\epsilon'^2}} \boldsymbol{v}_t\right) \quad \text{where} \quad \boldsymbol{v}_t \in \mathcal{C}\left(d, \arccos\left(\rho \cdot \frac{\epsilon'\sqrt{1+\delta^2}}{\delta\sqrt{1+\epsilon'^2}}\right)\right) \quad \forall \ 1 \leq t \leq T \ , \quad \text{(C.36)}$$

where $\rho < 1$; and $\mathcal{C}\left(d, \arccos\left(\rho \cdot \frac{\epsilon'\sqrt{1+\delta^2}}{\delta\sqrt{1+\epsilon'^2}}\right)\right)$ be the spherical code with the minimum separation $\theta = \arccos\left(\rho \cdot \frac{\epsilon'\sqrt{1+\delta^2}}{\delta\sqrt{1+\epsilon'^2}}\right)$ and size $A(d, \theta)$. Since $\boldsymbol{w}_t * \boldsymbol{w}_t = (\epsilon')^2 - 1 - (\epsilon')^2 = -1$, we have $\boldsymbol{w}_t * \boldsymbol{w}_t < 0$ for all $t$. This guarantees that the intersections of the decision boundaries defined by $\{\boldsymbol{w}_t\}$ and $\mathbb{L}^d$ are not empty. Moreover, $\{\boldsymbol{w}_t\}_t$ is an admissible sequence of classifiers with margin $\leq \epsilon$. To see this, note that, for $t = 1, \ldots, T$,

$$\boldsymbol{w}_t * \boldsymbol{x}_1 = \epsilon' > 0$$
$$\boldsymbol{w}_t * \boldsymbol{x}_2 = -\epsilon' < 0 \ ,$$

i.e., $\{w_t\}$ correctly classifies $\mathcal{S}$. Furthermore, with $-w_t * w_t = 1$, we have

$$\operatorname{asinh}\left(\frac{y_1(w_t * x_1)}{\sqrt{-w_t * w_t}}\right) = \operatorname{asinh}\left(\frac{y_2(w_t * x_2)}{\sqrt{-w_t * w_t}}\right) = \epsilon ,$$

which gives $\operatorname{margin}_\mathcal{S}(w_t) = \epsilon$.

Now we perturb $x_1, x_2$ on $\mathbb{L}^d$ such that the magnitude of the perturbation is at most $\alpha$, i.e., we want to find $\tilde{x}_1, \tilde{x}_2 \in \mathbb{L}^d$ such that both $d_\mathbb{L}(x_1, \tilde{x}_1)$ and $d_\mathbb{L}(x_2, \tilde{x}_2)$ are at most $\alpha$. For $1 \le t \le T$, consider adversarial examples of the form

$$\tilde{x}_{1t} = \begin{pmatrix} \sqrt{1+\delta^2} \\ \delta v_t \end{pmatrix} \quad \text{and} \quad \tilde{x}_{2t} = -\begin{pmatrix} \sqrt{1+\delta^2} \\ \delta v_t \end{pmatrix} .$$

Note that $\tilde{x}_{1t}, \tilde{x}_{2t} \in \mathbb{L}^d$ as $\tilde{x}_{1i} * \tilde{x}_{1t} = \tilde{x}_{2t} * \tilde{x}_{2i} = 1$. Let us verify the two conditions that we require the valid adversarial examples to satisfy:

- **Adversarial budget.** Note that we have

$$d_\mathbb{L}(x_1, \tilde{x}_{1t}) = d_\mathbb{L}(x_2, \tilde{x}_{2t}) = \operatorname{acosh}(\sqrt{1+\delta^2}) .$$

Thus, by choosing $\delta = \sqrt{\cosh^2(\alpha) - 1}$, we achieve the maximal permitted perturbation $\alpha$.

- **Inconsistent prediction for the current classifier, i.e.,** $h_{w_t}(\tilde{x}_{1t/2t}) \ne h_w(x_{1/2})$. Note that we have $\delta \ge \alpha > \epsilon$, which further implies that $\delta > \epsilon \ge \epsilon'$. In round $t$,

$$w_t * \tilde{x}_{1t} = \epsilon' \sqrt{1+\delta^2} - \delta\sqrt{1+\epsilon'^2} < 0$$
$$w_t * \tilde{x}_{2t} = -\epsilon'\sqrt{1+\delta^2} + \delta\sqrt{1+\epsilon'^2} > 0 ,$$

which is a consequence of the relation $\delta > \epsilon'$ as follows:

$$\delta^2 > \epsilon'^2 \Rightarrow \delta^2 + \epsilon'^2\delta^2 > \epsilon'^2 + \epsilon'^2\delta^2 \Rightarrow \delta^2(1+\epsilon'^2) > \epsilon'^2(1+\delta^2)$$
$$\Rightarrow \delta\sqrt{1+\epsilon'^2} > \epsilon'\sqrt{1+\delta^2} .$$

Recall that, in each round of Algorithm 2 with an ERM update, we create adversarial examples and add them to the training set, i.e., after round $t$ we have

$$\mathcal{S}_{<t} = \mathcal{S} \cup \bigcup_{i=0}^{t-1}\{(\tilde{x}_{1i}, y_{1i}), (\tilde{x}_{2i}, y_{2i})\} .$$

Now for each $t$ and any $i < t$, we have

$$w_t * \tilde{x}_{1i} = \epsilon'\sqrt{1+\delta^2} - \delta\sqrt{1+\epsilon'^2} \cdot \cos(\theta) > 0$$
$$w_t * \tilde{x}_{2i} = -\epsilon'\sqrt{1+\delta^2} + \delta\sqrt{1+\epsilon'^2} \cdot \cos(\theta) < 0 ,$$

i.e., $w_t$ linearly separates $\mathcal{S}_{<t}$.

Therefore, $\{w_t\}$ in (C.36) form an admissible sequence of the classifiers, where $w_t$ linearly separates $\mathcal{S}_t$ while achieving the margin of at most $\epsilon$ on the original dataset $\mathcal{S}$. The length of the sequence is bounded by the size of the spherical code $\mathcal{C}(d, \epsilon'\cosh(\alpha))$, which give us that

$$T = A\left(d, \arccos\left(\rho \cdot \frac{\epsilon'\sqrt{1+\delta^2}}{\delta\sqrt{1+\epsilon'^2}}\right)\right) = A\left(d, \arccos\left(\rho \cdot \frac{\sinh(\epsilon)\cosh(\alpha)}{\sqrt{\cosh^2(\alpha)-1}\sqrt{1+\sinh^2(\epsilon)}}\right)\right) .$$

$\square$

The following result (a restatement of Theorem 4.7 from the main text) then follows by applying a lower bound on the maximal size of spherical codes by Shannon.

**Theorem C.16** (Theorem 4.7). *Suppose Algorithm 2 (with an ERM update) outputs a linear seperator of $\mathcal{S} \cup \mathcal{S}'$. In the worst case, the number of iteration required to achieve the margin at least $\epsilon$ is $\Omega\left(\exp(d)\right)$.*

*Proof.* The statement of the theorem follows from combining Lemma C.15 with Shannon's lower bound (Theorem A.4) on the maximal size of spherical codes, namely

$$T \geq (1 + o(1))\sqrt{2\pi d}\,\frac{\cos(\theta)}{\sin^{d-1}(\theta)} \ .$$

We introduce the shorthand $\theta =: \arccos\left(\frac{A}{B}\right)$, where $A = \rho\sinh(\epsilon)\cosh(\alpha)$ and $B = \sqrt{\cosh^2(\alpha) - 1}\sqrt{1 + \sinh^2(\epsilon)}$, as given by Lemma C.15. We then use two well-known trigonometric identities

$$\cos(\arccos z) = z \quad \text{and} \quad \sin(\arccos z) = \sqrt{1 - z^2}$$

to simplify the trignometric fraction in Shannon's bound:

$$\frac{\cos\theta}{\sin^{d-1}\theta} = \frac{A}{B\left(1 - \frac{A^2}{B^2}\right)^{\frac{d-1}{2}}} = \frac{AB^{d-2}}{\left(B^2 - A^2\right)^{\frac{d-1}{2}}} \ .$$

For the denominator, note that

$$\begin{aligned}
B^2 - A^2 &= (\cosh^2(\alpha) - 1)(1 + \sinh^2(\epsilon)) - \rho^2\sinh^2(\epsilon)\cosh^2(\alpha) \\
&= (1 - \rho^2)\sinh^2(\epsilon)\cosh^2(\alpha) + \cosh^2(\alpha) - 1 - \sinh^2(\epsilon) \\
&\overset{(i)}{\simeq} \cosh^2(\alpha) - 1 - \sinh^2(\epsilon)
\end{aligned}$$

where (i) follows from the fact that we can choose $\rho$ arbitrary close to 1. Putting everything together, we have the lower bound

$$T \geq (1 + o(1))\sqrt{2d}\,\frac{\rho\sinh(\epsilon)\cosh(\alpha)\left(\sqrt{\cosh^2(\alpha) - 1}\sqrt{1 + \sinh^2(\epsilon)}\right)^{d-2}}{\left(\cosh^2(\alpha) - 1 - \sinh^2(\epsilon)\right)^{\frac{d-1}{2}}} = \Omega(\exp d) \ ,$$

which is exponential in $d$. $\qquad\square$

# D   Dimension-distortion trade-off

## D.1   Euclidean case

In the Euclidean case, we relate the distance of the support vectors and the size of margin via side length - altitude relations. Let $\boldsymbol{x}, \boldsymbol{y} \in \mathbb{R}^d$ denote support vectors, such that $\langle \boldsymbol{x}, \boldsymbol{w} \rangle > 0$ and $\langle \boldsymbol{y}, \boldsymbol{w} \rangle < 0$ and $\mathrm{margin}(\boldsymbol{w}) = \epsilon$. We can rotate the decision boundary, such that the support vectors are not unique. Wlog, assume that $\boldsymbol{x}_1, \boldsymbol{x}_2$ are equidistant from the decision boundary and $\|\boldsymbol{w}\| = 1$. In this setting, we show the following relation:

**Theorem D.1** (Thm. 5.1). $\epsilon' \geq \frac{\epsilon}{c_E^3}$.

*Proof.* Let $d_1 = d_\mathcal{X}(\phi_E^{-1}(\boldsymbol{x}_1), \phi_E^{-1}(\boldsymbol{y}))$, $d_2 = d_\mathcal{X}(\phi_E^{-1}(\boldsymbol{x}_2), \phi_E^{-1}(\boldsymbol{y}))$ and $d_3 = d_\mathcal{X}(\phi_E^{-1}(\boldsymbol{x}_1), \phi_E^{-1}(\boldsymbol{x}_2))$ the distances between the support vectors in the original space. In the Euclidean embedding space we have

$$d_1' = d_E(\boldsymbol{x}_1, \boldsymbol{y}) \geq \frac{d_1}{c_E}$$

$$d_2' = d_E(\boldsymbol{x}_2, \boldsymbol{y}) \geq \frac{d_2}{c_E}$$

$$d_3' = d_E(\boldsymbol{x}_1, \boldsymbol{x}_2) \geq \frac{d_3}{c_E} \ .$$

$d'_1, d'_2, d'_3$ are the side lengths of a triangle, whose altitude is given by the margin: $h = 2\epsilon'$. With Heron's equation we get

$$h = 2\epsilon' = \frac{2}{d'_3}\sqrt{s'(s'-d'_1)(s'-d'_2)(s'-d'_3)} \,,$$

where $s' = \frac{1}{2}(d'_1 + d'_2 + d'_3)$. In $\mathcal{X}$ we have $s' = \frac{1}{2c_E}(d_1 + d_2 + d_3) = \frac{s}{c_E}$. Then we have with respect to the actual distance relations

$$h = 2\epsilon' \geq \frac{2}{c_E d_3}\sqrt{c_E^{-4}s(s-d_1)(s-d_2)(s-d_3)} = 2\frac{\epsilon}{c_E^3} \,,$$

which gives the claim. $\qquad\square$

## D.2 Hyperbolic case

As in the Euclidean case, we want to relate the margin to the distance of the support vectors. Since the distortion can be expressed in terms of the distances of support vector in the original and the embedding space, this allows us to study the influence of distortion on the margin.

We will derive the relation in the half-space model ($\mathbb{P}^2$). However, since the theoretical guarantees above consider the upper sheet of the Lorentz model ($\mathbb{L}_+^{d'}$), we have to map between the two spaces.

**Assumption 4.** We make the following assumptions on the underlying data $\mathcal{X}$ and the embedding $\phi_H$:

1. $\mathcal{X}$ is linearly separable;

2. $\mathcal{X}$ is hierarchical, i.e., has a partial order relation;

3. $\phi_H$ preserves the partial order relation and the root is mapped onto the origin of the embedding space.

Under these assumptions, the hyperbolic embedding $\phi_H$ has two sources of distortion:

1. the (multiplicative) distortion of pairwise distances, measured by the factor $\frac{1}{c_H}$;

2. the distortion of order relations, in most embedding models captured by the alignment of ranks with the Euclidean norm.

Under Ass. 4, order relationships are preserved and the root is mapped to the origin. Therefore, the distortion on the Euclidean norms is given as follows:

$$\|\phi_H(x)\| = d_E(\phi_H(\boldsymbol{x}), \phi_H(0)) = \frac{d_{\mathcal{X}}(\boldsymbol{x}, 0)}{c_H} \,,$$

i.e., the distortion on both pairwise distances and norms is given by a factor $\frac{1}{c_H}$.

*Note on notation:* In the following, a bar over any symbol indicates the Euclidean expression.

### D.2.1 Mapping from $\mathbb{L}_+^{d'}$ to $\mathbb{P}^2$

First, note that a transformation $\boldsymbol{v} \mapsto B\boldsymbol{v}$ with $B = \begin{pmatrix} 1 & 0 \\ 0 & A \end{pmatrix}$ and an orthogonal matrix $A$ is isometric, i.e., it preserves the Minkowski product [6]:

$$(B\boldsymbol{u}) * (B\boldsymbol{v}) = u_0 v_0 - \boldsymbol{u}_{1:d'}^T A^T A \boldsymbol{v}_{1:d'} = u_0 v_0 - \boldsymbol{u}_{1:d'}^T \boldsymbol{v}_{1:d'} = \boldsymbol{u} * \boldsymbol{v} \,.$$

Setting the first column of $A$ to $\frac{\boldsymbol{w}_{1:d'}}{\|\boldsymbol{w}_{1:d'}\|}$ we can isometrically transform the decision hyperplane as $\hat{\boldsymbol{w}} = B\boldsymbol{w} = (\hat{\boldsymbol{w}}_0, \|\hat{\boldsymbol{w}}_{1:d'}\|, 0, \dots, 0)$. Analogously, we can transform any point in $\mathbb{L}_+^{d'}$. In the following, we will use the shorthand $\lambda = \frac{\hat{w}_0}{\hat{w}_1}$. We can then use the maps defined in section A.2 to map $\hat{\boldsymbol{x}} = Bx \in \mathbb{L}_+^2$ onto $\boldsymbol{z} \in \mathbb{P}^2$, i.e. applying $(\pi_{BP} \circ (\pi_{LB} \circ B))$ to any $\boldsymbol{x} \in \mathbb{L}_+^2$ gives $\boldsymbol{z} \in \mathbb{P}^2$.

**Remark D.2** (Effect of hyperbolic distortion on Euclidean distances in the Poincare half plane). Note that the hyperbolic distance in the Poincare half plane can be written as follows:

$$d_{\mathbb{P}}((x_0, x_1), (y_0, y_1)) = 2 \operatorname{asinh}\left(\frac{1}{2}\sqrt{\frac{(x_0 - y_0)^2 + (x_1 - y_1)^2}{x_1 y_1}}\right)$$

$$= 2 \operatorname{asinh}\left(\frac{1}{2}\frac{d_E((x_0, x_1), (y_0, y_1))}{\sqrt{x_1 y_1}}\right) .$$

If $c_H$ denotes the hyperbolic distortion, we get

$$d'_{\mathbb{P}} = \frac{d_{\mathbb{P}}}{c_H} = 2 \operatorname{asinh}\left(\frac{1}{2}\frac{d'_E}{\sqrt{x_1 y_1}}\right)$$

$$\Rightarrow \quad \frac{1}{2}\frac{d'_E}{\sqrt{x_1 y_1}} = \sinh\left(\frac{2 \operatorname{asinh}\left(\frac{1}{2}\frac{d_E}{\sqrt{x_1 y_1}}\right)}{2 c_H}\right) \gtrsim \frac{1}{2}\frac{d_E}{c_H \sqrt{x_1 y_1}} .$$

This suggests, that the effect of hyperbolic distortion on the Euclidean distances can be quantified by a comparable factor, i.e. $d'_E \gtrsim \frac{d_E}{c_H}$.

**Lemma D.3** (Relation between h-margin and E-margin). *Let $\gamma_H$ be the margin of a hyperbolic classifier $\boldsymbol{w} \in \mathbb{R}^{d'+1}$. Then the Euclidean margin $\gamma_E$ of $\boldsymbol{w}$ is bounded as follows: $\gamma_E \geq \sinh(\gamma_H)$.*

*Proof.* We again write the hyperbolic distance in the Poincare half plane in terms of the Euclidean distance of the ambient space:

$$d_{\mathbb{P}}((x_0, x_1), (y_0, y_1)) = 2 \operatorname{asinh}\left(\frac{1}{2}\sqrt{\frac{(x_0 - y_0)^2 + (x_1 - y_1)^2}{x_1 y_1}}\right)$$

$$= 2 \operatorname{asinh}\left(\frac{1}{2}\frac{d_E((x_0, x_1), (y_0, y_1))}{\sqrt{x_1 y_1}}\right) ,$$

where $\boldsymbol{y} \in \mathcal{H}_w$ is the point closest to the support vector $\boldsymbol{x} \in \mathbb{L}^{d'}_+$ on the decision boundary. Therefore, the hyperbolic margin is $d_{\mathbb{P}}(\boldsymbol{x}, \boldsymbol{y}) = \gamma_H$ and the Euclidean margin is $d_E(\boldsymbol{x}, \boldsymbol{y}) = \gamma_E$.

Since we mapped the feature space onto the Poincare half plane, $\boldsymbol{y}$ has the coordinates $\boldsymbol{y} = (\tilde{y}_0, \tilde{y}_1, 0, \ldots, 0)$ where $\tilde{y}_0 = y_0$ and $\tilde{y}_1 = \frac{\boldsymbol{w}'^T \boldsymbol{y}'}{\|\boldsymbol{w}'\|}$. Similarly, $\boldsymbol{x}$ has the coordinates $\boldsymbol{x} = (\tilde{x}_0, \tilde{x}_1, 0, \ldots, 0)$. The transformation preserves the Minkowski product. Therefore we have

$$\boldsymbol{y} * \boldsymbol{y} = y_0^2 - \boldsymbol{y}'^2 = \hat{y}_0^2 - \underbrace{\left(\frac{\boldsymbol{w}'^T \boldsymbol{y}'}{\|\boldsymbol{w}'\|}\right)^2}_{=\hat{y}_1} = 1$$

and similarly $\boldsymbol{x} * \boldsymbol{x} = \hat{x}_0^2 - \hat{x}_1^2 = 1$. This implies

$$\textcircled{1} \quad \hat{y}_1 = \sqrt{\hat{y}_0^2 - 1}, \ \hat{x}_1 = \sqrt{\hat{x}_0^2 - 1} ,$$

and further

$$\textcircled{2} \quad \hat{x}_0, \ \hat{y}_0 \geq 1 .$$

We want to show that $\hat{x}_1 \hat{y}_1 \geq 1$. For this, first, note that since $\boldsymbol{y} \in \mathcal{H}_w$ and the hyperbolic margin is $\gamma_H$, we have

$$0 = \boldsymbol{w} * \boldsymbol{y} = w_0 y_0 - \boldsymbol{w}'^T \boldsymbol{y}'$$

$$\Rightarrow \boldsymbol{w}'^T \boldsymbol{y}' = w_0 y_0 .$$

This gives

$$\boldsymbol{y} * \boldsymbol{y} = y_0^2 - \frac{w_0^2 y_0^2}{\|\boldsymbol{w}'\|^2} = 1$$

$$\Rightarrow 0 = y_0^2 - \frac{w_0^2 y_0^2}{\|\boldsymbol{w}'\|^2} - 1 ,$$

and therefore

$$\textcircled{3}\quad y_0 = \frac{1}{\sqrt{1 - \frac{w_0^2}{\|\boldsymbol{w}'\|}}} \ .$$

Since the hyperbolic margin is $\gamma_H$, we further have

$$d_{\mathbb{P}}(\boldsymbol{x}, \boldsymbol{y}) = \mathrm{acosh}(\boldsymbol{x} * \boldsymbol{y}) \geq \gamma_H \quad \Rightarrow \quad \boldsymbol{x} * \boldsymbol{y} \geq \cosh(\gamma_H) \geq 1 \ ,$$

and therefore

$$\begin{aligned}
x_0 y_0 - x_1 y_1 &\geq 1 \\
x_0 y_0 - \sqrt{x_0^2 - 1}\sqrt{y_0^2 - 1} &\geq 1 \\
(x_0 y_0 - 1)^2 &\geq (x_0^2 - 1)(y_0^2 - 1) \\
x_0^2 y_0^2 - 2x_0 y_0 + 1 &\geq x_0^2 y_0^2 - x_0^2 - y_0^2 + 1 \\
\Rightarrow 0 &\leq (x_0 - y_0)^2 \ ,
\end{aligned}$$

which implies

$$\textcircled{4}\quad x_0 \geq y_0 \ .$$

This gives for $x_1 y_1$ the following:

$$x_1 y_1 \overset{\textcircled{1}}{=} \sqrt{x_0^2 - 1}\sqrt{y_0^2 - 1} \overset{\textcircled{4}}{\geq} y_0^2 - 1 \overset{\textcircled{3}}{=} \frac{1}{1 - \frac{w_0^2}{\|\boldsymbol{w}'\|}} - 1 = \frac{w_0^2}{\|\boldsymbol{w}'\|^2 - w_0^2} \ .$$

By assumption we have $\boldsymbol{w} * \boldsymbol{w} = w_0^2 - \|\boldsymbol{w}'\|^2 = -1$, which gives for the denominator $-w_0^2 + \|\boldsymbol{w}'\|^2 = 1$. It remains to show that $w_0^2 \geq 1$.

For this last step, we want to show that mass concentrates on $w_0$ as the classifier is updated, ensuring $w_0 \geq 1$. By construction, we have initially $\boldsymbol{w} * \boldsymbol{w} = -1$. Wlog, assume that initially $w_0 \geq 1$. An initialization of this form can always be found, e.g., by setting $\boldsymbol{w} = (a, \sqrt{1 + a^2}, 0, \ldots, 0)$ for some $a \geq 0$. If the $i^{th}$ update is negative ($y^i x_0^i < 0$), then $|\boldsymbol{w}|_*$ will initially decrease, but the normalization step will scale away the effect on $w_0$. However, if the $i^{th}$ update is non-negative ($y^i x_0^i \geq 0$), it will increase $w_0$. Over time, the positive updates concentrate the mass on $w_0$. Since we initialized to $w_0 \geq 1$, the condition will always stay valid. With the arguments above, this implies $x_1 y_1 \geq 1$. Inserting the latter in the expression above, we get

$$d_H = 2\,\mathrm{asinh}\left(\frac{1}{2}\frac{d_E}{\sqrt{x_1 y_1}}\right) \leq 2\,\mathrm{asinh}\left(\frac{d_E}{2}\right)$$

$$\Rightarrow d_E \geq 2\sinh\left(\frac{d_H}{2}\right) \geq \sinh(d_H) \ .$$

$\square$

### D.2.2 Characterizing the margin

In $\mathbb{P}^2$ the decision hyperplane corresponding to $\hat{w} = Bw$ corresponds to a hypercircle $\mathcal{K}_w$. One can show, that its radius is given by $r_w = \sqrt{\frac{1-\lambda}{1+\lambda}}$ [6], by computing the hyperbolic distance between a point on the decision boundary and one of the hypercircle's ideal points. Further note, that the support vectors lie on hypercircles $\mathcal{K}_x$ and $\mathcal{K}_y$, which correspond to the set of points of hyperbolic distance $\epsilon$ (i.e., the margin) from the decision boundary. We again assume wlog that at least one support vector is not unique and let $x_1, x_2 \in \mathcal{K}_x$ and $y \in \mathcal{K}_y$ (see Fig. 5).

**Theorem D.4** (Thm. 5.2). $\epsilon' \approx \epsilon$.

*Proof.* Our proof consists of three steps:

Figure 5: Support vectors on hypercircles $\mathcal{K}_x$ and $\mathcal{K}_y$ with decision hypercircle $\mathcal{K}_w$.

Figure 6: Margin as distance between hypercircles $\mathcal{K}_x$ and $\mathcal{K}_y$.

**Step 1: Find Euclidean radii and centers of hypercircles.** The hypercircles $\mathcal{K}_x, \mathcal{K}_y$ correspond to arcs of Euclidean circles $\bar{\mathcal{K}}_x, \bar{\mathcal{K}}_y$ in the full plane that are related through circle inversion on the decision circle $\bar{\mathcal{K}}_w$ (i.e., the Euclidean circle corresponding to $\mathcal{K}_w$); see Fig. 5. We can construct a "mirror point" $y' \in \bar{\mathcal{K}}_x$ of $y$ by circle inversion on $\bar{\mathcal{K}}_w$. We have the following (Euclidean) distance relations: The circle inversion gives

$$\bar{d}(y', \bar{c}_w) \, \bar{d}(y, \bar{c}_w) = r_w^2 \,,$$

where $\bar{c}_w$ denotes the center of $\bar{\mathcal{K}}_w$. Furthermore, we have (see Fig. 7)

$$\bar{d}(y, \bar{c}_w) = \bar{d}(y', \bar{c}_w) + \bar{d}(y, y') \,.$$

Putting both together, we get an expression for the Euclidean distance of $y$ and $y'$:

$$\text{①} \quad \bar{d}(y, y') = \bar{d}(\bar{c}_w, y) - \frac{\bar{r}_w^2}{\bar{d}(\bar{c}_w, y)} \,.$$

Here, we have by construction $\bar{c}_w = (0, a, 0, \dots, 0)$ with a free parameter $a$. Wlog, assume $\bar{c}_w = (0, -1, 0, \dots, 0)$. Next, consider the triangle $\Delta(x_1, x_2, y)$. We can express its altitude $h$ in terms of the side length $\bar{d}(x_1, x_2) =: d_1$, $\bar{d}(x_1, y) =: d_2$ and $\bar{d}(x_2, y) =: d_3$ via Heron's formula:

$$h = \frac{2}{d_1} \sqrt{s(s - d_1)(s - d_2)(s - d_3)} \,,$$

where $s = \frac{1}{2}(d_1 + d_2 + d_3)$. Now, consider the triangle $\Delta(x_1, x_2, y')$. Due to the relation between $y$ and $y'$ in ①, its altitude $h_x$ is related to $h$ as

$$\text{②} \quad h_x = h - \bar{d}(y, y') \,.$$

Figure 7: Geometric construction for computing the center and radius of the hypercircle $\mathcal{K}_x$.

With the side length - altitude relations given in $\Delta(x_1, x_2, y)$ and ②, we can compute the length of the other sides $\bar{d}(x_1, y')$ and $\bar{d}(x_2, y')$ as follows (with Pythagoras theorem):

$$\bar{d}(x_1, y') = \left(h_x^2 + \bar{d}(x_1, y)^2 - h^2\right)^{1/2}$$
$$\bar{d}(x_2, y') = \left(h_x^2 + \bar{d}(x_2, y)^2 - h^2\right)^{1/2} \ .$$

With that, we can compute the radius of $\bar{\mathcal{K}}_x$ as follows: $\bar{\mathcal{K}}_x$ circumscribes $\Delta(x_1, x_2, y')$, therefore its radius $\bar{r}_x$ can be computed via Heron's formula as

$$\bar{r}_x = \frac{\bar{d}(x_1, y') + \bar{d}(x_2, y') + \bar{d}(x_1, x_2)}{4A}$$
$$A = \sqrt{s(s - \bar{d}(x_1, y'))(s - \bar{d}(x_2, y'))(s - \bar{d}(x_1, x_2))}$$

where $s = \frac{1}{2}(\bar{d}(x_1, y') + \bar{d}(x_2, y') + \bar{d}(x_1, x_2))$. With an analog construction, we can compute the radius $\bar{r}_y$ of $\bar{\mathcal{K}}_y$ as function of $\bar{d}(x_1', x_2')$, $\bar{d}(x_1', y)$ and $\bar{d}(x_2', y)$ via relations in the triangle $\Delta(x_1', x_2', y)$.

**Step 2: Express h-margin as distance between hypercircles.** As shown in Fig. 6, the margin is the hyperbolic distance from a point on $\mathcal{K}_x, \mathcal{K}_y$ to $\mathcal{K}_w$, corresponding to the length of a geodesic connecting the point with the closest point on $\mathcal{K}_w$. Let $v \in \mathcal{K}_x$ and $u \in \mathcal{K}_w$ the closest point on the decision circle. From the geometry of the Poincare half plane we know that there exists a Möbius transform $\theta \in \text{Möb}(\mathbb{P}^2)$ such that the images $\theta(u) = i\mu$ and $\theta(v) = i\nu$ of $u, v$ lie on the positive imaginary axis. Since the hyperbolic distance is invariant under Möbius transforms, we get

$$d(u, v) = d(\theta(u), \theta(v)) = d(i\mu, i\nu) = \left| \log \frac{\nu}{\mu} \right| \ .$$

Similarly, we can express the distance between between support vectors $x \in \mathcal{K}_x$ and $y \in \mathcal{K}$, which is twice the hyperbolic margin: Let $\theta(x) = i\mu_x$ and $\theta(y) = i\mu_y$, where $\mu_x, \mu_y$ are given by the intersection points of $\mathcal{K}_x, \mathcal{K}_y$ with the imaginary axis. Then

$$2\epsilon = d(x, y) = \left| \log \frac{\mu_y}{\mu_x} \right| \ .$$

We can express $\mu_x, \mu_y$ in terms of the centers and radii of $\mathcal{K}_x, \mathcal{K}_y$ as follows (Fig. 6)

$$\mu_x = \bar{c}_x^{(2)} + \bar{r}_x$$
$$\mu_y = \bar{c}_y^{(2)} + \bar{r}_y \ ,$$

where $c^{(2)}$ denotes the second coordinate of the point $c \in \mathbb{P}^m$. Putting everything together, we get the following expression for the margin:

$$\text{③} \quad \epsilon = \frac{1}{2} \left| \log \frac{\bar{c}_y^{(2)} + \bar{r}_y}{\bar{c}_x^{(2)} + \bar{r}_x} \right| .$$

**Step 3: Evaluate Distortion.** As discussed above (Prop. 5.1), the influence of distortion on the altitude $h$ in the triangle $\Delta(x_1, x_2, y)$ is given by the factor $\frac{1}{c_H}$.

$$\text{④} \quad h' = \frac{h}{c_H} .$$

$\bar{r}_x$ depends on pairwise distances between support vectors and $h$, which are distorted by a factor $\frac{1}{c_H}$ (by assumption on $\phi_H$ and ④). $\bar{r}_x$ depends further on $h_x$ which in turn depends on $\bar{d}(c_w, y)$. The latter depends on the Euclidean norm of the support vector $y$, i.e., $\|y\|$. With Ass. 4 the total multiplicative distortion is then at most of a factor $\frac{1}{c_H}$. We can derive an analogue result for $\bar{r}_y$. For the center $\bar{c}_x$ note the following:

$$\bar{c}_x^{(2)} = \frac{1}{2} \left[ (1 - \bar{r}_w^2)\tilde{x}_0 - (1 + \bar{r}_w^2)\tilde{x}_1 \right] ,$$

where $(\tilde{x}_0, \tilde{x}_1, 0, \ldots, 0) = \tilde{x} = (\pi_{BP} \circ (\pi_{LB} \circ B))$ and $\bar{r}_w = \sqrt{\frac{1-\lambda}{1+\lambda}}$. Rewriting

$$(1 - \bar{r}_w^2)\tilde{x}_0 = \frac{2\tilde{w}_0 \tilde{x}_0}{\tilde{w}_1 + \tilde{w}_0}$$
$$(1 + \bar{r}_w^2)\tilde{x}_1 = \frac{2\tilde{w}_1 \tilde{x}_1}{\tilde{w}_1 + \tilde{w}_0} ,$$

we get

$$\bar{c}_x^{(2)} = \frac{\tilde{w}^T \tilde{x}}{\tilde{w}_0 + \tilde{w}_1} .$$

Similarly, one can derive

$$\bar{c}_y^{(2)} = \frac{\tilde{w}^T \tilde{y}}{\tilde{w}_0 + \tilde{w}_1} ,$$

for $(\tilde{y}_0, \tilde{y}_1, 0, \ldots, 0) = \tilde{y} = (\pi_{BP} \circ (\pi_{LB} \circ B))$. Both are only affected by distortion of the form (2), i.e. the multiplicative distortion is given by a factor $\frac{1}{c_H}$. Inserting this into the margin expression (④) gives

$$\epsilon' = \frac{1}{2} \left| \log \frac{c_y' + r_y'}{c_x' + r_x'} \right| \gtrsim \frac{1}{2} \left| \log \frac{\frac{c_y}{c_H} + \frac{r_y}{c_H}}{c_H c_x + c_H r_x} \right| = \frac{1}{2} \left| \log \left( \frac{1}{c_H^2} \frac{c_y + r_y}{c_x + r_x} \right) \right|$$
$$= \frac{1}{2} \left| \underbrace{\log \frac{1}{c_H^2}}_{\approx 0} + \log \frac{c_y + r_y}{c_x + r_x} \right| \overset{\dagger}{\approx} \frac{1}{2} \left| \log \frac{c_y + r_y}{c_x + r_x} \right| = \epsilon ,$$

where (†) follows from $c_H = O(1 + \epsilon)$ with $\epsilon > 0$ small, by Thm. A.3. $\qquad \square$

## E  Adversarial perceptron

With the geometric tools introduced in Appendix D, we can now also proof Lemma C.4. We restate the result from the main text:

**Lemma E.1.** *(Adversarial perceptron, Lem. C.4) Let $\bar{w}$ be the max-margin classifier of $\mathcal{S}$ with margin $\gamma_H$. At each iteration of Alg. 2, $\bar{w}$ linearly separates $\mathcal{S} \cup \mathcal{S}'$ with margin at least $\frac{\gamma_H}{\cosh(\alpha)}$.*

*Proof.* In the following, we again use the shorthand $|\boldsymbol{u}| = \sqrt{\pm \boldsymbol{u} * \boldsymbol{u}}$, with "+", if $\boldsymbol{u}$ is space-like (i.e., $\boldsymbol{u} * \boldsymbol{u} > 0$) and "-", if $\boldsymbol{u}$ is time-like (i.e., $\boldsymbol{u} * \boldsymbol{u} < 0$). Since $\bar{\boldsymbol{w}}$ is "time-like" and $\boldsymbol{x}, \tilde{\boldsymbol{x}}$ space-like, we have

$$\text{①} \quad |\bar{\boldsymbol{w}} * \boldsymbol{x}| = |\bar{\boldsymbol{w}}|\,|\boldsymbol{x}|\,\cosh\angle(\bar{\boldsymbol{w}}, \boldsymbol{x})$$
$$|\bar{\boldsymbol{w}} * \tilde{\boldsymbol{x}}| = |\bar{\boldsymbol{w}}|\,|\tilde{\boldsymbol{x}}|\,\cosh\angle(\bar{\boldsymbol{w}}, \tilde{\boldsymbol{x}})\ .$$

To prove the statement, we first transform the problem from the Lorentz model $\mathbb{L}^d$ to the Poincare half plane $\mathbb{P}^2$ using the map $(\pi_{BP} \circ (\pi_{LB} \circ B))$. Then the adversarial margin is given by the Euclidean distance of the hypercircle $\mathcal{K}_{\tilde{x}}$ through $\tilde{\boldsymbol{x}}$ and the decision hypercircle $\mathcal{K}_w$. First, note that we can express this as the hyperbolic distance of the points $(0, \theta(\tilde{\boldsymbol{x}}))$ and $(0, r_w)$, where $\theta \in \mathrm{M\ddot{o}b}(\mathbb{P}^2)$ is a Möbius transform that maps $\tilde{\boldsymbol{x}}$ to the imaginary axis. Importantly, any such $\theta$ leaves the Minkowski product invariant. One can show [6] that

$$\theta(\tilde{\boldsymbol{x}}) = c_{\tilde{x}} + \sqrt{c_{\tilde{x}}^2 + r_w}$$

where $c_{\tilde{x}} = \frac{1}{2}\left((1 - r_w^2)\tilde{x}_0 - (1 + r_w^2)\tilde{x}_1\right)$ is the Euclidean center of $\mathcal{K}_{\tilde{x}}$. The hyperbolic distance is then given by

$$\text{②} \quad \left|\log\frac{\theta(\tilde{\boldsymbol{x}})}{r_w}\right| = \left|\log\left(\frac{c_{\tilde{x}}}{r_w} + \sqrt{\frac{c_{\tilde{x}}^2}{r_w^2} + 1}\right)\right| = \left|\mathrm{asinh}\left(\frac{c_{\tilde{x}}}{r_w}\right)\right|\ .$$

Note, that

$$\frac{c_{\tilde{x}}}{r_w} = \frac{1}{2}\left[\left(\frac{1}{r_w} - r_w\right)\tilde{x}_0 - \left(\frac{1}{r_w} + r_w\right)\tilde{x}_1\right]\ ,$$

where

$$\frac{1}{r_w} - r_w = \sqrt{\frac{1+\lambda}{1-\lambda}} - \sqrt{\frac{1-\lambda}{1+\lambda}} = \frac{2\lambda}{\sqrt{1-\lambda^2}} = \frac{2w_0}{\sqrt{w_1^2 - w_0^2}}$$

$$\frac{1}{r_w} + r_w = \frac{2}{\sqrt{1-\lambda^2}} = \frac{2w_1}{\sqrt{w_1^2 - w_0^2}}\ .$$

This gives

$$\text{③} \quad \left|\mathrm{asinh}\left(\frac{c_{\tilde{x}}}{r_w}\right)\right| = \left|\mathrm{asinh}\left(\frac{w_0\tilde{x}_0 - w_1\tilde{x}_0}{\sqrt{w_1^2 - w_0^2}}\right)\right| = \left|\mathrm{asinh}\left(\frac{\boldsymbol{w} * \tilde{\boldsymbol{x}}}{|\boldsymbol{w}|}\right)\right|\ .$$

Using ②, we can express the adversarial margin in terms of the margin and the distance between features and adversarial samples as follows:

$$\left|\mathrm{asinh}\left(\frac{c_{\tilde{x}}}{\boldsymbol{w}}\right)\right| = \left|\mathrm{asinh}\left(\frac{c_{\tilde{x}}}{c_x}\underbrace{\frac{c_x}{r_w}}_{\geq\sinh(\gamma_H)}\right)\right| \stackrel{\dagger}{\geq} \left|\mathrm{asinh}\left(\frac{c_{\tilde{x}}}{c_x}\sinh(\gamma_H)\right)\right|\ ,$$

where (†) follows from the assumption that $y(\boldsymbol{w} * \boldsymbol{x}) \geq \sinh(\gamma_H)$ (with margin $\gamma_H$). We further show above that we can express the Euclidean centers as

$$c_x = \frac{\boldsymbol{w} * \boldsymbol{x}}{w_0 + w_1}\ , \qquad c_{\tilde{x}} = \frac{\boldsymbol{w} * \tilde{\boldsymbol{x}}}{w_0 + w_1}\ .$$

Wlog, assume that $\boldsymbol{w} * \boldsymbol{x} > 0$; then $\boldsymbol{w} * \tilde{\boldsymbol{x}} < 0$ and therefore

$$c_x = \frac{|\boldsymbol{w} * \boldsymbol{x}|}{w_0 + w_1}\ , \qquad c_{\tilde{x}} = \frac{-|\boldsymbol{w} * \tilde{\boldsymbol{x}}|}{w_0 + w_1}\ .$$

Inserting ① above in ③, we get

$$\mathrm{asinh}\left(-\frac{|\boldsymbol{w} * \tilde{\boldsymbol{x}}|}{|\boldsymbol{w} * \boldsymbol{x}|}\sinh(\gamma_H)\right) \stackrel{\text{①}}{=} \mathrm{asinh}\left(-\frac{|\boldsymbol{w}|\,|\tilde{\boldsymbol{x}}|\,\cosh(\angle(\boldsymbol{w}, \tilde{\boldsymbol{x}}))}{|\boldsymbol{w}|\,|\boldsymbol{x}|\,\cosh(\angle(\boldsymbol{w}, \boldsymbol{x}))}\sinh(\gamma_H)\right)$$

$$= \mathrm{asinh}\left(-\frac{|\tilde{\boldsymbol{x}}|\,\cosh(\angle(\boldsymbol{w}, \tilde{\boldsymbol{x}}))}{|\boldsymbol{x}|\,\cosh(\angle(\boldsymbol{w}, \boldsymbol{x}))}\sinh(\gamma_H)\right)$$

$$\stackrel{\dagger}{\geq} \mathrm{asinh}\left(-\frac{|\tilde{\boldsymbol{x}}|}{|\boldsymbol{x}|}\sinh(\gamma_H)\right)\ ,$$

where (†) follows from $\boldsymbol{w}$ being a better classifier for $\boldsymbol{x}$ than for $\tilde{\boldsymbol{x}}$ by construction. Therefore, we have

$$\left| \operatorname{asinh}\left( -\frac{|\boldsymbol{w} * \tilde{\boldsymbol{x}}|}{|\boldsymbol{w} * \boldsymbol{x}|} \sinh(\gamma_H) \right) \right| \geq \operatorname{asinh}\left( \frac{|\tilde{\boldsymbol{x}}|}{|\boldsymbol{x}|} \sinh(\gamma_H) \right) \ .$$

Furthermore, note, that by construction we have $d_{\mathbb{L}}(\boldsymbol{x}, \tilde{\boldsymbol{x}}) \leq \alpha$ and therefore:

$$\operatorname{acosh}(\boldsymbol{x} * \tilde{\boldsymbol{x}}) \leq \alpha \ \Rightarrow \ \boldsymbol{x} * \tilde{\boldsymbol{x}} \leq \cosh(\alpha) \ .$$

Since $\boldsymbol{x}, \tilde{\boldsymbol{x}}$ are both space-like, we further have $|\boldsymbol{x}| \, |\tilde{\boldsymbol{x}}| \leq \boldsymbol{x} * \tilde{\boldsymbol{x}}$. In summary, this gives

$$④ \quad |\boldsymbol{x}| \leq \frac{\cosh(\alpha)}{|\tilde{\boldsymbol{x}}|} \ .$$

Inserting ④ above, we get

$$\operatorname{asinh}\left( \frac{|\tilde{\boldsymbol{x}}|}{|\boldsymbol{x}|} \sinh(\gamma_H) \right) \overset{④}{\geq} \operatorname{asinh}\left( \frac{|\tilde{\boldsymbol{x}}|^2}{\cosh(\alpha)} \sinh(\gamma_H) \right)$$
$$\overset{\ddagger}{=} \operatorname{asinh}\left( \frac{\sinh(\gamma_H)}{\cosh(\alpha)} \right) \ ,$$

where (‡) follows from $|\tilde{\boldsymbol{x}}|^2 = \tilde{\boldsymbol{x}} * \tilde{\boldsymbol{x}} = 1$, since $\tilde{\boldsymbol{x}} \in \mathbb{L}^m$. Finally, the claim follows from

$$\operatorname{asinh}\left( \frac{\sinh(\gamma_H)}{\cosh(\alpha)} \right) \geq \frac{\gamma_H}{\cosh(\alpha)} \ .$$

$\square$

# F  Additional Experimental Results

## F.1  Hyperbolic perceptron

To validate the hyperbolic perceptron algorithm, we performed two simple classification experiments. For the two-class data set (ImageNet n09246464 and n07831146), we observe that hyperbolic perceptron can successfully classify the points into the two groups, i.e., it achieves zero test error. In a second experiment, we try hyperbolic perceptron on a linearly non-separable dataset. The algorithm was still able to classify reasonably well.

## F.2  Adversarial Gradient decent

### F.2.1  Choice of loss function

Following the large body of work on large-margin learning in Euclidean space, we tested our approach with the classic hinge (Eq. C.2) and least squares losses (Eq. C.3). While both algorithms work well in practise (see § 6 and Section F.2.2), they do not fulfill Ass. 1 on the whole domain. Therefore, our theoretical guarantees are not valid for those loss functions.

We derive theoretical results for the hyperbolic logistic loss (Eq. C.8) instead, which fulfills Ass. 1. Unfortunately, the hyperparameter $R_\alpha$ is difficult to determine in practice. We therefore decided to omit validation experiments with the hyperbolic logistic loss.

For choosing an *adversarial budget* $\alpha$ in practice, note that Assumption 1(2) imposes a norm constraint on the adversarial examples, relative to the maximal norm of the training points. Given the constant $R_x$, one can estimate an upper bound on $\alpha$. In addition, an upper bound on $\alpha$ depends on how separable the data set is, i.e., the maximal possible margin. Within these constraints, the choice of $\alpha$ is guided by a trade-off between better robustness and longer training time.

### F.2.2  Adversarial GD via least squares loss

Using the same data set as described in § 6, we also try classification in hyperbolic space with adversarial examples using the least squares losses (Eq. C.3). We use the same procedure to find adversarial examples. The results are plotted in Figure 8 with similar conclusions.

Figure 8: Performance of Adversarial GD using smoothed square loss (Eq. C.3). **Left:** Loss $L(\boldsymbol{w})$ on the original data. **Middle:** $\alpha$-robust loss $L_\alpha(\boldsymbol{w})$. **Right:** Hyperbolic margin $\gamma_H$. We vary the adversarial budget $\alpha$ over $\{0, 0.25, 0.5, 0.75\}$. The case $\alpha = 0$ corresponds to the setup in [6].

### F.3 Dimension-distortion trade-off

Euclidean embeddings computed using implementation in Nickel and Kiela [22] by Facebook Research[2].

| $d$ | Euclidean | Hyperbolic |
|---|---|---|
| 4 | 0.54 | 0.51 |
| 8 | 0.53 | 1.00 |
| 16 | 0.68 | 1.00 |

Table 2: Classification performance (test error) in hyperbolic vs. Euclidean space of dimension $d$.

[Supplementary Material 2 · supplemental-only.pdf]

# A  Hyperbolic Space

Hyperbolic spaces are smooth Riemannian manifolds $\mathcal{M} = \mathbb{H}^d$ and as such locally Euclidean spaces. In the following we introduce basic notation for three popular models of hyperbolic spaces. For a comprehensive overview see Bridson and Haefliger [3].

## A.1  Models of hyperbolic spaces

Figure 4: Models of hyperbolic space: The Lorentz model $\mathbb{L}^d$, the Poincare ball $\mathbb{B}^d$, and the Poincare half-plane $\mathbb{P}^d$.

The Poincare ball defines a hyperbolic space within the Euclidean unit ball, i.e.

$$\mathbb{B}^d = \{\boldsymbol{x} \in \mathbb{R}^d :\ \|\boldsymbol{x}\| < 1\}$$

$$d_\mathbb{B}(\boldsymbol{x}, \boldsymbol{x}') = \operatorname{acosh}\left(1 + 2\frac{\|\boldsymbol{x} - \boldsymbol{x}'\|^2}{(1 - \|\boldsymbol{x}\|^2)(1 - \|\boldsymbol{x}'\|^2)}\right) \ .$$

Here, $\|\cdot\|$ is the usual Euclidean norm.

The closely related Poincare half-plane model is defined as

$$\mathbb{P}^2 = \{\boldsymbol{x} \in \mathbb{R}^2 :\ x_1 > 0\}$$

$$d_\mathbb{P}(\boldsymbol{x}, \boldsymbol{x}') = \operatorname{acosh}\left(1 + \frac{(x_0' - x_0)^2 + (x_1' - x_1)^2}{2x_1 x_1'}\right) \ .$$

Note that if $x_0 = x_0'$, the metric simplifies as

$$d_\mathbb{P}(\boldsymbol{x}, \boldsymbol{x}') = d_\mathbb{P}((x_0, x_1), (x_0, x_1')) = \left|\ln\frac{x_1'}{x_1}\right| \ .$$

The model can be generalized to higher dimensions with

$$\mathbb{P}^d = \{(x_0, \ldots, x_{d-1}) \in \mathbb{R}^d \mid x_{d-1} > 0\} \ ,$$

however, we will only use the two-dimensional model $\mathbb{P}^2$ here. We further define the hyperboloid as

$$\mathbb{L}^d = \{\boldsymbol{x} \in \mathbb{R}^{d+1} :\ \boldsymbol{x} * \boldsymbol{x} = 1\}$$

$$d_\mathbb{L}(\boldsymbol{x}, \boldsymbol{x}') = \operatorname{acosh}(\boldsymbol{x} * \boldsymbol{x}') \ ,$$

where $*$ denotes the Minkowski product $\boldsymbol{x} * \boldsymbol{x}' = x_0 x_0' - \sum_{i=1}^d x_i x_i'$.

**Remark A.1.** The Lorentz model

$$\mathbb{L}^d = \{x \in \mathbb{R}^{d+1} :\ \boldsymbol{x} * \boldsymbol{x} = 1\} \ .$$

is also called double-sheet model. We use this more general setting in sections 2-4. For simplicity, we restrict ourselves to the upper sheet

$$\mathbb{L}_+^d = \{\boldsymbol{x} \in \mathbb{R}^{d+1} :\ \boldsymbol{x} * \boldsymbol{x} = 1,\ x_0 > 0\} \ ,$$

in section 5. All constructions of mappings between the different models of hyperbolic space can be extended to the double-sheet $\mathbb{L}^d$.

## A.2 Equivalence of different models of hyperbolic spaces

The Poincare ball $\mathbb{B}^d$ and the Lorentz model $\mathbb{L}^d_+$ are equivalent models of hyperbolic space. A mapping is given by

$$\pi_{\mathrm{LB}} : \mathbb{L}^d_+ \to \mathbb{B}^d$$

$$\boldsymbol{x} = (x_0, \dots, x_d) \mapsto \left( \frac{x_1}{1 + x_0}, \dots, \frac{x_d}{1 + x_0} \right) .$$

We can further construct a mapping from $\mathbb{B}^d$ to $\mathbb{P}^d$ by inversion on a circle centered at $(-1, 0, \dots, 0)$:

$$\pi_{\mathrm{BP}} : \mathbb{B}^d \to \mathbb{P}^d$$

$$\boldsymbol{x} = (x_0, \dots, x_{d-1}) \mapsto \frac{(2x_1, \dots, 2x_{d-1}, 1 - \|\boldsymbol{x}\|^2)}{1 + 2x_0 + \|\boldsymbol{x}\|^2} .$$

## A.3 Embeddability

When analyzing the dimension-distortion trade-off, we make use of two key results on the embeddability (cf. §2.2) of trees into Euclidean and hyperbolic spaces. We state them below for reference.

**Theorem A.2** ([2]). *An $N$-point metric $\mathcal{X}$ (i.e., $|\mathcal{X}| = N$) embeds into Euclidean space $\mathbb{R}^{O(\log^2 N)}$ with the distortion $c_M = O(\log N)$.*

This bound in Theorem A.2 is tight for trees in the sense that embedding them in a Euclidean space (of any dimension) must incur the distortion $c_m = \Omega(\log N)$ [19].

**Theorem A.3** ([28]). *Tree metrics embed quasi-isometrically with $c_M = O(1 + \epsilon)$ into $\mathbb{H}^d$.*

## A.4 Spherical codes in hyperbolic space

Consider the unit sphere $\mathbb{S}^{d-1} \subseteq \mathbb{R}^d$. A *spherical code* is a subset of $\mathbb{S}^{d-1}$, such that any two distinct elements $\boldsymbol{x}, \boldsymbol{x}'$ are separated by at least an angle $\theta$, i.e. $\langle \boldsymbol{x}, \boldsymbol{x}' \rangle \leq \cos \theta$. We denote the size of the largest code as $A(d, \theta)$.

A similar construction of such "spherical caps" can be obtained in $\mathbb{H}^d$. Note that the induced geometry of these caps is spherical, hence they inherit a spherical geometric structure. This allows in particular the transfer of bounds on $A(d, \theta)$ to hyperbolic space [7]:

**Theorem A.4** (Chabauty, Shannon, Wyner (see, e.g., [29])). $A(d, \theta) \geq (1 + o(1)) \sqrt{2\pi d} \frac{\cos \theta}{\sin^{d-1} \theta}$.

# B Hyperbolic Perceptron

In this section we analyze the convergence and generalization properties of the hyperbolic perceptron (cf. Algorithm 1). Note that the update $\boldsymbol{v}_t \leftarrow \boldsymbol{w}_t + y_j \boldsymbol{x}_j$ in Algorithm 1 always leads to a valid hyperplane, i.e., $\mathbb{L}^d \cap \mathcal{H}_{\boldsymbol{v}_t} \neq \emptyset$, which happens iff $\boldsymbol{v}_t * \boldsymbol{v}_t < 0$. This can be verified as follows:

$$\boldsymbol{v}_t * \boldsymbol{v}_t = (\boldsymbol{w}_t + y_j \boldsymbol{x}_j) * (\boldsymbol{w}_t + y_j \boldsymbol{x}_j) = \underbrace{\boldsymbol{w}_t * \boldsymbol{w}_t}_{\overset{(i)}{\leq} -1} + 2 \underbrace{y_j (\boldsymbol{x}_j * \boldsymbol{w}_t)}_{\overset{(ii)}{<} 0} + \underbrace{y^2}_{= 1} \underbrace{(\boldsymbol{x}_j * \boldsymbol{x}_j)}_{\overset{(iii)}{=} 1} < 0 ,$$

where $(i)$ is a consequence of the normalization step in Algorithm 1 and $(iii)$ follows as $\boldsymbol{x} * \boldsymbol{x} = 1$, since $\boldsymbol{x} \in \mathbb{L}^d$. As for $(ii)$, note that we perform the update $\boldsymbol{v}_t \leftarrow \boldsymbol{w}_t + y_j \boldsymbol{x}_j$ only when $y_j \neq \mathrm{sign}(\boldsymbol{x}_j * \boldsymbol{w})$ (cf. Algorithm 1).

We now restate Theorem 3.1 and present a detailed proof of the result.

**Theorem B.1** (Convergence hyperbolic Perceptron in Algorithm 1 (Theorem 3.1)). *Assume that there is some $\bar{\boldsymbol{w}} \in \mathbb{R}^{d+1}$ with $\sqrt{-\bar{\boldsymbol{w}} * \bar{\boldsymbol{w}}} = 1$ and $\boldsymbol{w}_0 * \bar{\boldsymbol{w}} \leq 0$, and some $\gamma_H > 0$, such that $y_j (\bar{\boldsymbol{w}} * \boldsymbol{x}_j) \geq \sinh(\gamma_H)$ for $j = 1, \dots, |\mathcal{S}|$. Then, Algorithm 1 converges in $O\left( \frac{1}{\sinh(\gamma_H)} \right)$ steps and returns a solution with margin $\gamma_H$.*

*Proof.* Assume wlog $\boldsymbol{w}_0 = (0, 1, 0, \ldots, 0) \in \mathbb{R}^{d+1}$. Then $\boldsymbol{w}_0 * \boldsymbol{w}_0 = -1$, i.e., $\mathbb{L}^d \cap \mathcal{H}_{\boldsymbol{w}_0} \neq \emptyset$. Hence, $\boldsymbol{w}_0$ is a valid initialization. Furthermore, assume that the $t$th error is made at the $j$th sample, i.e. update $\boldsymbol{v}_t \leftarrow \boldsymbol{w}_t + y_j \boldsymbol{x}_j$. For $\boldsymbol{u} \in \mathbb{R}^{d+1}$, let $|\boldsymbol{u}| = \sqrt{-\boldsymbol{u} * \boldsymbol{u}}$.

Now let us consider two cases:

- **Case 1**. In this case, we assume that the normalization is not performed in $t$th step, i.e.,
$$\boldsymbol{w}_{t+1} = \boldsymbol{w}_t + y_j \boldsymbol{x}_j.$$
  Therefore,
$$\boldsymbol{w}_{t+1} * \bar{\boldsymbol{w}} = (\boldsymbol{w}_t + y_j \boldsymbol{x}_j) * \bar{\boldsymbol{w}} = \boldsymbol{w}_t * \bar{\boldsymbol{w}} + y_j(\boldsymbol{x}_j * \bar{\boldsymbol{w}}) \geq \boldsymbol{w}_t * \bar{\boldsymbol{w}} + \underbrace{\gamma_H'}_{:=\sinh(\gamma_H)} . \tag{B.1}$$

- **Case 2**. In this case, the normalization is performed in the $t$th step of Algorithm 1, i.e.,
$$\boldsymbol{w}_{t+1} = \frac{\boldsymbol{w}_t + y_j \boldsymbol{x}_j}{|\boldsymbol{w}_t + y_j \boldsymbol{x}_j|}.$$
  Thus,
$$\boldsymbol{w}_{t+1} * \bar{\boldsymbol{w}} = \frac{\boldsymbol{w}_t + y_j \boldsymbol{x}_j}{|\boldsymbol{w}_t + y_j \boldsymbol{x}_j|} * \bar{\boldsymbol{w}} \overset{(i)}{\geq} (\boldsymbol{w}_t + y_j \boldsymbol{x}_j) * \bar{\boldsymbol{w}}$$
$$\geq \boldsymbol{w}_t * \bar{\boldsymbol{w}} + \underbrace{y_j(\boldsymbol{x}_j * \bar{\boldsymbol{w}})}_{\geq \gamma_H'} \geq \boldsymbol{w}_t * \bar{\boldsymbol{w}} + \gamma_H' , \tag{B.2}$$

where $(i)$ follows as the normalization is performed only if $|\boldsymbol{w}_t + y_j \boldsymbol{x}_j| < 1$ and numerator is positive by induction.

By utilizing (B.1) and (B.2), we obtain the following telescoping sum
$$\sum_{k=0}^{T-1} (-\boldsymbol{w}_{k+1} + \boldsymbol{w}_k) * \bar{\boldsymbol{w}} \leq \sum_{k=0}^{T-1} -\gamma_H'$$
$$\Rightarrow -\boldsymbol{w}_T * \bar{\boldsymbol{w}} \leq -\boldsymbol{w}_0 * \bar{\boldsymbol{w}} - T\gamma_H' . \tag{B.3}$$

Recall that, for the Minkowski product, we have
$$\cosh(\angle(\boldsymbol{u}, \boldsymbol{u}')) = -\frac{\boldsymbol{u} * \boldsymbol{u}'}{\sqrt{-\boldsymbol{u} * \boldsymbol{u}} \sqrt{-\boldsymbol{u}' * \boldsymbol{u}'}} = \frac{-\boldsymbol{u} * \boldsymbol{u}'}{|\boldsymbol{u}| |\boldsymbol{u}'|} . \tag{B.4}$$

By utilizing (B.4) with $(\boldsymbol{u}, \boldsymbol{u}') = (\boldsymbol{w}_T, \bar{\boldsymbol{w}})$ and $(\boldsymbol{u}, \boldsymbol{u}') = (\boldsymbol{w}_0, \bar{\boldsymbol{w}})$ in (B.3), we obtain that
$$|\boldsymbol{w}_T||\bar{\boldsymbol{w}}| \cosh(\angle(\boldsymbol{w}_T, \bar{\boldsymbol{w}})) \leq |\boldsymbol{w}_0||\bar{\boldsymbol{w}}| \cosh(\angle(\boldsymbol{w}_0, \bar{\boldsymbol{w}})) - T\gamma_H'. \tag{B.5}$$

Since, we have $|\bar{\boldsymbol{w}}| = |\boldsymbol{w}_0| = 1$, it follows from (B.5) that
$$|\boldsymbol{w}_T| \cosh(\angle(\boldsymbol{w}_T, \bar{\boldsymbol{w}})) \leq \cosh(\angle(\boldsymbol{w}_0, \bar{\boldsymbol{w}})) - T\gamma_H'. \tag{B.6}$$

Further, using the facts that, due to normalization in Algorithm 1, $|\boldsymbol{w}_T| \geq 1$ and $\cosh(\cdot) \geq 1$, it follows from (B.6) that
$$1 \leq \cosh(\angle(\boldsymbol{w}_0, \bar{\boldsymbol{w}})) - T\gamma_H' \overset{(ii)}{\leq} C - T\gamma_H', \tag{B.7}$$

where $(ii)$ follows as $\angle(\boldsymbol{w}_i, \bar{\boldsymbol{w}}) < \pi$, since the orientation is fixed by the requirement that $\mathbb{L}^d \cap \mathcal{H}_{\boldsymbol{w}_i} \neq \emptyset$; as a result, we can find an upper bound $\cosh(\angle(\boldsymbol{w}_0, \bar{\boldsymbol{w}})) < \cosh(\pi) = C$. Now, it follows from (B.7) that
$$T \leq \frac{C-1}{\gamma_H'}, \tag{B.8}$$

which completes the proof of the convergence guarantee. The margin is given by
$$\text{margin}_{\mathcal{S}}(w) = \inf_{(x,y) \in \mathcal{S}} \text{asinh}\left(\frac{y(\boldsymbol{w} * \boldsymbol{x})}{\sqrt{-\boldsymbol{w} * \boldsymbol{w}}}\right) = \text{asinh}(\gamma_H') = \text{asinh}\left(\sinh(\gamma_H)\right) = \gamma_H ,$$

which implies that a margin of $\gamma_H$ is achieved in $O\left(\frac{1}{\sinh(\gamma_H)}\right)$ steps. $\qquad\square$

## C Adversarial Learning

### C.1 Loss functions

For training the classifier, we consider the margin losses that have the following form

$$l(\boldsymbol{x}, y; \boldsymbol{w}) = f(y \cdot (\boldsymbol{w} * \boldsymbol{x})), \tag{C.1}$$

where $f \colon \mathbb{R} \to \mathbb{R}_+$ is some convex, non-increasing function. Cho et al. [6] introduce the *hinge loss* in the hyperbolic setting which is defined by the (hyperbolic) hinge function $f(s) = \max\{0, \operatorname{asinh}(1) - \operatorname{asinh}(s)\}$, i.e.,

$$l(\boldsymbol{x}, y; \boldsymbol{w}) = \max\{0, \operatorname{asinh}(1) - \operatorname{asinh}(y(\boldsymbol{w} * \boldsymbol{x}))\} . \tag{C.2}$$

A significant shortcoming of this notion is its non-smoothness and non-convexity. Therefore, we additionally consider a smoothed *least squares loss*:

$$l(\boldsymbol{x}_i, y_i; \boldsymbol{w}) = \begin{cases} \frac{1}{2}\left(\operatorname{asinh}(1) - \operatorname{asinh}(y_i(\boldsymbol{w} * \boldsymbol{x}_i))\right)^2, & y_i(\boldsymbol{w} * \boldsymbol{x}_i) \le 1 \\ 0, & \text{else} \end{cases}, \tag{C.3}$$

We present experimental results for both losses.

The majority of the paper employs a hyperbolic version of the logistic loss to introduce the logistic regression problem in hyperbolic space. First, recall the logistic regression problem in the Euclidean setting. Given an input $\boldsymbol{x}$ and a linear classifier defined by $\boldsymbol{w}$, the prediction of the classifier is defined as

$$p(y|\boldsymbol{x}; \boldsymbol{w}) = 1/\left(1 + \exp(-y\langle\boldsymbol{x}, \boldsymbol{w}\rangle)\right) \tag{C.4}$$

Thus the log-loss takes the following form

$$\begin{aligned} l(\boldsymbol{x}, y; \boldsymbol{w}) = -\log p(y|\boldsymbol{x}; \boldsymbol{w}) &= \log\left(1 + \exp(-y\langle\boldsymbol{x}, \boldsymbol{w}\rangle)\right) \\ &= \log\left(1 + \exp(-y\|\boldsymbol{w}\|\langle\boldsymbol{x}, \bar{\boldsymbol{w}}\rangle)\right) \\ &= \log\left(1 + \exp(-y\operatorname{sgn}(\langle\boldsymbol{x}, \bar{\boldsymbol{w}}\rangle)\|\boldsymbol{w}\|d(\boldsymbol{x}, \partial H_{\bar{\boldsymbol{w}}}))\right) \end{aligned} \tag{C.5}$$

where $\bar{\boldsymbol{w}} = \boldsymbol{w}/\|\boldsymbol{w}\|$ and $d(\boldsymbol{x}, \partial H_{\bar{\boldsymbol{w}}})$ is the distance of $\boldsymbol{x}$ from the decision boundary $\partial H_{\bar{\boldsymbol{w}}} := \{\boldsymbol{z} \in \mathbb{R}^{d+1} : \langle\boldsymbol{z}, \bar{\boldsymbol{w}}\rangle = 0\}$. Note that $y\operatorname{sgn}(\langle\boldsymbol{x}, \bar{\boldsymbol{w}}\rangle)d(\boldsymbol{x}, \partial H_{\bar{\boldsymbol{w}}}))$ denotes the Euclidean margin of the $(\boldsymbol{x}, y)$ with respect to the decision boundary defined by $\bar{\boldsymbol{w}}$.

We can define a hyperbolic version of the logistic regression problem, where we replace the Euclidean margin with the hyperbolic margin with respect to the linear classifier $\boldsymbol{w}$. Recall that the hyperbolic margin has the following form (cf.. (2.2)):

$$y\operatorname{sgn}(\boldsymbol{x} * \boldsymbol{w})d(\boldsymbol{x}, \partial\mathcal{H}_{\boldsymbol{w}}) = y\operatorname{sgn}(\boldsymbol{x} * \boldsymbol{w})\left|\operatorname{asinh}\left(\frac{\boldsymbol{w} * \boldsymbol{x}}{\sqrt{-\boldsymbol{w} * \boldsymbol{w}}}\right)\right| = \operatorname{asinh}\left(\frac{y(\boldsymbol{w} * \boldsymbol{x})}{\sqrt{-\boldsymbol{w} * \boldsymbol{w}}}\right) \tag{C.6}$$

Therefore, by combining (C.5) and (C.6), the hyperbolic logistic regression problem with a linear classifier corresponds to minimizing the following loss:

$$l(\boldsymbol{x}, y; \boldsymbol{w}) = \ln\left(1 + \exp\left(-\operatorname{asinh}\left(\frac{y(\boldsymbol{w} * \boldsymbol{x})}{\sqrt{-\boldsymbol{w} * \boldsymbol{w}}}\right)\right)\right) . \tag{C.7}$$

Note that the hyperbolic logistic loss and the Euclidean logistic loss differ in the scaling factor $\|\boldsymbol{w}\|$. In order to ensure that the hyperbolic logistic loss satisfies Assumption 1, we introduce additional explicit scaling to obtain the following form of the loss.

$$l(\boldsymbol{x}, y; \boldsymbol{w}) = \ln\left(1 + \exp\left(-\operatorname{asinh}\left(\frac{y(\boldsymbol{w} * \boldsymbol{x})}{2R}\right)\right)\right) . \tag{C.8}$$

The following result verifies that the loss in (C.8) indeed satisfies Assumption 1.

**Lemma C.1.** *For valid inputs* $(\boldsymbol{x}, y; \boldsymbol{w})$, *the hyperbolic logistic loss in* (C.8) *fulfills Assumption 1.*

*Proof.* The robust loss (Eq. 4.1) is evaluated over inputs $(\boldsymbol{x}, y; \boldsymbol{w})$ only if $y(\boldsymbol{w} * \boldsymbol{x}) < 0$. A simple calculation shows, that Assumption 1.3 holds iff $\frac{|\boldsymbol{w} * \boldsymbol{x}|}{R_\alpha} \le 1$, where $R_\alpha$ is as given in Assumption 1.2.

As a results, we want to show $|\boldsymbol{w} * \boldsymbol{x}| \leq R_\alpha$ for all allowable inputs $(\boldsymbol{x}, y; \boldsymbol{w})$. Recall that

$$\boldsymbol{w} * \boldsymbol{x} = w_0 x_0 - \sum_{i=1}^{d} w_i x_i$$

$$\boldsymbol{w} \cdot \boldsymbol{x} = w_0 x_0 + \sum_{i=1}^{d} w_i x_i \ .$$

We consider the following cases:

1. $w_0 x_0 > 0$ *and* $\sum_{i=1}^{d} w_i x_i < 0$: $|\boldsymbol{w} * \boldsymbol{x}| \geq |\boldsymbol{w} \cdot \boldsymbol{x}|$;

2. $w_0 x_0 > 0$ *and* $\sum_{i=1}^{d} w_i x_i > 0$: $|\boldsymbol{w} \cdot \boldsymbol{x}| \geq |\boldsymbol{w} * \boldsymbol{x}|$;

3. $w_0 x_0 < 0$ *and* $\sum_{i=1}^{d} w_i x_i > 0$: $|\boldsymbol{w} * \boldsymbol{x}| \geq |\boldsymbol{w} \cdot \boldsymbol{x}|$;

4. $w_0 x_0 < 0$ *and* $\sum_{i=1}^{d} w_i x_i < 0$: $|\boldsymbol{w} \cdot \boldsymbol{x}| \geq |\boldsymbol{w} * \boldsymbol{x}|$.

In case (2) and (4) we have

$$|\boldsymbol{w} * \boldsymbol{x}| \leq |\boldsymbol{w} \cdot \boldsymbol{x}| \overset{(i)}{\leq} \|\boldsymbol{w}\| \, \|\boldsymbol{x}\| \overset{(ii)}{\leq} R_x R_w = R_\alpha \ ,$$

where $(i)$ follows from the Cauchy-Schwartz inequality and $(ii)$ follows from Assumption 1.2. In case (1) and (3), we have

$$|\boldsymbol{w} * \boldsymbol{x}| = |\boldsymbol{w} \cdot \hat{\boldsymbol{x}}| \overset{(i)}{\leq} \|\boldsymbol{w}\| \, \|\hat{\boldsymbol{x}}\| \overset{(ii)}{\leq} R_x R_w = R_\alpha \ , \tag{C.9}$$

where $\hat{\boldsymbol{x}} = (x_0, -x_1, \ldots, -x_n)$ and $(i)$ and $(ii)$ again follow from the Cauchy-Schwartz inequality, respectively. This completes the proof. $\square$

**Remark C.2.** A conceptually similar logistic loss is introduced in [17] for multinomial manifold. Max-margin learning with the above hyperbolic hinge loss was studied in [6].

## C.2  Generating adversarial examples (Certification problem)

Recall that to train a classifier with large margin, we enrich the training set with adversarial examples (cf. Algorithm 2). For a classifier $\boldsymbol{w}$, an adversarial example $\tilde{\boldsymbol{x}}$ for a given $(\boldsymbol{x}, y)$ is generated by perturbing $\boldsymbol{x}$ in the hyperbolic space up to the maximum allowed perturbation budget $\alpha$ such that

$$\tilde{\boldsymbol{x}} \leftarrow \underset{\substack{\boldsymbol{z} \in \mathbb{L}^d \\ d_{\mathbb{L}}(\boldsymbol{x}, \boldsymbol{z}) \leq \alpha}}{\arg\max} \ l(\boldsymbol{z}, y; \boldsymbol{w}) \ .$$

For the underlying loss function (cf. Section C.1), due to the monotonicity of $\mathrm{asinh}$, the above problem can be equivalently expressed as

$$\tilde{\boldsymbol{x}} \leftarrow \underset{\substack{\boldsymbol{z} \in \mathbb{L}^d \\ d_{\mathbb{L}}(\boldsymbol{x}, \boldsymbol{z}) \leq \alpha}}{\arg\min} \ y \cdot (\boldsymbol{w} * \boldsymbol{z}) = \underset{\substack{\boldsymbol{z} \in \mathbb{L}^d \\ d_{\mathbb{L}}(\boldsymbol{x}, \boldsymbol{z}) \leq \alpha}}{\arg\max} \ -\boldsymbol{w}' * \boldsymbol{z}$$

$$= \underset{\substack{\boldsymbol{z} \in \mathbb{L}^d \\ d_{\mathbb{L}}(\boldsymbol{x}, \boldsymbol{z}) \leq \alpha}}{\arg\max} \ -w_0' z_0 + \sum_i w_i' z_i \tag{C.10}$$

where $\boldsymbol{w}' = -y\boldsymbol{w}$. Since $\boldsymbol{w}', \boldsymbol{z} \in \mathbb{R}^{d+1}$, we can rewrite (C.10) as a constraint optimization task in the ambient Euclidean space:

$$\underset{\boldsymbol{z} \in \mathbb{R}^{d+1}}{\max} \ -w_0 z_0 + \sum_i w_i z_i \tag{C.11}$$

$$\text{s.t.} \quad d_{\mathbb{L}}(\boldsymbol{x}, \boldsymbol{z}) \leq \alpha$$

$$z_0^2 - \sum_{i=1}^{d} z_i^2 = 1 \ .$$

Assuming that we guess $z_0$ based on $x_0$, the constraint $z_0^2 - \sum_{i=1}^d z_i^2 = 1$ confines the solution space onto a $d$-dimensional sphere of radius $r = \sqrt{z_0^2 - 1}$, which also implies that $z_0 \geq 1$. On the other hand the constraint $d_{\mathbb{L}}(\boldsymbol{x}, \boldsymbol{z}) \leq \alpha$ is equivalent to

$$d_{\mathbb{L}}(\boldsymbol{x}, \boldsymbol{z}) = \operatorname{acosh}(\boldsymbol{x} * \boldsymbol{z}) = \operatorname{acosh}(x_0 z_0 - \sum_{i=1}^d x_i z_i) < \alpha \quad \text{or} \quad \sum_i -x_i z_i \leq \cosh(\alpha) - x_0 z_0 \ .$$

Thus, the problem in (C.11) reduces to the following linear program with a spherical constraint.

$$\text{(CERT)} \quad \max_{\boldsymbol{z}_{\backslash 0} \in \mathbb{R}^d} \ -w_0 z_0 + \sum_i w_i z_i \tag{C.12}$$

$$\text{s.t.} \quad \sum_{i=1}^d -x_i z_i \leq \cosh(\alpha) - x_0 z_0$$

$$\|\boldsymbol{z}_{\backslash 0}\|^2 = z_0^2 - 1 \ ,$$

where $\boldsymbol{z}_{\backslash 0} = (z_1, \ldots, z_d)$. We now present a proof of Theorem 4.1 which characterizes a solution of the program in (C.12). For the sake of readability, we first restate the result from the main text:

**Theorem C.3** (Theorem 4.1). *Given the input example $(\boldsymbol{x}, y)$, let $\boldsymbol{x}_{\backslash 0} = (x_1, \ldots, x_d)$. We can efficiently compute a solution to* (CERT) *or decide that no solution exists. If a solution exists, then based on a guess of $z_0$ a maximizing adversarial example has the form $\tilde{\boldsymbol{x}} = \left(z_0, \sqrt{z_0^2 - 1} \left(b\check{\boldsymbol{x}} + \sqrt{1 - b^2}\check{\boldsymbol{x}}^\perp\right)\right)$. Here, $b = \frac{(\cosh(\alpha) - x_0 z_0)}{(\|\boldsymbol{x}_{\backslash 0}\| \sqrt{z_0^2 - 1})}$ depends on the adversarial budget $\alpha$, and $\check{\boldsymbol{x}}_{\backslash 0}^\perp$ is a unit vector orthogonal to $\check{\boldsymbol{x}} = -\boldsymbol{x}_{\backslash 0}/\|\boldsymbol{x}_{\backslash 0}\|$ along $\boldsymbol{w}$.*

*Proof.* First, note that (CERT) can be rewritten as

$$\text{(CĚRT)} \quad \max \ \langle \check{\boldsymbol{w}}, \ \check{\boldsymbol{z}} \rangle$$
$$\text{s.t.} \ \langle \check{\boldsymbol{x}}, \ \check{\boldsymbol{z}} \rangle \leq b$$
$$\|\check{\boldsymbol{z}}\| = 1 \ ,$$

where $\check{\boldsymbol{w}} = \boldsymbol{w}_{\backslash 0}/\|\boldsymbol{w}_{\backslash 0}\|$, $\check{\boldsymbol{x}} = -\boldsymbol{x}_{\backslash 0}/\|\boldsymbol{x}_{\backslash 0}\|$, and $b = (\cosh(\alpha) - x_0 z_0)/(\|\boldsymbol{x}_{\backslash 0}\| \|\boldsymbol{z}_{\backslash 0}\|)$. We further set $\check{\boldsymbol{z}} = \boldsymbol{z}_{\backslash 0}/\|\boldsymbol{z}_{\backslash 0}\|$ so that the norm constraint confines the solution to the unit sphere to simplify the derivation. We can later rescale the solution to have the norm $\sqrt{z_0^2 - 1}$.

The solution of CĚRT lies on the cone $\langle \check{\boldsymbol{x}}, \ \check{\boldsymbol{z}} \rangle = b$. We decompose $\check{\boldsymbol{w}}$ along $\check{\boldsymbol{x}}$ and its orthogonal complement $\check{\boldsymbol{x}}^\perp$, i.e.

$$\check{\boldsymbol{w}} = \xi\check{\boldsymbol{x}} + \zeta\check{\boldsymbol{x}}^\perp \ .$$

with $\zeta \geq 0$ and $\|\check{\boldsymbol{x}}^\perp\| = 1$. Without loss of generality, such a decomposition always exists. Note that

$$\langle \check{\boldsymbol{w}}, \ \check{\boldsymbol{z}}^* \rangle = \xi\langle \check{\boldsymbol{x}}, \ \check{\boldsymbol{z}}^* \rangle + \zeta\langle \check{\boldsymbol{x}}^\perp, \ \check{\boldsymbol{z}}^* \rangle = \xi b + \zeta\langle \check{\boldsymbol{x}}^\perp, \ \check{\boldsymbol{z}}^* \rangle \ ,$$

where the second equality follows from $\langle \check{\boldsymbol{x}}, \ \check{\boldsymbol{z}}^* \rangle = b$. This implies that for the objective $\langle \check{\boldsymbol{w}}, \ \check{\boldsymbol{z}} \rangle$ to be maximized, $\check{\boldsymbol{z}}^*$ has to have all of its remaining mass along $\check{\boldsymbol{x}}^\perp$, i.e.,

$$\check{\boldsymbol{z}}^* = b\check{\boldsymbol{x}} + \sqrt{1 - b^2}\check{\boldsymbol{x}}^\perp.$$

After rescaling to satisfy the original norm constraint in CERT, the maximizing adversarial example (for a given $z_0$) is given as

$$\tilde{\boldsymbol{x}} = \left(z_0, \sqrt{z_0^2 - 1} \cdot \check{\boldsymbol{z}}^*\right) = \left(z_0, \sqrt{z_0^2 - 1} \left(b\check{\boldsymbol{x}} + \sqrt{1 - b^2}\check{\boldsymbol{x}}^\perp\right)\right).$$

$\square$

## C.3 Adversarial Perceptron

For the convergence analysis of the gradient-based update, we first need to analyze the convergence of the adversarial perceptron. We first state the following lemma that relates the adversarial margin to the max-margin classifier.

**Lemma C.4.** *Let $\bar{w}$ be the max-margin classifier of $\mathcal{S}$ with margin $\gamma_H$. At each iteration of Algorithm 2, $\bar{w}$ linearly separates $\mathcal{S} \cup \mathcal{S}'$ with margin at least $\frac{\gamma_H}{\cosh(\alpha)}$.*

**Remark C.5.** Note that this "adversarial Perceptron" corresponds to a gradient update of the form $\boldsymbol{w}_{t+1} \leftarrow \boldsymbol{w}_t + y\tilde{\boldsymbol{x}}$, which resembles the adversarial SGD.

*Proof.* The proof reduces the problem to Euclidean geometry in the Poincare half plane. We defer the proof until Section E, since the respective geometric tools are introduced only in Section D.2. $\quad\square$

With this result, we can show the following bound on the sample complexity of the adversarial perceptron:

**Theorem C.6.** *Assume that there is some $\bar{w} \in \mathbb{R}^{d+1}$ with $\sqrt{-\bar{w} * \bar{w}} = 1$ and $\boldsymbol{w}_0 * \bar{w} \leq 0$, and some $\gamma_H > 0$, such that $y_j(\bar{w} * \boldsymbol{x}_j) \geq \sinh(\gamma_H)$ for $j = 1, \dots, |\mathcal{S}|$. Then, adversarial perceptron (with adversarial budget $\alpha$) converges after $O\left(\frac{\cosh(\alpha)}{\sinh(\gamma_H)}\right)$ steps, at which it has margin of at least $\frac{\gamma_H}{\cosh(\alpha)}$.*

*Proof.* Without loss of generality, we initialize the classifier as $w_0 = (0, 1, 0, \dots, 0)$. Furthermore, assume that the $t$th error is made at the $j$th sample. For the ease of exposition, we assume that the normalization step is not performed at this update. (The case with normalization after the update can be handled as in the Proof of Theorem B.1.) Thus,

$$\boldsymbol{w}_{t+1} \leftarrow \boldsymbol{w}_t + y_j\tilde{\boldsymbol{x}}_i \; ,$$

which implies that

$$(\boldsymbol{w}_{t+1} - \boldsymbol{w}_t) * \bar{w} = (y_j\tilde{\boldsymbol{x}}_j) * \bar{w} = y_j\left(\tilde{\boldsymbol{x}}_j * \bar{w}\right) \geq \frac{\gamma_H'}{\cosh(\alpha)} \; ,$$

where $\gamma_H' = \sinh(\gamma_H)$ and the last inequality follows from Lemma C.4. By summing and telescoping, we obtain that

$$\sum_{k=0}^{t} (\boldsymbol{w}_{k+1} - \boldsymbol{w}_k) * \bar{w} \geq \sum_{k=0}^{t} \frac{\gamma_H'}{\cosh(\alpha)}$$

$$\Rightarrow \quad (\boldsymbol{w}_{t+1} - \boldsymbol{w}_0) * \bar{w} \geq \frac{t\gamma_H'}{\cosh(\alpha)} \; .$$

Now, by multiplying both sides by $-1$ and rewriting the Minkowski product gives us that

$$-\boldsymbol{w}_{t+1} * \bar{w} \leq -\boldsymbol{w}_0 * \bar{w} - \frac{t\gamma_H'}{\cosh(\alpha)}$$

$$\leq \underbrace{|\boldsymbol{w}_0|}_{=1} \underbrace{|\bar{w}|}_{=1} \underbrace{\cosh(\angle(\boldsymbol{w}_0, \bar{w}))}_{\leq \cosh(\pi) =: C} - \frac{t\gamma_H'}{\cosh(\alpha)}$$

$$\leq C - \frac{t\gamma_H'}{\cosh(\alpha)} \; . \tag{C.13}$$

Now, note that

$$1 \leq \cosh(\angle(\boldsymbol{w}_{t+1}, \bar{w})) \leq \frac{-\boldsymbol{w}_{t+1} * \bar{w}}{\underbrace{|\boldsymbol{w}_{t+1}|}_{\geq 1} \underbrace{|\bar{w}|}_{=1}} \leq -\boldsymbol{w}_{t+1} * \bar{w} \overset{(i)}{\leq} C - \frac{t\gamma_H'}{\cosh(\alpha)} \; ,$$

where $(i)$ utilizes (C.13). Now, solving for $t$ gives us that

$$t \leq (C - 1) \cdot \frac{\cosh(\alpha)}{\gamma_H'}.$$

Further, it follows from (2.2) that an adversarial hyperbolic margin of $\frac{\gamma_H}{\cosh(\alpha)}$ is then achieved after $O\left(\frac{\cosh(\alpha)}{\sinh(\gamma_H)}\right)$ steps. $\quad\square$

## C.4 Gradient-based update

Recall that, our objective in Algorithm 2 consists of an inner optimization (that computes the adversarial example) and an outer optimization (that updates the classifier). In particular, we consider

$$\min_{\boldsymbol{w}\in\mathbb{R}^{d+1}} L_{\mathrm{rob}}(\boldsymbol{w};\mathcal{S}) := \frac{1}{|\mathcal{S}|}\sum_{(x,y)\in\mathcal{S}} l_{\mathrm{rob}}(\boldsymbol{x},y;\boldsymbol{w}) \ ,$$

where the robust loss is given by

$$l_{\mathrm{rob}}(\boldsymbol{x},y;\boldsymbol{w}) := \max_{\boldsymbol{z}\in\mathbb{L}^d, d_{\mathbb{L}}(\boldsymbol{x},\boldsymbol{z})\leq\alpha} l(\boldsymbol{x},y;\boldsymbol{w}) = l(\tilde{\boldsymbol{x}},y;\boldsymbol{w}) \ ,$$

where $\tilde{\boldsymbol{x}} \in \operatorname{argmax}_{\boldsymbol{z}\in\mathbb{L}^d, d_{\mathbb{L}}(\boldsymbol{x},\boldsymbol{z})\leq\alpha} l(\boldsymbol{x},y;\boldsymbol{w})$.

Recall that, to compute the update, we need to compute gradients of the outer minimization problem, i.e., $\nabla_w \, l_{\mathrm{rob}}$ over $\mathcal{S}$. However, the function $l_{\mathrm{rob}}$ is itself a maximization problem (referred to as the inner maximization problem above). Therefore, we compute the gradient at the maximizer of the inner problem. Danskin's theorem ensures that this gives a valid decent direction. For the sake of completeness, we recall the Danskin's theorem here.

**Theorem C.7** ( Danskin [9], Bertsekas [1]). *Suppose $X$ is a non-empty compact topological space and $g : \mathbb{R}^d \times X \to \mathbb{R}$ is a continuous function such that $g(\cdot, \delta)$ is differentiable for every $\delta \in X$. Let $\delta^*_{\boldsymbol{w}} = \operatorname{argmax}_{\delta\in X} g(\boldsymbol{w}, \delta)$. Then, the function $\psi(\boldsymbol{w}) = \max_{\delta\in X} g(\boldsymbol{w}, \delta)$ is subdifferentiable and the subdifferential is given by*

$$\partial\psi(\boldsymbol{w}) = \operatorname{conv}\left(\{\nabla_{\boldsymbol{w}} \, g(\boldsymbol{w},\delta)|\ \delta\in\delta^*_{\boldsymbol{w}}\}\right) \ .$$

This approach has been previously used in Madry et al. [20] and Charles et al. [5]. Note that when we find an adversarial example in Algorithm 2, we can write it in a closed form (cf. Theorem C.3). In particular,

$$l_{\mathrm{rob}}(\boldsymbol{x},y;\boldsymbol{w}) = \max_{d_{\mathbb{L}}(\boldsymbol{x},\boldsymbol{z})\leq\alpha} l(\boldsymbol{z},y;\boldsymbol{w}) = l(\tilde{\boldsymbol{x}},y;\boldsymbol{w}) \quad \text{with } \tilde{\boldsymbol{x}} = \left(\tilde{x}_0, \sqrt{\tilde{x}_0^2 - 1}\left(b\,\breve{\boldsymbol{x}} + \sqrt{1-b^2}\tilde{\boldsymbol{x}}^\perp\right)\right) \ .$$

Note that

$$\nabla_{\boldsymbol{w}} \, l(\tilde{\boldsymbol{x}},y;\boldsymbol{w}) = f'(y(\boldsymbol{w}*\tilde{\boldsymbol{x}}))\cdot\nabla_{\boldsymbol{w}}\, y(\boldsymbol{w}*\tilde{\boldsymbol{x}}) \ = f'(y(\boldsymbol{w}*\tilde{\boldsymbol{x}}))\cdot y\widehat{(\tilde{\boldsymbol{x}})}^T \ ,$$

where we have used the fact that $\nabla_{\boldsymbol{w}} \, y(\boldsymbol{w}*\tilde{\boldsymbol{x}}) = y\widehat{(\tilde{\boldsymbol{x}})}^T = y(\tilde{x}_0, -\tilde{x}_1, \ldots, -\tilde{x}_n)^T$. From Danskin's theorem, we have $\nabla_{\boldsymbol{w}} l(\tilde{\boldsymbol{x}},y;\boldsymbol{w}) \in \partial \, l_{\mathrm{rob}}(\boldsymbol{x},y;\boldsymbol{w})$. This enables us to compute the decent direction and perform the update step with

$$\nabla \, L(\boldsymbol{w};\mathcal{S}') = \frac{1}{|\mathcal{S}'|}\sum_{(\tilde{\boldsymbol{x}},y)\in\mathcal{S}'} \nabla \, l(\tilde{\boldsymbol{x}},y;\boldsymbol{w}) \ \in \partial L_{\mathrm{rob}}(\boldsymbol{w};\mathcal{S}) \ ,$$

Furthermore, we have

$$\nabla^2_{\boldsymbol{w}} \, l(\tilde{\boldsymbol{x}},y;\boldsymbol{w}) = f''(y(\boldsymbol{w}*\tilde{\boldsymbol{x}}))\tilde{\boldsymbol{x}}\tilde{\boldsymbol{x}}^T \in \partial^2 l_{\mathrm{rob}}(\boldsymbol{x},y;\boldsymbol{w}) \ , \qquad (C.14)$$

which enable the computation of the Hessian of $L(\boldsymbol{w};\mathcal{S}')$.

The convergence results in this section build on hyperbolic analogues of comparable Euclidean results in [30, 14].

We first show a bound on the Hessian of the loss:

**Lemma C.8.**

$$\nabla^2 L(\boldsymbol{w}_t;\mathcal{S}'_t) \preceq \beta\sigma^2_{\max}\cdot I \ ,$$

*where $\sigma_{\max}$ is an upper bound on the maximum singular value of the data matrix $\frac{1}{|\mathcal{S}'_t|}\sum_{(\tilde{\boldsymbol{x}},y)\in\mathcal{S}'_t} \tilde{\boldsymbol{x}}\tilde{\boldsymbol{x}}^T$.*

*Proof.*

$$\nabla^2 L(\boldsymbol{w}_t;\mathcal{S}'_t) = \frac{1}{|\mathcal{S}'_t|}\sum_{(\tilde{\boldsymbol{x}},y)\in\mathcal{S}'_t} \nabla^2 l(\tilde{\boldsymbol{x}},y;\boldsymbol{w}_t) \stackrel{(i)}{=} \frac{1}{|\mathcal{S}'_t|}\sum_{(\tilde{\boldsymbol{x}},y)\in\mathcal{S}'_t} f''(y(\tilde{\boldsymbol{x}}*\boldsymbol{w}_t))\tilde{\boldsymbol{x}}\tilde{\boldsymbol{x}}^T$$

$$\stackrel{(ii)}{\preceq} \beta\cdot\frac{1}{|\mathcal{S}'_t|}\sum_{(\tilde{\boldsymbol{x}},y)\in\mathcal{S}'_t} \tilde{\boldsymbol{x}}\tilde{\boldsymbol{x}}^T \preceq \beta\sigma^2_{\max}\cdot I \ ,$$

where $(i)$ and $(ii)$ follow from (C.14) and the assumption that $f$ is $\beta$-smooth. $\qquad\square$

With the help of Lemma C.8, we can show the following result (a restatement of Theorem 4.4), which establishes that the gradient updates are guaranteed to converge to a large-margin classifier:

**Theorem C.9** (Theorem 4.3). *Let $\{w_t\}$ be the GD iterates*

$$w_{t+1} \leftarrow w_t - \frac{\eta}{|\mathcal{S}'_t|} \sum_{(\tilde{x},y)\in\mathcal{S}'_t} \nabla l(\tilde{x}, y; w)$$

$$w_{t+1} \leftarrow \frac{w_{t+1}}{\sqrt{-w_{t+1} * w_{t+1}}}$$

*with constant step size $\eta < \frac{2}{\beta\sigma_{\max}^2}$ and an initialization $w_0$ with $w_0 * w_0 < 0$. Then, we have $\lim_{t\to\infty} L(w_t; \mathcal{S} \cup \mathcal{S}'_t) = 0$.*

*Proof.* By Assumption 1.1 we can find a $\bar{w}$ that linearly separates $\mathcal{S}$. Then, we have

$$\langle \bar{w}, \nabla L(w; \mathcal{S}'_t)\rangle = \langle \bar{w}, \frac{1}{|\mathcal{S}'_t|} \sum_{(\tilde{x},y)\in\mathcal{S}'_t} f'(y(\tilde{x} * w_t))\, y\widehat{\tilde{x}}\rangle$$

$$= \underbrace{\left( \frac{1}{|\mathcal{S}'_t|} \sum_{(x,y)\in\mathcal{S}'_t} f'(y(\tilde{x} * w_t)) \right)}_{<0} \underbrace{y\langle \bar{w}, \widehat{\tilde{x}}\rangle}_{=y(\bar{w}*\tilde{x})<0} \,,$$

where the negativity of the first term follows from the assumptions on $f$ (cf. Assumption 1.3) and the upper bound on the second term from the separability assumption. This implies that $\langle \bar{w}, \nabla L(w; \mathcal{S}'_t)\rangle \neq 0$ for any finite $w$. Therefore, there are no finite critical points $w$ for which $\nabla L(w; \mathcal{S}'_t) = 0$. However, GD is guaranteed to converge to a critical point for smooth objectives with an appropriate step size. Therefore, $\|w_t\| \to \infty$ and $y(w_t * x) > 0 \ \forall \ (x, y) \in \mathcal{S} \cup \mathcal{S}'_t$ and large enough $t$. Then, we have $l(x, y; w_t) \to 0$, for all $(x, y) \in \mathcal{S} \cup \mathcal{S}'_t$. This further implies that $L(w_t; \mathcal{S} \cup \mathcal{S}'_t) = \frac{1}{|\mathcal{S} \cup \mathcal{S}'_t|} \sum_{(x,y)\in\mathcal{S}\cup\mathcal{S}'_t} l(x, y; w_t) \to 0$. $\qquad\square$

We further show that the enrichment of the training set with adversarial examples is critical for polynomial-time convergence: Without adversarial training, we can construct a simple max-margin problem, that cannot be solved in polynomial time.

**Theorem C.10** (Theorem 3.3). *Consider $\mathcal{S} = \{(e_1, 1), (-e_1, -1)\} \subset \mathbb{R}^{d+1} \times \{+1, -1\}$ and a typical initialization $w_0 = e_2 \in \mathbb{R}^{d+1}$ (with the standard basis vectors $e_1, e_2 \in \mathbb{R}^{d+1}$). Let $\{w_t\}_t$ is a sequence of classifiers generated by the GD updates (with fixed step size $\eta$)*

$$w_{t+1} \leftarrow w_t - \frac{\eta}{|\mathcal{S}|} \sum_{(x,y)\in\mathcal{S}} \nabla l(x, y; w)$$

$$w_{t+1} \leftarrow \frac{w_{t+1}}{\sqrt{-w_{t+1} * w_{t+1}}} \,.$$

*Then, the number of iterations needed to achieve margin $\gamma_H$ is $\Omega(\exp(\gamma_H))$.*

*Proof.* First, note that the initialization $w_0$ is valid as $w_0 * w_0 = -1 < 0$. The gradient of the loss can be computed as

$$\nabla l(x_i, y_i; w_t) = f'(y_i(x_i * w_t))y_i\widehat{x}_i$$

where

$$f'(s) = -\frac{\exp\left(-\operatorname{asinh}\left(\frac{s}{2R}\right)\right)}{R\sqrt{\frac{s^2}{4R^2} + 1}\left(\exp\left(-\operatorname{asinh}\left(\frac{s}{2R}\right)\right) + 1\right)}$$

is the derivative of the hyperbolic logistic regression loss (cf. (4.5)). Note that due to the structure of $\mathcal{S}$ and $w_0$, the GD update will produce the following iteration sequence

$$a_{t+1} = a_t - f'(a_t)$$

$$w_t = (a_t, \sqrt{a_t^2 + 1}, 0, \dots, 0) \,,$$

where the first coordinate is determined through the GD update and the second through normalization to ensure the validaty of the classifier, i.e., $\boldsymbol{w}_t * \boldsymbol{w}_t < 0$. In order to see this, note that $\boldsymbol{w}_t * \boldsymbol{w}_t = a_t^2 - (\sqrt{a_t^2 + 1})^2 = -1 < 0$. We now want to show that

$$a_t \leq \sinh(\ln(t+1)) \ .$$

For the induction, note that $a_0 = 0 = \ln(1) = \sinh(\ln(1))$. Assume, that $a_t \leq \sinh(\ln(t+1))$. We want to show

$$a_{t+1} \leq \sinh(\ln(t+2)) \ .$$

Note, that

$$a_{t+1} = \underbrace{a_t}_{\textcircled{1}} + \underbrace{\frac{\exp\left(-\operatorname{asinh}\left(\frac{a_t}{2R}\right)\right)}{R\sqrt{\frac{a_t^2}{4R^2}+1}\left(\exp\left(-\operatorname{asinh}\left(\frac{a_t}{2R}\right)\right)+1\right)}}_{\textcircled{2}} \ .$$

Since $\exp\left(-\operatorname{asinh}\left(-\frac{a_t}{2R}\right)\right) \leq \exp\left(-\operatorname{asinh}\left(-\frac{a_t}{2R}\right)\right) + 1$ and $R\sqrt{\frac{a_t^2}{4R^2}+1} \geq 1$, clearly $\textcircled{2}$ is bounded by 1. Inserting this above and replacing $\textcircled{1}$ with the induction assumption, we have

$$a_{t+1} \leq \sinh(\ln(t+1)) + 1 \ .$$

Note that, by definition, $\sinh(z) = \frac{1}{2}\left(e^x - e^{-x}\right)$. Thus,

$$\sinh(\ln(t+1)) = \frac{1}{2}\left(t+1 - (-(t+1))\right) = t+1 \ ,$$

which further implies that

$$a_{t+1} = \sinh(\ln(t+1)) + 1 \leq t+2 = \sinh(\ln(t+2)) \ . \tag{C.15}$$

This finishes the induction proof. Assuming a margin of at least $\gamma_H$, we have

$$\gamma_H \leq \operatorname{margin}_{\mathcal{S}}(\boldsymbol{w}_t) = \operatorname{asinh}\left(\frac{y(\boldsymbol{x} * \boldsymbol{w}_t)}{\sqrt{-\boldsymbol{w}_t * \boldsymbol{w}_t}}\right) \overset{(i)}{=} \operatorname{asinh}(a_{t+1}) \overset{(ii)}{\leq} \operatorname{asinh}(\sinh(\ln(t+2))) \leq \ln(t+2) \ ,$$

where $(i)$ follows $\boldsymbol{w}_t * \boldsymbol{w}_t = -1$ after normalization and $(ii)$ from the upper bound in (C.15). Now, by solving for $t$, we obtain that $t = \Omega(\exp(\gamma_H))$. $\qquad\square$

Next, we quantify the convergence rate of adversarial training with GD updates (cf. (4.4)). We start by presenting some auxiliary results.

**Lemma C.11** (Smoothness bound)**.** *Let* $\eta_t =: \eta < \frac{2\sinh^2(\gamma_H)}{\beta\sigma_{\max}^2 \cosh^2(\alpha)R_\alpha^2}$ *be the fixed step size and* $\boldsymbol{w}_0$ *a valid initialization, i.e.* $\boldsymbol{w}_0 * \boldsymbol{w}_0 < 0$. *Then, for the GD update (with fixed step size* $\eta_t =: \eta$)

$$\boldsymbol{w}_{t+1} \leftarrow \boldsymbol{w}_t - \eta_t \underbrace{\nabla L(\boldsymbol{w}_t; \mathcal{S}_t')}_{\in \partial L_{\mathrm{rob}}(\boldsymbol{w}_t; \mathcal{S}_t)}$$

$$\boldsymbol{w}_{t+1} \leftarrow \frac{\boldsymbol{w}_{t+1}}{\sqrt{-\boldsymbol{w}_{t+1} * \boldsymbol{w}_{t+1}}} \ .$$

*we have*

1. $L_{\mathrm{rob}}(\boldsymbol{w}_{t+1}; \mathcal{S}) \leq L_{\mathrm{rob}}(\boldsymbol{w}_t; \mathcal{S}) - \eta\left(\frac{\sinh(\gamma_H)^2}{\cosh^2(\alpha)R_\alpha^2} - \frac{\beta\sigma_{\max}^2\eta}{2}\right) \| \underbrace{\nabla L(\boldsymbol{w}_t; \mathcal{S}_t')}_{\in \partial L_{\mathrm{rob}}(\boldsymbol{w}_t; \mathcal{S}_t)} \|^2$;

2. $\sum_{k=0}^{\infty} \|\nabla L(\boldsymbol{w}_k; \mathcal{S}_k')\|^2 < \infty$; *as a result,* $\lim_{t \to \infty} \|\nabla L(\boldsymbol{w}_t); \mathcal{S}_t'\|^2 = 0$.

*Proof.* In Algorithm 2 with gradient update rule, we have

$$\begin{aligned}
\boldsymbol{w}_{t+1} &= \boldsymbol{w}_t - \eta\nabla L(\boldsymbol{w}_t; \mathcal{S}_t') \\
&= \boldsymbol{w}_t - \frac{\eta}{|\mathcal{S}_t'|} \sum_{(\tilde{\boldsymbol{x}}, y) \in \mathcal{S}_t'} l(\tilde{\boldsymbol{x}}, y; \boldsymbol{w}_t) \\
&= \boldsymbol{w}_t - \frac{\eta}{|\mathcal{S}_t'|} \sum_{(\tilde{\boldsymbol{x}}, y) \in \mathcal{S}_t'} f'(y(\tilde{\boldsymbol{x}} * \boldsymbol{w}_t))y\widehat{\tilde{\boldsymbol{x}}} \ .
\end{aligned}$$

Now, consider the inner product $\langle \boldsymbol{w}_{t+1}, \bar{\boldsymbol{w}} \rangle$, where $\bar{\boldsymbol{w}}$ is the optimal classifier. With out loss of generality, we assume $\|\bar{\boldsymbol{w}}\| = 1$.

$$\langle \boldsymbol{w}_{t+1}, \bar{\boldsymbol{w}} \rangle = \langle \boldsymbol{w}_t, \bar{\boldsymbol{w}} \rangle - \frac{\eta}{|\mathcal{S}'|} \sum_{(\tilde{\boldsymbol{x}},y) \in \mathcal{S}'} f'(y(\tilde{\boldsymbol{x}} * \boldsymbol{w}_t)) y \langle \widehat{\tilde{\boldsymbol{x}}}, \bar{\boldsymbol{w}} \rangle$$

$$\overset{(i)}{=} \langle \boldsymbol{w}_t, \bar{\boldsymbol{w}} \rangle - \frac{\eta}{|\mathcal{S}'_t|} \sum_{(\tilde{\boldsymbol{x}},y) \in \mathcal{S}'_t} f'(y(\tilde{\boldsymbol{x}} * \boldsymbol{w}_t)) y (\tilde{\boldsymbol{x}} * \bar{\boldsymbol{w}})$$

$$\overset{(ii)}{\geq} \langle \boldsymbol{w}_t, \bar{\boldsymbol{w}} \rangle - \frac{\eta \gamma'_H}{|\mathcal{S}'_t| \cosh(\alpha)} \sum_{(\tilde{\boldsymbol{x}},y) \in \mathcal{S}'_t} f'(y(\tilde{\boldsymbol{x}} * \boldsymbol{w}_t)) \,,$$

where $(i)$ and $(ii)$ follow from $\langle \widehat{\tilde{\boldsymbol{x}}}, \bar{\boldsymbol{w}} \rangle = \tilde{\boldsymbol{x}} * \bar{\boldsymbol{w}}$ and $y(\tilde{\boldsymbol{x}} * \bar{\boldsymbol{w}}) \geq \frac{\gamma'_H}{\cosh(\alpha)}$ (cf. Lemma C.4), respectively. We use the shorthand $\gamma'_H = \sinh(\gamma_H)$. With the linearity of the inner product, we get

$$\langle \boldsymbol{w}_{t+1} - \boldsymbol{w}_t, \bar{\boldsymbol{w}} \rangle \geq - \frac{\eta \gamma'_H}{|\mathcal{S}'_t| \cosh(\alpha)} \sum_{(\tilde{\boldsymbol{x}},y) \in \mathcal{S}'_t} f'(y(\tilde{\boldsymbol{x}} * \boldsymbol{w}_t)) \,.$$

Since $f'$ is negative (cf. Assumption 1.3), we can replace $-f'(y(\tilde{\boldsymbol{x}} * \boldsymbol{w}_t))$ with $|f'(y(\tilde{\boldsymbol{x}} * \boldsymbol{w}_t))|$ to get

$$\langle \boldsymbol{w}_t - \boldsymbol{w}_{t+1}, \bar{\boldsymbol{w}} \rangle \geq \frac{\eta \gamma'_H}{|\mathcal{S}'_t| \cosh(\alpha)} \sum_{(\tilde{\boldsymbol{x}},y) \in \mathcal{S}'_t} |f'(y(\tilde{\boldsymbol{x}} * \boldsymbol{w}_t))|$$

$$\overset{(i)}{=} \frac{\eta \gamma'_H}{\cosh(\alpha) R_\alpha} \|\nabla L(\boldsymbol{w}_t; \mathcal{S}'_t)\| \,, \tag{C.16}$$

where $(i)$ holds as follows: Recall, that $\|\nabla l(\tilde{\boldsymbol{x}}, y; \boldsymbol{w}_t)\| \leq |f'(y(\tilde{\boldsymbol{x}} * \boldsymbol{w}_t))| \|\widehat{\tilde{\boldsymbol{x}}}\|$. Thus,

$$\|\nabla L(\boldsymbol{w}_t; \mathcal{S}'_t)\| = \|\frac{1}{|\mathcal{S}'_t|} \sum_{(\tilde{\boldsymbol{x}},y) \in \mathcal{S}'_t} l(\tilde{\boldsymbol{x}}, y; \boldsymbol{w}_t)\| \leq \frac{1}{|\mathcal{S}'_t|} \sum_{(\tilde{\boldsymbol{x}},y) \in \mathcal{S}'_t} \|l(\tilde{\boldsymbol{x}}, y; \boldsymbol{w}_t)\|$$

$$\leq \frac{1}{|\mathcal{S}'_t|} \sum_{(\tilde{\boldsymbol{x}},y) \in \mathcal{S}'_t} |f'(y(\tilde{\boldsymbol{x}} * \boldsymbol{w}_t))| \|\widehat{\tilde{\boldsymbol{x}}}\| \leq \frac{R_\alpha}{|\mathcal{S}'_t|} \sum_{(\tilde{\boldsymbol{x}},y) \in \mathcal{S}'_t} |f'(y(\tilde{\boldsymbol{x}} * \boldsymbol{w}_t))| \,.$$

This implies that

$$\frac{1}{|\mathcal{S}'_t|} \sum_{(\tilde{\boldsymbol{x}},y) \in \mathcal{S}'_t} |f'(y(\tilde{\boldsymbol{x}} * \boldsymbol{w}_t))| = \frac{1}{R_\alpha} \|\nabla L(\boldsymbol{w}_t; \mathcal{S}'_t)\| \,.$$

Applying Cauchy-Schwarz to the left hand side of (C.16) gives us that

$$\|\boldsymbol{w}_t - \boldsymbol{w}_{t+1}\| \|\bar{\boldsymbol{w}}\| \geq \langle \boldsymbol{w}_t - \boldsymbol{w}_{t+1}, \bar{\boldsymbol{w}} \rangle \geq \frac{\eta \gamma'_H}{\cosh(\alpha) R_\alpha} \|\nabla L(\boldsymbol{w}_t; \mathcal{S}'_t)\| \,. \tag{C.17}$$

Now, using the fact that $\|\bar{\boldsymbol{w}}\| = 1$ in (C.17), we get

$$\|\boldsymbol{w}_{t+1} - \boldsymbol{w}_t\| \geq \frac{\eta \gamma'_H}{\cosh(\alpha) R_\alpha} \|\nabla L(\boldsymbol{w}_t; \mathcal{S}'_t)\| \,. \tag{C.18}$$

Now, consider the following Taylor approximation:

$$L_{\mathrm{rob}}(\boldsymbol{w}_{t+1}; \mathcal{S}) = L_{\mathrm{rob}}(\boldsymbol{w}_t; \mathcal{S}) + \langle \underbrace{\nabla L(\boldsymbol{w}_t; \mathcal{S}'_t)}_{\in \partial L_{\mathrm{rob}}(\boldsymbol{w}_t; \mathcal{S})}, \boldsymbol{w}_{t+1} - \boldsymbol{w}_t \rangle +$$

$$(\boldsymbol{w}_{t+1} - \boldsymbol{w}_t)^T \underbrace{\nabla^2 L(\boldsymbol{v}; \mathcal{S}'_t)}_{\in \partial^2 L_{\mathrm{rob}}(\boldsymbol{v}; \mathcal{S})} (\boldsymbol{w}_{t+1} - \boldsymbol{w}_t)/2, \tag{C.19}$$

where $\boldsymbol{v} \in \mathrm{conv}(\boldsymbol{w}_{t+1}, \boldsymbol{w}_t)$. By utilizing Lemma C.8 in (C.19), we get that

$$L_{\mathrm{rob}}(\boldsymbol{w}_{t+1}; \mathcal{S}) \leq L_{\mathrm{rob}}(\boldsymbol{w}_t; \mathcal{S}) + \langle \nabla L(\boldsymbol{w}_t; \mathcal{S}'_t), \boldsymbol{w}_{t+1} - \boldsymbol{w}_t \rangle + \frac{\beta \sigma_{\max}^2}{2} \|\boldsymbol{w}_{t+1} - \boldsymbol{w}_t\|^2 \,. \tag{C.20}$$

Recall the update rule

$$\boldsymbol{w}_{t+1} = \boldsymbol{w}_t - \eta \nabla L(\boldsymbol{w}_t; \mathcal{S}'_t) \tag{C.21}$$

$$\Rightarrow \boldsymbol{w}_{t+1} - \boldsymbol{w}_t = -\eta \nabla L(\boldsymbol{w}_t; \mathcal{S}'_t) . \tag{C.22}$$

Inserting this in (C.20), we get

$$L_{\text{rob}}(\boldsymbol{w}_{t+1}; \mathcal{S}) = L_{\text{rob}}(\boldsymbol{w}_t; \mathcal{S}) + \langle -\eta^{-1}(\boldsymbol{w}_{t+1} - \boldsymbol{w}_t), \boldsymbol{w}_{t+1} - \boldsymbol{w}_t \rangle + \frac{\beta\sigma_{\max}^2}{2}\|\boldsymbol{w}_{t+1} - \boldsymbol{w}_t\|^2$$

$$= L_{\text{rob}}(\boldsymbol{w}_t; \mathcal{S}) - \eta^{-1}\|\boldsymbol{w}_{t+1} - \boldsymbol{w}_t\|^2 + \frac{\beta\sigma_{\max}^2}{2}\|\boldsymbol{w}_{t+1} - \boldsymbol{w}_t\|^2 . \tag{C.23}$$

By combining (C.18) and (C.23), we obtain that

$$L_{\text{rob}}(\boldsymbol{w}_{t+1}; \mathcal{S}) \le L_{\text{rob}}(\boldsymbol{w}_t; \mathcal{S}) - \frac{\eta\gamma_H'^2}{\cosh^2(\alpha)R_\alpha^2}\|\nabla L(\boldsymbol{w}_t; \mathcal{S}'_t)\|^2 + \frac{\beta\sigma_{\max}^2}{2}\|\boldsymbol{w}_{t+1} - \boldsymbol{w}_t\|^2 . \tag{C.24}$$

Again, utilizing (C.21), it follows from (C.24) that

$$L_{\text{rob}}(\boldsymbol{w}_{t+1}; \mathcal{S}) \le L_{\text{rob}}(\boldsymbol{w}_t; \mathcal{S}) - \frac{\eta\gamma_H'^2}{\cosh^2(\alpha)R_\alpha^2}\|\nabla L(\boldsymbol{w}_t; \mathcal{S}'_t)\|^2 + \frac{\beta\sigma_{\max}^2\eta^2}{2}\|\nabla L(\boldsymbol{w}_t; \mathcal{S}'_t)\|^2$$

$$\tag{C.25}$$

$$= L_{\text{rob}}(\boldsymbol{w}_t; \mathcal{S}) - \eta\left(\frac{\gamma_H'^2}{\cosh^2(\alpha)R_\alpha^2} - \frac{\beta\sigma_{\max}^2\eta}{2}\right)\|\nabla L(\boldsymbol{w}_t; \mathcal{S}'_t)\|^2 . \tag{C.26}$$

This establishes the first claim of Lemma C.11. Now, we can rewrite (C.25) to obtain the following.

$$\frac{L_{\text{rob}}(\boldsymbol{w}_t; \mathcal{S}) - L_{\text{rob}}(\boldsymbol{w}_{t+1}; \mathcal{S})}{\eta\left(\frac{\gamma_H'^2}{\cosh^2(\alpha)R_\alpha^2} - \frac{\beta\sigma_{\max}^2\eta}{2}\right)} \ge \|\nabla L(\boldsymbol{w}_t; \mathcal{S}'_t)\|^2 .$$

Note that our assumption on the step size $\eta$ ensures that the denominator in (C.25) is $\ne 0$.

Next, summing and telescoping gives us that

$$\sum_{k=0}^{t}\|\nabla L(\boldsymbol{w}_k; \mathcal{S}'_k)\|^2 \le \sum_{k=0}^{t}\frac{L_{\text{rob}}(\boldsymbol{w}_k; \mathcal{S}) - L_{\text{rob}}(\boldsymbol{w}_{k+1}; \mathcal{S})}{\eta\left(\frac{\gamma_H'^2}{\cosh^2(\alpha)R_\alpha^2} - \frac{\beta\sigma_{\max}^2\eta}{2}\right)} = \frac{L_{\text{rob}}(\boldsymbol{w}_0; \mathcal{S}) - L_{\text{rob}}(\boldsymbol{w}_{t+1}; \mathcal{S})}{\eta\left(\frac{\gamma_H'^2}{\cosh^2(\alpha)R_\alpha^2} - \frac{\beta\sigma_{\max}^2\eta}{2}\right)} ,$$

where the right term is bounded, since $L_{\text{rob}}(\boldsymbol{w}_0; \mathcal{S}) < \infty$ and $0 \le L_{\text{rob}}(\boldsymbol{w}_{t+1}; \mathcal{S})$. This establishes the second claim of Lemma C.11 as

$$\sum_{k=0}^{\infty}\|\nabla L(\boldsymbol{w}_k; \mathcal{S}'_k)\|^2 < \infty \quad \Rightarrow \quad \lim_{t\to\infty}\|\nabla L(\boldsymbol{w}_t; \mathcal{S}'_t)\|^2 = 0 .$$

$$\square$$

**Lemma C.12.** *With the assumptions of Lemma C.11, Lemma C.11.1 implies for all $\boldsymbol{w} \in \mathbb{R}^{d+1}$*

$$2\sum_{k=0}^{t-1}\eta_k\big(L_{\text{rob}}(\boldsymbol{w}_k; \mathcal{S}) - L_{\text{rob}}(\boldsymbol{w}; \mathcal{S})\big) +$$

$$\sum_{k=0}^{t-1}\frac{\eta_k^2}{\bar{\eta}_k}\big(L_{\text{rob}}(\boldsymbol{w}_{k+1}; \mathcal{S}) - L_{\text{rob}}(\boldsymbol{w}_k; \mathcal{S})\big) \le \|\boldsymbol{w}_0 - \boldsymbol{w}\|^2 - \|\boldsymbol{w}_t - \boldsymbol{w}\|^2 ,$$

*where $\bar{\eta}_k = \eta_k\left(\frac{\gamma_H'^2}{\cosh(\alpha)^2 R_\alpha^2} - \frac{\beta\sigma_{\max}^2\eta_k}{4}\right)$.*

*Proof.* First, note that the GD update

$$\boldsymbol{w}_{t+1} = \boldsymbol{w}_t - \eta_t \underbrace{\nabla L(\boldsymbol{w}_t; \mathcal{S}'_t)}_{\in \partial L_{\text{rob}}(\boldsymbol{w}_t; \mathcal{S})}$$

implies that

$$\|\boldsymbol{w}_{t+1} - \boldsymbol{w}\|^2 = \|\boldsymbol{w}_t - \boldsymbol{w}\|^2 - 2\eta_t\langle\nabla L(\boldsymbol{w}_t;\mathcal{S}_t'), \boldsymbol{w}_t - \boldsymbol{w}\rangle + \eta_t^2\|\nabla L(\boldsymbol{w}_t;\mathcal{S}_t')\|^2 \qquad \text{(C.27)}$$

$$= \|\boldsymbol{w}_t - \boldsymbol{w}\|^2 + 2\eta_t\langle\nabla L(\boldsymbol{w}_t;\mathcal{S}_t'), \boldsymbol{w} - \boldsymbol{w}_t\rangle + \eta_t^2\|\nabla L(\boldsymbol{w}_t;\mathcal{S}_t')\|^2 . \qquad \text{(C.28)}$$

Note that the hyperbolic logistic regression loss $f(z)$ in (4.5) is convex for $z < 0$. As a consequence, $l_{\text{rob}}(\boldsymbol{x}, y; \boldsymbol{w})$ is convex for any adversarial example with $\text{sgn}(\boldsymbol{w}_t * \boldsymbol{x}) \neq \text{sgn}(\boldsymbol{w}_t * \tilde{\boldsymbol{x}})$. This implies that

$$l_{\text{rob}}(\boldsymbol{x}, y; \boldsymbol{w}) \geq l_{\text{rob}}(\boldsymbol{x}, y; \boldsymbol{w}_t) + \langle\partial l_{\text{rob}}(\boldsymbol{x}, y; \boldsymbol{w}_t), \boldsymbol{w} - \boldsymbol{w}_t\rangle ,$$

for any $\boldsymbol{w} \in \mathbb{R}^{d+1}$ and any pair $(\boldsymbol{x}, y)$ for which an adversarial example exists.

Since the sum of convex function is convex, we further have

$$L_{\text{rob}}(\boldsymbol{w};\mathcal{S}) \geq L_{\text{rob}}(\boldsymbol{w}_t;\mathcal{S}) + \langle\underbrace{\nabla L(\boldsymbol{w}_t;\mathcal{S}_t')}_{\in \partial L_{\text{rob}}(\boldsymbol{w}_t;\mathcal{S})}, \boldsymbol{w} - \boldsymbol{w}_t\rangle . \qquad \text{(C.29)}$$

By combining (C.27) and (C.29), we obtain that

$$\|\boldsymbol{w}_{t+1} - \boldsymbol{w}\|^2 \leq \|\boldsymbol{w}_t - \boldsymbol{w}\|^2 + 2\eta_t\big(L_{\text{rob}}(\boldsymbol{w};\mathcal{S}) - L_{\text{rob}}(\boldsymbol{w}_t;\mathcal{S})\big) + \eta_t^2\|\nabla L(\boldsymbol{w}_t;\mathcal{S}_t')\|^2$$

$$\overset{(i)}{\leq} \|\boldsymbol{w}_t - \boldsymbol{w}\|^2 + 2\eta_t\big(L_{\text{rob}}(\boldsymbol{w};\mathcal{S}) - L_{\text{rob}}(\boldsymbol{w}_t;\mathcal{S})\big) + \frac{\eta_t^2\big(L_{\text{rob}}(\boldsymbol{w}_t;\mathcal{S}) - L_{\text{rob}}(\boldsymbol{w}_{t+1};\mathcal{S})\big)}{\bar{\eta}_t},$$

where $(i)$ follows from the first claim in Lemma C.11 and $\bar{\eta}_t := \eta_t\left(\frac{\gamma_H'^2}{\cosh^2(\alpha)R_\alpha^2} - \frac{\beta\sigma_{\max}^2\eta_t}{4}\right)$.

Next, summing and telescoping gives us that

$$\sum_{k=0}^{t-1}\|\boldsymbol{w}_{k+1} - \boldsymbol{w}\|^2 - \|\boldsymbol{w}_k - \boldsymbol{w}\|^2 \leq \sum_{k=0}^{t-1}\Big[2\eta_k\big(L_{\text{rob}}(\boldsymbol{w};\mathcal{S}) - L_{\text{rob}}(\boldsymbol{w}_k;\mathcal{S})\big) +$$

$$\frac{\eta_k^2}{\bar{\eta}_k}\big(L_{\text{rob}}(\boldsymbol{w}_k;\mathcal{S}) - L_{\text{rob}}(\boldsymbol{w}_{k+1};\mathcal{S})\big)\Big]$$

or

$$\|\boldsymbol{w}_t - \boldsymbol{w}\|^2 - \|\boldsymbol{w}_0 - \boldsymbol{w}\|^2 \leq 2\sum_{k=0}^{t-1}\eta_k\big(L_{\text{rob}}(\boldsymbol{w};\mathcal{S}) - L_{\text{rob}}(\boldsymbol{w}_k;\mathcal{S})\big) +$$

$$\sum_{k=0}^{t-1}\frac{\eta_k^2}{\bar{\eta}_k}\big(L_{\text{rob}}(\boldsymbol{w}_k;\mathcal{S}) - L_{\text{rob}}(\boldsymbol{w}_{k+1};\mathcal{S})\big).$$

Now, multiplying both sides by $-1$ completes the proof as follow.

$$2\sum_{k=0}^{t-1}\eta_k\big(L_{\text{rob}}(\boldsymbol{w}_k;\mathcal{S}) - L_{\text{rob}}(\boldsymbol{w};\mathcal{S})\big) +$$

$$\sum_{k=0}^{t-1}\frac{\eta_k^2}{\bar{\eta}_k}\big(L_{\text{rob}}(\boldsymbol{w}_{k+1};\mathcal{S}) - L_{\text{rob}}(\boldsymbol{w}_k;\mathcal{S})\big) \leq \|\boldsymbol{w}_0 - \boldsymbol{w}\|^2 - \|\boldsymbol{w}_t - \boldsymbol{w}\|^2 .$$

$\square$

We are now in a position to present the desired convergence result.

**Theorem C.13** (Convergence GD update, Algorithm 2)**.** *For a fixed constant $c \in (0, 1)$, let the step size $\eta_t := \eta = c \cdot \frac{2\sinh^2(\gamma_H)}{\beta\sigma_{\max}^2\cosh^2(\alpha)R_\alpha^2}$ and $\mathcal{A}$ be the GD update as defined in (4.4). Then, the iterates $\{\boldsymbol{w}_t\}$ in Algorithm 2 satisfy*

$$L_{\text{rob}}(\boldsymbol{w}_t;\mathcal{S}) = O\left(\frac{\sinh^2(\ln(t))}{t} \cdot \left(\frac{\sinh(\gamma_H)}{\cosh(\alpha)}\right)^{-4}\right) .$$

*Proof.* Without loss of generality, assume that $\boldsymbol{w}_0 = (0, \boldsymbol{e}_i)$ where $\boldsymbol{e}_i \in \mathbb{R}^d$ is a standard basis vector whose $i$-th coordinate is 1. Note that this is a valid initialization, since $\boldsymbol{w}_0 * \boldsymbol{w}_0 < 0$; furthermore, we have $\|\boldsymbol{w}_0\| = 1$. Let $\boldsymbol{w}^* \in \mathbb{R}^{d+1}$ be a classifier that achieves the margin $\gamma_H$ on $\mathcal{S}$, i.e., $\forall (\boldsymbol{x}, y) \in \mathcal{S}$,

$$y(\boldsymbol{x} * \boldsymbol{w}^*) \geq \sinh(\gamma_H) \quad \Longleftrightarrow \quad \operatorname{asinh}\left( \frac{y(\boldsymbol{w}^* * \boldsymbol{x})}{\sqrt{-\boldsymbol{w}^* * \boldsymbol{w}^*}} \right) \geq \gamma_H.$$

Without loss of generality, assume that $\|\boldsymbol{w}^*\| = 1$. Let $\boldsymbol{u}_t := \frac{2R_\alpha \sinh(\ln(t)) \cosh(\alpha)}{\sinh(\gamma_H)} \boldsymbol{w}^*$; then $\|\boldsymbol{u}_t\| = \frac{2R_\alpha \sinh(\ln(t)) \cosh(\alpha)}{\sinh(\gamma_H)}$. We have

$$\begin{aligned}
L_{\mathrm{rob}}(\boldsymbol{u}_t; \mathcal{S}_t') &= \frac{1}{|\mathcal{S}_t'|} \sum_{(\boldsymbol{x}, y) \in \mathcal{S}_t'} l_{\mathrm{rob}}(\boldsymbol{x}, y; \boldsymbol{u}_t) = \frac{1}{|\mathcal{S}_t'|} \sum_{(\boldsymbol{x}, y) \in \mathcal{S}_t'} f(y(\tilde{\boldsymbol{x}} * \boldsymbol{u}_t)) \\
&\overset{(i)}{\leq} \frac{1}{|\mathcal{S}_t'|} \sum_{(\tilde{\boldsymbol{x}}, y) \in \mathcal{S}_t'} f\big(2R_\alpha \sinh(\ln(t))\big) = f\big(2R_\alpha \sinh(\ln(t))\big) \\
&\overset{(ii)}{\leq} \ln\left(1 + \exp\left(-\ln(t)\right)\right) \overset{(iii)}{\leq} \frac{1}{t} ,
\end{aligned} \tag{C.30}$$

where $(i)$ follows from

$$y(\tilde{\boldsymbol{x}} * \boldsymbol{u}_t) = \frac{2R_\alpha \sinh(\ln(t)) \cosh(\alpha)}{\sinh(\gamma_H)} \underbrace{y(\tilde{\boldsymbol{x}} * \boldsymbol{w}^*)}_{\geq \frac{\sinh(\gamma_H)}{\cosh(\alpha)}} \geq 2R_\alpha \sinh(\ln(t)) ,$$

(ii) from $\sqrt{-\boldsymbol{u}_t * \boldsymbol{u}_t} \geq 1$ and $(iii)$ follows from the fact that $\ln(1 + x) \leq x$.

Now, consider

$$\begin{aligned}
2\eta(t-1)\big(L_{\mathrm{rob}}(\boldsymbol{w}_t; \mathcal{S}) - L_{\mathrm{rob}}(\boldsymbol{u}_t; \mathcal{S})\big) &\overset{(i)}{=} 2\sum_{k=0}^{t-1} \eta_k\big(L_{\mathrm{rob}}(\boldsymbol{w}_t; \mathcal{S}) - L_{\mathrm{rob}}(\boldsymbol{u}_t; \mathcal{S})\big) \\
&= 2\sum_{k=0}^{t-1} \eta_k\big(L_{\mathrm{rob}}(\boldsymbol{w}_t; \mathcal{S}) - L_{\mathrm{rob}}(\boldsymbol{u}_t; \mathcal{S}) + L_{\mathrm{rob}}(\boldsymbol{w}_k; \mathcal{S}) - L_{\mathrm{rob}}(\boldsymbol{w}_k; \mathcal{S})\big) \\
&= 2\sum_{k=0}^{t-1} \eta_k\big(L_{\mathrm{rob}}(\boldsymbol{w}_k; \mathcal{S}) - L_{\mathrm{rob}}(\boldsymbol{u}_t; \mathcal{S})\big) + 2\sum_{k=0}^{t-1} \eta_k\big(L_{\mathrm{rob}}(\boldsymbol{w}_t; \mathcal{S}) - L_{\mathrm{rob}}(\boldsymbol{w}_k; \mathcal{S})\big) \\
&\overset{(ii)}{\leq} 2\sum_{k=0}^{t-1} \eta_k\big(L_{\mathrm{rob}}(\boldsymbol{w}_k; \mathcal{S}) - L_{\mathrm{rob}}(\boldsymbol{u}_t; \mathcal{S})\big) + \sum_{k=0}^{t-1} \eta_k\big(L_{\mathrm{rob}}(\boldsymbol{w}_{k+1}; \mathcal{S}) - L_{\mathrm{rob}}(\boldsymbol{w}_k; \mathcal{S})\big) \\
&\leq 2\sum_{k=0}^{t-1} \eta_k\big(L_{\mathrm{rob}}(\boldsymbol{w}_k; \mathcal{S}) - L_{\mathrm{rob}}(\boldsymbol{u}_t; \mathcal{S})\big) + \sum_{k=0}^{t-1} \frac{\eta_k^2}{\bar{\eta}_k}\big(L_{\mathrm{rob}}(\boldsymbol{w}_{k+1}; \mathcal{S}) - L_{\mathrm{rob}}(\boldsymbol{w}_k; \mathcal{S})\big) \\
&\overset{(iii)}{\leq} \|\boldsymbol{w}_0 - \boldsymbol{u}_t\|^2 - \|\boldsymbol{w}_t - \boldsymbol{u}_t\|^2 ,
\end{aligned}$$

where $(i)$ holds as we have a constant step-size, i.e., $\eta_k = \eta$ and $(ii)$ follows from the fact that

$$L_{\mathrm{rob}}(\boldsymbol{w}_t; \mathcal{S}) \leq L_{\mathrm{rob}}(\boldsymbol{w}_{k+1}; \mathcal{S}) \quad \text{for } 0 \leq k \leq t - 1.$$

The inequality in $(iii)$ follows from Lemma C.12 with $\boldsymbol{w} = \boldsymbol{u}_t$.

We can rewrite this as

$$\begin{aligned}
L_{\mathrm{rob}}(\boldsymbol{w}_t; \mathcal{S}) &\leq L_{\mathrm{rob}}(\boldsymbol{u}_t; \mathcal{S}) + \frac{\|\boldsymbol{w}_0 - \boldsymbol{u}_t\|^2 - \|\boldsymbol{w}_t - \boldsymbol{u}_t\|^2}{2\sum_{k=0}^{t-1} \eta_k} \\
&\overset{(i)}{\leq} \frac{1}{t} + \frac{\|\boldsymbol{w}_0 - \boldsymbol{u}_t\|^2}{2(t-1)\eta} \\
&\overset{(ii)}{\leq} \frac{1}{t} + \frac{2\|\boldsymbol{w}_0\|^2 + 2\|\boldsymbol{u}_t\|^2}{2(t-1)\eta} = \frac{1}{t} + \frac{\|\boldsymbol{w}_0\|^2 + \|\boldsymbol{u}_t\|^2}{(t-1)\eta}
\end{aligned}$$

where $(i)$ follows from (C.30) and $(ii)$ follows from $(a+b)^2 \leq 2a^2 + 2b^2$. Now, using the fact that $\|\boldsymbol{w}_0\| = 1$ and $\|\boldsymbol{u}_t\| = \frac{2R_\alpha \sinh(\ln(t))\cosh(\alpha)}{\sinh(\gamma_H)}$, we obtain that

$$L_{\mathrm{rob}}(\boldsymbol{w}_t; \mathcal{S}) \leq \frac{1}{t} + \frac{1 + 4R_\alpha^2 \left(\cosh(\alpha)/\sinh(\gamma_H)\right)^2 \cdot \sinh^2(\ln(t))}{(t-1)\eta} . \tag{C.31}$$

By substituting $\eta = c \cdot \frac{2\sinh^2(\gamma_H)}{\beta\sigma_{\max}^2 \cosh^2(\alpha)R_\alpha^2}$, we get

$$L_{\mathrm{rob}}(\boldsymbol{w}_t; \mathcal{S}) = O\left(\frac{\sinh^2(\ln(t))}{t} \cdot \left(\frac{\sinh(\gamma_H)}{\cosh(\alpha)}\right)^{-4}\right) .$$

$\square$

**Theorem C.14** (Iteration complexity). *Consider Algorithm 2 with $\eta_t := \eta = c \cdot \frac{2\sinh^2(\gamma_H)}{\beta\sigma_{\max}^2 \cosh^2(\alpha)R_\alpha^2}$ and $\mathcal{A}$ being the GD update. Then Algorithm 2 converges as $\Omega\left(\mathrm{poly}\left(\frac{\sinh(\gamma_H)}{\cosh(\alpha)}\right)\right)$.*

*Proof.* Let $\varrho = \frac{\ln(1+1/e)}{\ln(1+e)}$. We first argue that

$$L_{\mathrm{rob}}(\boldsymbol{w}_t; \mathcal{S}) \leq \varrho \cdot \ln\left(1 + \exp\left(-\frac{\gamma_H}{\cosh(\alpha)}\right)\right) \tag{C.32}$$

implies that $\boldsymbol{w}_t$ achieves margin $\gamma_H/\cosh(\alpha)$ on $\mathcal{S}$. To see this, note that

$$L_{\mathrm{rob}}(\boldsymbol{w}_t; \mathcal{S}) = \frac{1}{|\mathcal{S}|} \sum_{(\boldsymbol{x},y)\in\mathcal{S}} l_{\mathrm{rob}}(\boldsymbol{x}, y; \boldsymbol{w}_t)$$

$$= \underbrace{\max_{(\boldsymbol{x},y)\in\mathcal{S}} l_{\mathrm{rob}}(\boldsymbol{x}, y; \boldsymbol{w}_t)}_{:=l_{\mathrm{rob}}^{\max}(\mathcal{S})} \cdot \frac{1}{|\mathcal{S}|} \sum_{(\boldsymbol{x},y)\in\mathcal{S}} \frac{l_{\mathrm{rob}}(\boldsymbol{x}, y; \boldsymbol{w}_t)}{l_{\mathrm{rob}}^{\max}(\mathcal{S})}$$

$$\geq l_{\mathrm{rob}}^{\max} \cdot \frac{1}{|\mathcal{S}|} \sum_{(\boldsymbol{x},y)\in\mathcal{S}} \varrho = \varrho \cdot l_{\mathrm{rob}}^{\max}. \tag{C.33}$$

The last inequality in (C.33) holds as, for each $(\boldsymbol{x}, y) \in \mathcal{S}$, we have

$$\ln(1 + 1/e) \overset{(i)}{\leq} \underbrace{\ln\left(1 + \exp\left(-\mathrm{asinh}\left(\frac{y(\tilde{\boldsymbol{x}} * \boldsymbol{w}_t)}{2R_\alpha}\right)\right)\right)}_{=l_{\mathrm{rob}}(\boldsymbol{x},y;\boldsymbol{w}_t)} \leq \max_{(\boldsymbol{x},y)\in\mathcal{S}} l_{\mathrm{rob}}(\boldsymbol{x}, y; \boldsymbol{w}_t) \overset{(ii)}{\leq} \ln(1 + e),$$

where $(i)$ and $(ii)$ follows from Assumption 1.2. Thus, for each $(\boldsymbol{x}, y) \in \mathcal{S}$, we have

$$\frac{l_{\mathrm{rob}}(\boldsymbol{x}, y; \boldsymbol{w}_t)}{l_{\mathrm{rob}}^{\max}} \geq \frac{\ln(1 + 1/e)}{\ln(1 + e)} = \varrho. \tag{C.34}$$

Now, by combining (C.32) and (C.33), we obtain that

$$l_{\mathrm{rob}}(\boldsymbol{x}, y; \boldsymbol{w}_t) \leq \ln\left(1 + \exp\left(-\frac{\gamma_H}{\cosh(\alpha)}\right)\right)$$

for any $(\boldsymbol{x}, y) \in \mathcal{S}$. Equivalently, for each $(\boldsymbol{x}, y) \in \mathcal{S}$,

$$l_{\mathrm{rob}}(\boldsymbol{x}, y; \boldsymbol{w}_t) = \ln\left(1 + \exp\left(-\mathrm{asinh}\left(\frac{y(\tilde{\boldsymbol{x}} * \boldsymbol{w}_t)}{2R_\alpha}\right)\right)\right)$$

$$\leq \ln\left(1 + \exp\left(-\frac{\gamma_H}{\cosh(\alpha)}\right)\right) . \tag{C.35}$$

Thus, for each $(\boldsymbol{x}, y) \in \mathcal{S}$, we have

$$\mathrm{asinh}\left(\frac{y(\tilde{\boldsymbol{x}} * \boldsymbol{w}_t)}{\sqrt{-\boldsymbol{w}_t * \boldsymbol{w}_t}}\right) \overset{(i)}{\geq} \mathrm{asinh}\left(\frac{y(\tilde{\boldsymbol{x}} * \boldsymbol{w}_t)}{2R_\alpha}\right) \overset{(ii)}{\geq} \frac{\gamma_H}{\cosh(\alpha)} .$$

where (i) from the definition of $R_\alpha$ (cf. Assumption 1) and (ii) from Eq. C.35. Thus, $\boldsymbol{w}_t$ achieves margin $\gamma_H/\cosh(\alpha)$ on $\mathcal{S}$.

Next, introduce the following constant:

$$C_q := \inf\{t \geq 2 : \ 2 + \ln(t)^2 \leq (t-1)t^{-1/q}\} \ .$$

With this, for $t \geq C_q$, we can rewrite the bound in (C.31) as follows:

$$L_{\mathrm{rob}}(\boldsymbol{w}_t; \mathcal{S}) \leq \underbrace{\frac{1}{t}}_{\leq \frac{1}{(t-1)\eta}} + \frac{1 + \sinh(\ln(t))^2 \left(\frac{\sinh(\gamma_H)}{\cosh(\alpha)}\right)^{-2}}{(t-1)\eta} \leq \frac{2 + \sinh(\ln(t))^2 \left(\frac{\sinh(\gamma_H)}{\cosh(\alpha)}\right)^{-2}}{(t-1)\eta}$$

$$\leq \frac{2 + \ln(t)^2 \left(\frac{\sinh(\gamma_H)}{\cosh(\alpha)}\right)^{-2}}{(t-1)\eta} \leq \frac{(t-1)t^{-1/q}}{\eta(t-1)} \left(\frac{\sinh(\gamma_H)}{\cosh(\alpha)}\right)^{-2} \leq \frac{t^{-1/q}}{\eta \left(\frac{\sinh(\gamma_H)}{\cosh(\alpha)}\right)^2} \ .$$

Solving for $t$ and plugging in the above bound on $L_{\mathrm{rob}}$ for which $\boldsymbol{w}_t$ achieves the desire margin, as well as $\eta = c \cdot \frac{2\sinh^2(\gamma_H)}{\beta\sigma_{\max}^2 \cosh^2(\alpha)R_\alpha^2}$, we get

$$t = \max\{C_q, \Omega\left(\left((\sinh(\gamma_H)^4/\cosh(\alpha)^4)\right)^{-q}\right)\} \ ,$$

from which the claim follows directly. $\qquad\square$

### C.5 Algorithm 2 with an ERM update

Consider the unit sphere $\mathbb{S}^{d-1} \subseteq \mathbb{R}^d$. A spherical code with minimum separation $\theta$ is a subset of $\mathbb{S}^{d-1}$, such that any two distinct elements $\boldsymbol{u}, \boldsymbol{u}'$ in the subset are separated by at least an angle $\theta$, i.e. $\langle \boldsymbol{u}, \boldsymbol{u}'\rangle \leq \cos\theta$. We denote the size of the largest such code as $A(d, \theta)$. A similar construction can be made in hyperbolic space, which allows the transfer of bounds on $A(d, \theta)$ to hyperbolic space [7].

The following lemma shows that a spherical code with a suitable minimum separation $\theta$ enables a simple pathological training set such that Algorithm 2 along with an ERM update rule cannot produce a classifier with a desired margin in a small number of iteration. In particular, the lemma shows that the number of iterations required to find the desire margin is lower-bounded by the size of the underlying spherical code.

**Lemma C.15.** *Consider* $\mathcal{S} = \{(\boldsymbol{x}_1, y_1) = ((1, 0, \dots, 0), 1), (\boldsymbol{x}_2, y_2) = ((-1, 0, \dots, 0), -1)\}$, *where* $\boldsymbol{x}_1, \boldsymbol{x}_2 \in \mathbb{L}^d$ *and* $y_1, y_2$ *the corresponding labels. For any* $\epsilon < \alpha$, *there is an admissible sequence of classifiers* $\{\boldsymbol{w}_t\}_{1 \leq t \leq T}$, *with*

$$T = A\left(d, \arccos\left(\rho \cdot \frac{\sinh(\epsilon)\cosh(\alpha)}{\sqrt{\cosh^2(\alpha) - 1}\sqrt{1 + \sinh^2(\epsilon)}}\right)\right)$$

*Proof.* First, note that $\boldsymbol{x}_1 * \boldsymbol{x}_1 = \boldsymbol{x}_2 * \boldsymbol{x}_2 = 1$, i.e., $\boldsymbol{x}_1, \boldsymbol{x}_2 \in \mathbb{L}^d$ as desired. Let $\epsilon' = \sinh(\epsilon)$ and $\boldsymbol{e}_i \in \mathbb{R}^{d+1}$ denotes the standard basis vector that has its $i$-th coordinate equal to 1. Now, consider classifiers of the form

$$\boldsymbol{w}_t = \left(\frac{\epsilon'}{\sqrt{1+\epsilon'^2}} \boldsymbol{v}_t\right) \quad \text{where} \quad \boldsymbol{v}_t \in \mathcal{C}\left(d, \arccos\left(\rho \cdot \frac{\epsilon'\sqrt{1+\delta^2}}{\delta\sqrt{1+\epsilon'^2}}\right)\right) \quad \forall\, 1 \leq t \leq T \ , \quad \text{(C.36)}$$

where $\rho < 1$; and $\mathcal{C}\left(d, \arccos\left(\rho \cdot \frac{\epsilon'\sqrt{1+\delta^2}}{\delta\sqrt{1+\epsilon'^2}}\right)\right)$ be the spherical code with the minimum separation $\theta = \arccos\left(\rho \cdot \frac{\epsilon'\sqrt{1+\delta^2}}{\delta\sqrt{1+\epsilon'^2}}\right)$ and size $A(d, \theta)$. Since $\boldsymbol{w}_t * \boldsymbol{w}_t = (\epsilon')^2 - 1 - (\epsilon')^2 = -1$, we have $\boldsymbol{w}_t * \boldsymbol{w}_t < 0$ for all $t$. This guarantees that the intersections of the decision boundaries defined by $\{\boldsymbol{w}_t\}$ and $\mathbb{L}^d$ are not empty. Moreover, $\{\boldsymbol{w}_t\}_t$ is an admissible sequence of classifiers with margin $\leq \epsilon$. To see this, note that, for $t = 1, \dots, T$,

$$\boldsymbol{w}_t * \boldsymbol{x}_1 = \epsilon' > 0$$
$$\boldsymbol{w}_t * \boldsymbol{x}_2 = -\epsilon' < 0 \ ,$$

i.e., $\{\boldsymbol{w}_t\}$ correctly classifies $\mathcal{S}$. Furthermore, with $-\boldsymbol{w}_t * \boldsymbol{w}_t = 1$, we have

$$\operatorname{asinh}\left(\frac{y_1(\boldsymbol{w}_t * \boldsymbol{x}_1)}{\sqrt{-\boldsymbol{w}_t * \boldsymbol{w}_t}}\right) = \operatorname{asinh}\left(\frac{y_2(\boldsymbol{w}_t * \boldsymbol{x}_2)}{\sqrt{-\boldsymbol{w}_t * \boldsymbol{w}_t}}\right) = \epsilon\,,$$

which gives $\operatorname{margin}_{\mathcal{S}}(\boldsymbol{w}_t) = \epsilon$.

Now we perturb $\boldsymbol{x}_1, \boldsymbol{x}_2$ on $\mathbb{L}^d$ such that the magnitude of the perturbation is at most $\alpha$, i.e., we want to find $\tilde{\boldsymbol{x}}_1, \tilde{\boldsymbol{x}}_2 \in \mathbb{L}^d$ such that both $d_{\mathbb{L}}(\boldsymbol{x}_1, \tilde{\boldsymbol{x}}_1)$ and $d_{\mathbb{L}}(\boldsymbol{x}_2, \tilde{\boldsymbol{x}}_2)$ are at most $\alpha$. For $1 \le t \le T$, consider adversarial examples of the form

$$\tilde{\boldsymbol{x}}_{1t} = \begin{pmatrix} \sqrt{1+\delta^2} \\ \delta \boldsymbol{v}_t \end{pmatrix} \quad \text{and} \quad \tilde{\boldsymbol{x}}_{2t} = -\begin{pmatrix} \sqrt{1+\delta^2} \\ \delta \boldsymbol{v}_t \end{pmatrix}.$$

Note that $\tilde{\boldsymbol{x}}_{1t}, \tilde{\boldsymbol{x}}_{2t} \in \mathbb{L}^d$ as $\tilde{\boldsymbol{x}}_{1i} * \tilde{\boldsymbol{x}}_{1t} = \tilde{\boldsymbol{x}}_{2t} * \tilde{\boldsymbol{x}}_{2i} = 1$. Let us verify the two conditions that we require the valid adversarial examples to satisfy:

- **Adversarial budget.** Note that we have

$$d_{\mathbb{L}}(\boldsymbol{x}_1, \tilde{\boldsymbol{x}}_{1t}) = d_{\mathbb{L}}(\boldsymbol{x}_2, \tilde{\boldsymbol{x}}_{2t}) = \operatorname{acosh}(\sqrt{1+\delta^2})\,.$$

  Thus, by choosing $\delta = \sqrt{\cosh^2(\alpha) - 1}$, we achieve the maximal permitted perturbation $\alpha$.

- **Inconsistent prediction for the current classifier, i.e.,** $h_{\boldsymbol{w}_t}(\tilde{\boldsymbol{x}}_{1t/2t}) \ne h_{\boldsymbol{w}}(\boldsymbol{x}_{1/2})$. Note that we have $\delta \ge \alpha > \epsilon$, which further implies that $\delta > \epsilon \ge \epsilon'$. In round $t$,

$$\boldsymbol{w}_t * \tilde{\boldsymbol{x}}_{1t} = \epsilon'\sqrt{1+\delta^2} - \delta\sqrt{1+\epsilon'^2} < 0$$
$$\boldsymbol{w}_t * \tilde{\boldsymbol{x}}_{2t} = -\epsilon'\sqrt{1+\delta^2} + \delta\sqrt{1+\epsilon'^2} > 0\,,$$

  which is a consequence of the relation $\delta > \epsilon'$ as follows:

$$\delta^2 > \epsilon'^2 \Rightarrow \delta^2 + \epsilon'^2\delta^2 > \epsilon'^2 + \epsilon'^2\delta^2 \Rightarrow \delta^2(1+\epsilon'^2) > \epsilon'^2(1+\delta^2)$$
$$\Rightarrow \delta\sqrt{1+\epsilon'^2} > \epsilon'\sqrt{1+\delta^2}\,.$$

Recall that, in each round of Algorithm 2 with an ERM update, we create adversarial examples and add them to the training set, i.e., after round $t$ we have

$$\mathcal{S}_{<t} = \mathcal{S} \cup \bigcup_{i=0}^{t-1}\{(\tilde{\boldsymbol{x}}_{1i}, y_{1i}), (\tilde{\boldsymbol{x}}_{2i}, y_{2i})\}\,.$$

Now for each $t$ and any $i < t$, we have

$$\boldsymbol{w}_t * \tilde{\boldsymbol{x}}_{1i} = \epsilon'\sqrt{1+\delta^2} - \delta\sqrt{1+\epsilon'^2} \cdot \cos(\theta) > 0$$
$$\boldsymbol{w}_t * \tilde{\boldsymbol{x}}_{2i} = -\epsilon'\sqrt{1+\delta^2} + \delta\sqrt{1+\epsilon'^2} \cdot \cos(\theta) < 0\,,$$

i.e., $\boldsymbol{w}_t$ linearly separates $\mathcal{S}_{<t}$.

Therefore, $\{\boldsymbol{w}_t\}$ in (C.36) form an admissible sequence of the classifiers, where $\boldsymbol{w}_t$ linearly separates $\mathcal{S}_t$ while achieving the margin of at most $\epsilon$ on the original dataset $\mathcal{S}$. The length of the sequence is bounded by the size of the spherical code $\mathcal{C}(d, \epsilon'\cosh(\alpha))$, which give us that

$$T = A\left(d, \arccos\left(\rho \cdot \frac{\epsilon'\sqrt{1+\delta^2}}{\delta\sqrt{1+\epsilon'^2}}\right)\right) = A\left(d, \arccos\left(\rho \cdot \frac{\sinh(\epsilon)\cosh(\alpha)}{\sqrt{\cosh^2(\alpha)-1}\sqrt{1+\sinh^2(\epsilon)}}\right)\right).$$

$\square$

The following result (a restatement of Theorem 4.7 from the main text) then follows by applying a lower bound on the maximal size of spherical codes by Shannon.

**Theorem C.16** (Theorem 4.7). *Suppose Algorithm 2 (with an ERM update) outputs a linear seperator of $\mathcal{S} \cup \mathcal{S}'$. In the worst case, the number of iteration required to achieve the margin at least $\epsilon$ is $\Omega(\exp(d))$.*

*Proof.* The statement of the theorem follows from combining Lemma C.15 with Shannon's lower bound (Theorem A.4) on the maximal size of spherical codes, namely

$$T \geq (1 + o(1))\sqrt{2\pi d}\frac{\cos(\theta)}{\sin^{d-1}(\theta)} .$$

We introduce the shorthand $\theta =: \arccos\left(\frac{A}{B}\right)$, where $A = \rho \sinh(\epsilon)\cosh(\alpha)$ and $B = \sqrt{\cosh^2(\alpha) - 1}\sqrt{1 + \sinh^2(\epsilon)}$, as given by Lemma C.15. We then use two well-known trigonometric identities

$$\cos(\arccos z) = z \quad \text{and} \quad \sin(\arccos z) = \sqrt{1 - z^2}$$

to simplify the trignometric fraction in Shannon's bound:

$$\frac{\cos\theta}{\sin^{d-1}\theta} = \frac{A}{B\left(1 - \frac{A^2}{B^2}\right)^{\frac{d-1}{2}}} = \frac{AB^{d-2}}{(B^2 - A^2)^{\frac{d-1}{2}}} .$$

For the denominator, note that

$$
\begin{aligned}
B^2 - A^2 &= (\cosh^2(\alpha) - 1)(1 + \sinh^2(\epsilon)) - \rho^2 \sinh^2(\epsilon)\cosh^2(\alpha) \\
&= (1 - \rho^2)\sinh^2(\epsilon)\cosh^2(\alpha) + \cosh^2(\alpha) - 1 - \sinh^2(\epsilon) \\
&\overset{(i)}{\simeq} \cosh^2(\alpha) - 1 - \sinh^2(\epsilon)
\end{aligned}
$$

where (i) follows from the fact that we can choose $\rho$ arbitrary close to 1. Putting everything together, we have the lower bound

$$T \geq (1 + o(1))\sqrt{2d}\frac{\rho \sinh(\epsilon)\cosh(\alpha)\left(\sqrt{\cosh^2(\alpha) - 1}\sqrt{1 + \sinh^2(\epsilon)}\right)^{d-2}}{\left(\cosh^2(\alpha) - 1 - \sinh^2(\epsilon)\right)^{\frac{d-1}{2}}} = \Omega(\exp d) ,$$

which is exponential in $d$. □

# D   Dimension-distortion trade-off

## D.1   Euclidean case

In the Euclidean case, we relate the distance of the support vectors and the size of margin via side length - altitude relations. Let $\boldsymbol{x}, \boldsymbol{y} \in \mathbb{R}^d$ denote support vectors, such that $\langle \boldsymbol{x}, \boldsymbol{w}\rangle > 0$ and $\langle \boldsymbol{y}, \boldsymbol{w}\rangle < 0$ and $\mathrm{margin}(\boldsymbol{w}) = \epsilon$. We can rotate the decision boundary, such that the support vectors are not unique. Wlog, assume that $\boldsymbol{x}_1, \boldsymbol{x}_2$ are equidistant from the decision boundary and $\|\boldsymbol{w}\| = 1$. In this setting, we show the following relation:

**Theorem D.1** (Thm. 5.1). $\epsilon' \geq \frac{\epsilon}{c_E^3}$.

*Proof.* Let $d_1 = d_{\mathcal{X}}(\phi_E^{-1}(\boldsymbol{x}_1), \phi_E^{-1}(\boldsymbol{y}))$, $d_2 = d_{\mathcal{X}}(\phi_E^{-1}(\boldsymbol{x}_2), \phi_E^{-1}(\boldsymbol{y}))$ and $d_3 = d_{\mathcal{X}}(\phi_E^{-1}(\boldsymbol{x}_1), \phi_E^{-1}(\boldsymbol{x}_2))$ the distances between the support vectors in the original space. In the Euclidean embedding space we have

$$d_1' = d_E(\boldsymbol{x}_1, \boldsymbol{y}) \geq \frac{d_1}{c_E}$$

$$d_2' = d_E(\boldsymbol{x}_2, \boldsymbol{y}) \geq \frac{d_2}{c_E}$$

$$d_3' = d_E(\boldsymbol{x}_1, \boldsymbol{x}_2) \geq \frac{d_3}{c_E} .$$

$d'_1, d'_2, d'_3$ are the side lengths of a triangle, whose altitude is given by the margin: $h = 2\epsilon'$. With Heron's equation we get

$$h = 2\epsilon' = \frac{2}{d'_3}\sqrt{s'(s' - d'_1)(s' - d'_2)(s' - d'_3)} \,,$$

where $s' = \frac{1}{2}(d'_1 + d'_2 + d'_3)$. In $\mathcal{X}$ we have $s' = \frac{1}{2c_E}(d_1 + d_2 + d_3) = \frac{s}{c_E}$. Then we have with respect to the actual distance relations

$$h = 2\epsilon' \geq \frac{2}{c_E d_3}\sqrt{c_E^{-4} s(s - d_1)(s - d_2)(s - d_3)} = 2\frac{\epsilon}{c_E^3} \,,$$

which gives the claim. $\qquad\qquad\qquad\qquad\qquad\qquad\qquad\qquad\qquad\qquad\qquad\qquad\qquad\qquad\quad \square$

## D.2 Hyperbolic case

As in the Euclidean case, we want to relate the margin to the distance of the support vectors. Since the distortion can be expressed in terms of the distances of support vector in the original and the embedding space, this allows us to study the influence of distortion on the margin.

We will derive the relation in the half-space model ($\mathbb{P}^2$). However, since the theoretical guarantees above consider the upper sheet of the Lorentz model ($\mathbb{L}_+^{d'}$), we have to map between the two spaces.

**Assumption 4.** We make the following assumptions on the underlying data $\mathcal{X}$ and the embedding $\phi_H$:

1. $\mathcal{X}$ is linearly separable;

2. $\mathcal{X}$ is hierarchical, i.e., has a partial order relation;

3. $\phi_H$ preserves the partial order relation and the root is mapped onto the origin of the embedding space.

Under these assumptions, the hyperbolic embedding $\phi_H$ has two sources of distortion:

1. the (multiplicative) distortion of pairwise distances, measured by the factor $\frac{1}{c_H}$;

2. the distortion of order relations, in most embedding models captured by the alignment of ranks with the Euclidean norm.

Under Ass. 4, order relationships are preserved and the root is mapped to the origin. Therefore, the distortion on the Euclidean norms is given as follows:

$$\|\phi_H(x)\| = d_E(\phi_H(\boldsymbol{x}), \phi_H(0)) = \frac{d_{\mathcal{X}}(\boldsymbol{x}, 0)}{c_H} \,,$$

i.e., the distortion on both pairwise distances and norms is given by a factor $\frac{1}{c_H}$.

*Note on notation:* In the following, a bar over any symbol indicates the Euclidean expression.

### D.2.1 Mapping from $\mathbb{L}_+^{d'}$ to $\mathbb{P}^2$

First, note that a transformation $\boldsymbol{v} \mapsto B\boldsymbol{v}$ with $B = \begin{pmatrix} 1 & 0 \\ 0 & A \end{pmatrix}$ and an orthogonal matrix $A$ is isometric, i.e., it preserves the Minkowski product [6]:

$$(B\boldsymbol{u}) * (B\boldsymbol{v}) = u_0 v_0 - \boldsymbol{u}_{1:d'}^T A^T A \boldsymbol{v}_{1:d'} = u_0 v_0 - \boldsymbol{u}_{1:d'}^T \boldsymbol{v}_{1:d'} = \boldsymbol{u} * \boldsymbol{v} \,.$$

Setting the first column of $A$ to $\frac{\boldsymbol{w}_{1:d'}}{\|\boldsymbol{w}_{1:d'}\|}$ we can isometrically transform the decision hyperplane as $\hat{\boldsymbol{w}} = B\boldsymbol{w} = (\hat{\boldsymbol{w}}_0, \|\hat{\boldsymbol{w}}_{1:d'}\|, 0, \ldots, 0)$. Analogously, we can transform any point in $\mathbb{L}_+^{d'}$. In the following, we will use the shorthand $\lambda = \frac{\hat{w}_0}{\hat{w}_1}$. We can then use the maps defined in section A.2 to map $\hat{\boldsymbol{x}} = B\boldsymbol{x} \in \mathbb{L}_+^2$ onto $\boldsymbol{z} \in \mathbb{P}^2$, i.e. applying $(\pi_{BP} \circ (\pi_{LB} \circ B))$ to any $\boldsymbol{x} \in \mathbb{L}_+^2$ gives $\boldsymbol{z} \in \mathbb{P}^2$.

**Remark D.2** (Effect of hyperbolic distortion on Euclidean distances in the Poincare half plane)**.** Note that the hyperbolic distance in the Poincare half plane can be written as follows:

$$d_{\mathbb{P}}((x_0, x_1), (y_0, y_1)) = 2 \operatorname{asinh}\left(\frac{1}{2}\sqrt{\frac{(x_0 - y_0)^2 + (x_1 - y_1)^2}{x_1 y_1}}\right)$$
$$= 2 \operatorname{asinh}\left(\frac{1}{2}\frac{d_E((x_0, x_1), (y_0, y_1))}{\sqrt{x_1 y_1}}\right) \,.$$

If $c_H$ denotes the hyperbolic distortion, we get

$$d'_{\mathbb{P}} = \frac{d_{\mathbb{P}}}{c_H} = 2 \operatorname{asinh}\left(\frac{1}{2}\frac{d'_E}{\sqrt{x_1 y_1}}\right)$$
$$\Rightarrow \quad \frac{1}{2}\frac{d'_E}{\sqrt{x_1 y_1}} = \sinh\left(\frac{2\operatorname{asinh}\left(\frac{1}{2}\frac{d_E}{\sqrt{x_1 y_1}}\right)}{2 c_H}\right) \gtrsim \frac{1}{2}\frac{d_E}{c_H\sqrt{x_1 y_1}} \,.$$

This suggests, that the effect of hyperbolic distortion on the Euclidean distances can be quantified by a comparable factor, i.e. $d'_E \gtrsim \frac{d_E}{c_H}$.

**Lemma D.3** (Relation between h-margin and E-margin)**.** *Let $\gamma_H$ be the margin of a hyperbolic classifier $\boldsymbol{w} \in \mathbb{R}^{d'+1}$. Then the Euclidean margin $\gamma_E$ of $\boldsymbol{w}$ is bounded as follows: $\gamma_E \geq \sinh(\gamma_H)$.*

*Proof.* We again write the hyperbolic distance in the Poincare half plane in terms of the Euclidean distance of the ambient space:

$$d_{\mathbb{P}}((x_0, x_1), (y_0, y_1)) = 2 \operatorname{asinh}\left(\frac{1}{2}\sqrt{\frac{(x_0 - y_0)^2 + (x_1 - y_1)^2}{x_1 y_1}}\right)$$
$$= 2 \operatorname{asinh}\left(\frac{1}{2}\frac{d_E((x_0, x_1), (y_0, y_1))}{\sqrt{x_1 y_1}}\right) \,,$$

where $\boldsymbol{y} \in \mathcal{H}_w$ is the point closest to the support vector $\boldsymbol{x} \in \mathbb{L}_+^{d'}$ on the decision boundary. Therefore, the hyperbolic margin is $d_{\mathbb{P}}(\boldsymbol{x}, \boldsymbol{y}) = \gamma_H$ and the Euclidean margin is $d_E(\boldsymbol{x}, \boldsymbol{y}) = \gamma_E$.

Since we mapped the feature space onto the Poincare half plane, $\boldsymbol{y}$ has the coordinates $\boldsymbol{y} = (\tilde{y}_0, \tilde{y}_1, 0, \ldots, 0)$ where $\tilde{y}_0 = y_0$ and $\tilde{y}_1 = \frac{\boldsymbol{w}'^T \boldsymbol{y}'}{\|\boldsymbol{w}'\|}$. Similarly, $\boldsymbol{x}$ has the coordinates $\boldsymbol{x} = (\tilde{x}_0, \tilde{x}_1, 0, \ldots, 0)$. The transformation preserves the Minkowski product. Therefore we have

$$\boldsymbol{y} * \boldsymbol{y} = y_0^2 - \boldsymbol{y}'^2 = \hat{y}_0^2 - \underbrace{\left(\frac{\boldsymbol{w}'^T \boldsymbol{y}'}{\|\boldsymbol{w}'\|}\right)^2}_{=\hat{y}_1} = 1$$

and similarly $\boldsymbol{x} * \boldsymbol{x} = \hat{x}_0^2 - \hat{x}_1^2 = 1$. This implies

$$①\quad \hat{y}_1 = \sqrt{\hat{y}_0^2 - 1}, \ \hat{x}_1 = \sqrt{\hat{x}_0^2 - 1} \,,$$

and further

$$②\quad \hat{x}_0, \ \hat{y}_0 \geq 1 \,.$$

We want to show that $\hat{x}_1 \hat{y}_1 \geq 1$. For this, first, note that since $\boldsymbol{y} \in \mathcal{H}_w$ and the hyperbolic margin is $\gamma_H$, we have

$$0 = \boldsymbol{w} * \boldsymbol{y} = w_0 y_0 - \boldsymbol{w}'^T \boldsymbol{y}'$$
$$\Rightarrow \boldsymbol{w}'^T \boldsymbol{y}' = w_0 y_0 \,.$$

This gives

$$\boldsymbol{y} * \boldsymbol{y} = y_0^2 - \frac{w_0^2 y_0^2}{\|\boldsymbol{w}'\|^2} = 1$$
$$\Rightarrow 0 = y_0^2 - \frac{w_0^2 y_0^2}{\|\boldsymbol{w}'\|^2} - 1 \,,$$

and therefore

$$\textcircled{3} \quad y_0 = \frac{1}{\sqrt{1 - \frac{w_0^2}{\|\boldsymbol{w}'\|}}} \; .$$

Since the hyperbolic margin is $\gamma_H$, we further have

$$d_{\mathbb{P}}(\boldsymbol{x}, \boldsymbol{y}) = \mathrm{acosh}(\boldsymbol{x} * \boldsymbol{y}) \geq \gamma_H \quad \Rightarrow \quad \boldsymbol{x} * \boldsymbol{y} \geq \cosh(\gamma_H) \geq 1 \; ,$$

and therefore

$$x_0 y_0 - x_1 y_1 \geq 1$$
$$x_0 y_0 - \sqrt{x_0^2 - 1}\sqrt{y_0^2 - 1} \geq 1$$
$$(x_0 y_0 - 1)^2 \geq (x_0^2 - 1)(y_0^2 - 1)$$
$$x_0^2 y_0^2 - 2 x_0 y_0 + 1 \geq x_0^2 y_0^2 - x_0^2 - y_0^2 + 1$$
$$\Rightarrow 0 \leq (x_0 - y_0)^2 \; ,$$

which implies

$$\textcircled{4} \quad x_0 \geq y_0 \; .$$

This gives for $x_1 y_1$ the following:

$$x_1 y_1 \overset{\textcircled{1}}{=} \sqrt{x_0^2 - 1}\sqrt{y_0^2 - 1} \overset{\textcircled{4}}{\geq} y_0^2 - 1 \overset{\textcircled{3}}{=} \frac{1}{1 - \frac{w_0^2}{\|\boldsymbol{w}'\|}} - 1 = \frac{w_0^2}{\|\boldsymbol{w}'\|^2 - w_0^2} \; .$$

By assumption we have $\boldsymbol{w} * \boldsymbol{w} = w_0^2 - \|\boldsymbol{w}'\|^2 = -1$, which gives for the denominator $-w_0^2 + \|\boldsymbol{w}'\|^2 = 1$. It remains to show that $w_0^2 \geq 1$.

For this last step, we want to show that mass concentrates on $w_0$ as the classifier is updated, ensuring $w_0 \geq 1$. By construction, we have initially $\boldsymbol{w} * \boldsymbol{w} = -1$. Wlog, assume that initially $w_0 \geq 1$. An initialization of this form can always be found, e.g., by setting $\boldsymbol{w} = (a, \sqrt{1 + a^2}, 0, \dots, 0)$ for some $a \geq 0$. If the $i^{th}$ update is negative ($y^i x_0^i < 0$), then $|\boldsymbol{w}|_*$ will initially decrease, but the normalization step will scale away the effect on $w_0$. However, if the $i^{th}$ update is non-negative ($y^i x_0^i \geq 0$), it will increase $w_0$. Over time, the positive updates concentrate the mass on $w_0$. Since we initialized to $w_0 \geq 1$, the condition will always stay valid. With the arguments above, this implies $x_1 y_1 \geq 1$. Inserting the latter in the expression above, we get

$$d_H = 2 \,\mathrm{asinh}\left(\frac{1}{2} \frac{d_E}{\sqrt{x_1 y_1}}\right) \leq 2 \,\mathrm{asinh}\left(\frac{d_E}{2}\right)$$
$$\Rightarrow d_E \geq 2 \sinh\left(\frac{d_H}{2}\right) \geq \sinh(d_H) \; .$$

$\square$

### D.2.2 Characterizing the margin

In $\mathbb{P}^2$ the decision hyperplane corresponding to $\hat{w} = Bw$ corresponds to a hypercircle $\mathcal{K}_w$. One can show, that its radius is given by $r_w = \sqrt{\frac{1-\lambda}{1+\lambda}}$ [6], by computing the hyperbolic distance between a point on the decision boundary and one of the hypercircle's ideal points. Further note, that the support vectors lie on hypercircles $\mathcal{K}_x$ and $\mathcal{K}_y$, which correspond to the set of points of hyperbolic distance $\epsilon$ (i.e., the margin) from the decision boundary. We again assume wlog that at least one support vector is not unique and let $x_1, x_2 \in \mathcal{K}_x$ and $y \in \mathcal{K}_y$ (see Fig. 5).

**Theorem D.4** (Thm. 5.2). $\epsilon' \approx \epsilon$.

*Proof.* Our proof consists of three steps:

Figure 5: Support vectors on hypercircles $\mathcal{K}_x$ and $\mathcal{K}_y$ with decision hypercircle $\mathcal{K}_w$.

Figure 6: Margin as distance between hypercircles $\mathcal{K}_x$ and $\mathcal{K}_y$.

**Step 1: Find Euclidean radii and centers of hypercircles.** The hypercircles $\mathcal{K}_x, \mathcal{K}_y$ correspond to arcs of Euclidean circles $\bar{\mathcal{K}}_x, \bar{\mathcal{K}}_y$ in the full plane that are related through circle inversion on the decision circle $\bar{\mathcal{K}}_w$ (i.e., the Euclidean circle corresponding to $\mathcal{K}_w$); see Fig. 5. We can construct a "mirror point" $y' \in \bar{\mathcal{K}}_x$ of $y$ by circle inversion on $\bar{\mathcal{K}}_w$. We have the following (Euclidean) distance relations: The circle inversion gives

$$\bar{d}(y', \bar{c}_w)\, \bar{d}(y, \bar{c}_w) = r_w^2 \,,$$

where $\bar{c}_w$ denotes the center of $\bar{\mathcal{K}}_w$. Furthermore, we have (see Fig. 7)

$$\bar{d}(y, \bar{c}_w) = \bar{d}(y', \bar{c}_w) + \bar{d}(y, y') \,.$$

Putting both together, we get an expression for the Euclidean distance of $y$ and $y'$:

$$①\quad \bar{d}(y, y') = \bar{d}(\bar{c}_w, y) - \frac{\bar{r}_w^2}{\bar{d}(\bar{c}_w, y)} \,.$$

Here, we have by construction $\bar{c}_w = (0, a, 0, \ldots, 0)$ with a free parameter $a$. Wlog, assume $\bar{c}_w = (0, -1, 0, \ldots, 0)$. Next, consider the triangle $\Delta(x_1, x_2, y)$. We can express its altitude $h$ in terms of the side length $\bar{d}(x_1, x_2) =: d_1$, $\bar{d}(x_1, y) =: d_2$ and $\bar{d}(x_2, y) =: d_3$ via Heron's formula:

$$h = \frac{2}{d_1} \sqrt{s(s - d_1)(s - d_2)(s - d_3)} \,,$$

where $s = \frac{1}{2}(d_1 + d_2 + d_3)$. Now, consider the triangle $\Delta(x_1, x_2, y')$. Due to the relation between $y$ and $y'$ in $①$, its altitude $h_x$ is related to $h$ as

$$②\quad h_x = h - \bar{d}(y, y') \,.$$

Figure 7: Geometric construction for computing the center and radius of the hypercircle $\mathcal{K}_x$.

With the side length - altitude relations given in $\Delta(x_1, x_2, y)$ and ②, we can compute the length of the other sides $\bar{d}(x_1, y')$ and $\bar{d}(x_2, y')$ as follows (with Pythagoras theorem):

$$\bar{d}(x_1, y') = \left(h_x^2 + \bar{d}(x_1, y)^2 - h^2\right)^{1/2}$$
$$\bar{d}(x_2, y') = \left(h_x^2 + \bar{d}(x_2, y)^2 - h^2\right)^{1/2} .$$

With that, we can compute the radius of $\bar{\mathcal{K}}_x$ as follows: $\bar{\mathcal{K}}_x$ circumscribes $\Delta(x_1, x_2, y')$, therefore its radius $\bar{r}_x$ can be computed via Heron's formula as

$$\bar{r}_x = \frac{\bar{d}(x_1, y') + \bar{d}(x_2, y') + \bar{d}(x_1, x_2)}{4A}$$
$$A = \sqrt{s(s - \bar{d}(x_1, y'))(s - \bar{d}(x_2, y'))(s - \bar{d}(x_1, x_2))}$$

where $s = \frac{1}{2}(\bar{d}(x_1, y') + \bar{d}(x_2, y') + \bar{d}(x_1, x_2))$. With an analog construction, we can compute the radius $\bar{r}_y$ of $\bar{\mathcal{K}}_y$ as function of $\bar{d}(x_1', x_2')$, $\bar{d}(x_1', y)$ and $\bar{d}(x_2', y)$ via relations in the triangle $\Delta(x_1', x_2', y)$.

**Step 2: Express h-margin as distance between hypercircles.** As shown in Fig. 6, the margin is the hyperbolic distance from a point on $\mathcal{K}_x, \mathcal{K}_y$ to $\mathcal{K}_w$, corresponding to the length of a geodesic connecting the point with the closest point on $\mathcal{K}_w$. Let $v \in \mathcal{K}_x$ and $u \in \mathcal{K}_w$ the closest point on the decision circle. From the geometry of the Poincare half plane we know that there exists a Möbius transform $\theta \in \text{Möb}(\mathbb{P}^2)$ such that the images $\theta(u) = i\mu$ and $\theta(v) = i\nu$ of $u, v$ lie on the positive imaginary axis. Since the hyperbolic distance is invariant under Möbius transforms, we get

$$d(u, v) = d(\theta(u), \theta(v)) = d(i\mu, i\nu) = \left| \log \frac{\nu}{\mu} \right| .$$

Similarly, we can express the distance between between support vectors $x \in \mathcal{K}_x$ and $y \in \mathcal{K}$, which is twice the hyperbolic margin: Let $\theta(x) = i\mu_x$ and $\theta(y) = i\mu_y$, where $\mu_x, \mu_y$ are given by the intersection points of $\mathcal{K}_x, \mathcal{K}_y$ with the imaginary axis. Then

$$2\epsilon = d(x, y) = \left| \log \frac{\mu_y}{\mu_x} \right| .$$

We can express $\mu_x, \mu_y$ in terms of the centers and radii of $\mathcal{K}_x, \mathcal{K}_y$ as follows (Fig. 6)

$$\mu_x = \bar{c}_x^{(2)} + \bar{r}_x$$
$$\mu_y = \bar{c}_y^{(2)} + \bar{r}_y ,$$

where $c^{(2)}$ denotes the second coordinate of the point $c \in \mathbb{P}^m$. Putting everything together, we get the following expression for the margin:

$$\textcircled{3} \quad \epsilon = \frac{1}{2}\left| \log \frac{\bar{c}_y^{(2)} + \bar{r}_y}{\bar{c}_x^{(2)} + \bar{r}_x} \right| .$$

**Step 3: Evaluate Distortion.** As discussed above (Prop. 5.1), the influence of distortion on the altitude $h$ in the triangle $\Delta(x_1, x_2, y)$ is given by the factor $\frac{1}{c_H}$.

$$\textcircled{4} \quad h' = \frac{h}{c_H} .$$

$\bar{r}_x$ depends on pairwise distances between support vectors and $h$, which are distorted by a factor $\frac{1}{c_H}$ (by assumption on $\phi_H$ and $\textcircled{4}$). $\bar{r}_x$ depends further on $h_x$ which in turn depends on $\bar{d}(c_w, y)$. The latter depends on the Euclidean norm of the support vector $y$, i.e., $\|y\|$. With Ass. 4 the total multiplicative distortion is then at most of a factor $\frac{1}{c_H}$. We can derive an analogue result for $\bar{r}_y$. For the center $\bar{c}_x$ note the following:

$$\bar{c}_x^{(2)} = \frac{1}{2}\left[ (1 - \bar{r}_w^2)\tilde{x}_0 - (1 + \bar{r}_w^2)\tilde{x}_1 \right] ,$$

where $(\tilde{x}_0, \tilde{x}_1, 0, \ldots, 0) = \tilde{x} = (\pi_{BP} \circ (\pi_{LB} \circ B))$ and $\bar{r}_w = \sqrt{\frac{1-\lambda}{1+\lambda}}$. Rewriting

$$(1 - \bar{r}_w^2)\tilde{x}_0 = \frac{2\tilde{w}_0\tilde{x}_0}{\tilde{w}_1 + \tilde{w}_0}$$

$$(1 + \bar{r}_w^2)\tilde{x}_1 = \frac{2\tilde{w}_1\tilde{x}_1}{\tilde{w}_1 + \tilde{w}_0} ,$$

we get

$$\bar{c}_x^{(2)} = \frac{\tilde{w}^T \tilde{x}}{\tilde{w}_0 + \tilde{w}_1} .$$

Similarly, one can derive

$$\bar{c}_y^{(2)} = \frac{\tilde{w}^T \tilde{y}}{\tilde{w}_0 + \tilde{w}_1} ,$$

for $(\tilde{y}_0, \tilde{y}_1, 0, \ldots, 0) = \tilde{y} = (\pi_{BP} \circ (\pi_{LB} \circ B))$. Both are only affected by distortion of the form (2), i.e. the multiplicative distortion is given by a factor $\frac{1}{c_H}$. Inserting this into the margin expression ($\textcircled{4}$) gives

$$\epsilon' = \frac{1}{2}\left| \log \frac{c_y' + r_y'}{c_x' + r_x'} \right| \gtrsim \frac{1}{2}\left| \log \frac{\frac{c_y}{c_H} + \frac{r_y}{c_H}}{c_H c_x + c_H r_x} \right| = \frac{1}{2}\left| \log \left( \frac{1}{c_H^2} \frac{c_y + r_y}{c_x + r_x} \right) \right|$$

$$= \frac{1}{2}\left| \underbrace{\log \frac{1}{c_H^2}}_{\approx 0} + \log \frac{c_y + r_y}{c_x + r_x} \right| \overset{\dagger}{\approx} \frac{1}{2}\left| \log \frac{c_y + r_y}{c_x + r_x} \right| = \epsilon ,$$

where (†) follows from $c_H = O(1 + \epsilon)$ with $\epsilon > 0$ small, by Thm. A.3. $\qquad \square$

# E  Adversarial perceptron

With the geometric tools introduced in Appendix D, we can now also proof Lemma C.4. We restate the result from the main text:

**Lemma E.1.** *(Adversarial perceptron, Lem. C.4) Let $\bar{w}$ be the max-margin classifier of $\mathcal{S}$ with margin $\gamma_H$. At each iteration of Alg. 2, $\bar{w}$ linearly separates $\mathcal{S} \cup \mathcal{S}'$ with margin at least $\frac{\gamma_H}{\cosh(\alpha)}$.*

*Proof.* In the following, we again use the shorthand $|\boldsymbol{u}| = \sqrt{\pm\boldsymbol{u}\ast\boldsymbol{u}}$, with "+", if $\boldsymbol{u}$ is space-like (i.e., $\boldsymbol{u}\ast\boldsymbol{u} > 0$) and "-", if $\boldsymbol{u}$ is time-like (i.e., $\boldsymbol{u}\ast\boldsymbol{u} < 0$). Since $\bar{\boldsymbol{w}}$ is "time-like" and $\boldsymbol{x}, \tilde{\boldsymbol{x}}$ space-like, we have

$$\textcircled{1} \quad |\bar{\boldsymbol{w}}\ast\boldsymbol{x}| = |\bar{\boldsymbol{w}}|\,|\boldsymbol{x}|\,\cosh\angle(\bar{\boldsymbol{w}},\boldsymbol{x})$$
$$|\bar{\boldsymbol{w}}\ast\tilde{\boldsymbol{x}}| = |\bar{\boldsymbol{w}}|\,|\tilde{\boldsymbol{x}}|\,\cosh\angle(\bar{\boldsymbol{w}},\tilde{\boldsymbol{x}})\,.$$

To prove the statement, we first transform the problem from the Lorentz model $\mathbb{L}^d$ to the Poincare half plane $\mathbb{P}^2$ using the map $(\pi_{BP}\circ(\pi_{LB}\circ B))$. Then the adversarial margin is given by the Euclidean distance of the hypercircle $\mathcal{K}_{\tilde{x}}$ through $\tilde{\boldsymbol{x}}$ and the decision hypercircle $\mathcal{K}_w$. First, note that we can express this as the hyperbolic distance of the points $(0, \theta(\tilde{\boldsymbol{x}}))$ and $(0, r_w)$, where $\theta \in \text{Möb}(\mathbb{P}^2)$ is a Möbius transform that maps $\tilde{\boldsymbol{x}}$ to the imaginary axis. Importantly, any such $\theta$ leaves the Minkowski product invariant. One can show [6] that

$$\theta(\tilde{\boldsymbol{x}}) = c_{\tilde{x}} + \sqrt{c_{\tilde{x}}^2 + r_w}$$

where $c_{\tilde{x}} = \frac{1}{2}\left((1 - r_w^2)\tilde{x}_0 - (1 + r_w^2)\tilde{x}_1\right)$ is the Euclidean center of $\mathcal{K}_{\tilde{x}}$. The hyperbolic distance is then given by

$$\textcircled{2} \quad \left|\log\frac{\theta(\tilde{\boldsymbol{x}})}{r_w}\right| = \left|\log\left(\frac{c_{\tilde{x}}}{r_w} + \sqrt{\frac{c_{\tilde{x}}^2}{r_w^2} + 1}\right)\right| = \left|\text{asinh}\left(\frac{c_{\tilde{x}}}{r_w}\right)\right|\,.$$

Note, that

$$\frac{c_{\tilde{x}}}{r_w} = \frac{1}{2}\left[\left(\frac{1}{r_w} - r_w\right)\tilde{x}_0 - \left(\frac{1}{r_w} + r_w\right)\tilde{x}_1\right]\,,$$

where

$$\frac{1}{r_w} - r_w = \sqrt{\frac{1+\lambda}{1-\lambda}} - \sqrt{\frac{1-\lambda}{1+\lambda}} = \frac{2\lambda}{\sqrt{1-\lambda^2}} = \frac{2w_0}{\sqrt{w_1^2 - w_0^2}}$$

$$\frac{1}{r_w} + r_w = \frac{2}{\sqrt{1-\lambda^2}} = \frac{2w_1}{\sqrt{w_1^2 - w_0^2}}\,.$$

This gives

$$\textcircled{3} \quad \left|\text{asinh}\left(\frac{c_{\tilde{x}}}{r_w}\right)\right| = \left|\text{asinh}\left(\frac{w_0\tilde{x}_0 - w_1\tilde{x}_0}{\sqrt{w_1^2 - w_0^2}}\right)\right| = \left|\text{asinh}\left(\frac{\boldsymbol{w}\ast\tilde{\boldsymbol{x}}}{|\boldsymbol{w}|}\right)\right|\,.$$

Using $\textcircled{2}$, we can express the adversarial margin in terms of the margin and the distance between features and adversarial samples as follows:

$$\left|\text{asinh}\left(\frac{c_{\tilde{x}}}{\boldsymbol{w}}\right)\right| = \left|\text{asinh}\left(\frac{c_{\tilde{x}}}{c_x}\underbrace{\frac{c_x}{r_w}}_{\geq\sinh(\gamma_H)}\right)\right| \stackrel{\dagger}{\geq} \left|\text{asinh}\left(\frac{c_{\tilde{x}}}{c_x}\sinh(\gamma_H)\right)\right|\,,$$

where (†) follows from the assumption that $y(\boldsymbol{w}\ast\boldsymbol{x}) \geq \sinh(\gamma_H)$ (with margin $\gamma_H$). We further show above that we can express the Euclidean centers as

$$c_x = \frac{\boldsymbol{w}\ast\boldsymbol{x}}{w_0 + w_1}\,, \qquad c_{\tilde{x}} = \frac{\boldsymbol{w}\ast\tilde{\boldsymbol{x}}}{w_0 + w_1}\,.$$

Wlog, assume that $\boldsymbol{w}\ast\boldsymbol{x} > 0$; then $\boldsymbol{w}\ast\tilde{\boldsymbol{x}} < 0$ and therefore

$$c_x = \frac{|\boldsymbol{w}\ast\boldsymbol{x}|}{w_0 + w_1}\,, \qquad c_{\tilde{x}} = \frac{-|\boldsymbol{w}\ast\tilde{\boldsymbol{x}}|}{w_0 + w_1}\,.$$

Inserting $\textcircled{1}$ above in $\textcircled{3}$, we get

$$\text{asinh}\left(-\frac{|\boldsymbol{w}\ast\tilde{\boldsymbol{x}}|}{|\boldsymbol{w}\ast\boldsymbol{x}|}\sinh(\gamma_H)\right) \stackrel{\textcircled{1}}{=} \text{asinh}\left(-\frac{|\boldsymbol{w}|\,|\tilde{\boldsymbol{x}}|\,\cosh(\angle(\boldsymbol{w},\tilde{\boldsymbol{x}}))}{|\boldsymbol{w}|\,|\boldsymbol{x}|\,\cosh(\angle(\boldsymbol{w},\boldsymbol{x}))}\sinh(\gamma_H)\right)$$

$$= \text{asinh}\left(-\frac{|\tilde{\boldsymbol{x}}|\,\cosh(\angle(\boldsymbol{w},\tilde{\boldsymbol{x}}))}{|\boldsymbol{x}|\,\cosh(\angle(\boldsymbol{w},\boldsymbol{x}))}\sinh(\gamma_H)\right)$$

$$\stackrel{\dagger}{\geq} \text{asinh}\left(-\frac{|\tilde{\boldsymbol{x}}|}{|\boldsymbol{x}|}\sinh(\gamma_H)\right)\,,$$

where (†) follows from $\boldsymbol{w}$ being a better classifier for $\boldsymbol{x}$ than for $\tilde{\boldsymbol{x}}$ by construction. Therefore, we have

$$\left| \operatorname{asinh}\left( -\frac{|\boldsymbol{w} * \tilde{\boldsymbol{x}}|}{|\boldsymbol{w} * \boldsymbol{x}|} \sinh(\gamma_H) \right) \right| \geq \operatorname{asinh}\left( \frac{|\tilde{\boldsymbol{x}}|}{|\boldsymbol{x}|} \sinh(\gamma_H) \right) \; .$$

Furthermore, note, that by construction we have $d_{\mathbb{L}}(\boldsymbol{x}, \tilde{\boldsymbol{x}}) \leq \alpha$ and therefore:

$$\operatorname{acosh}(\boldsymbol{x} * \tilde{\boldsymbol{x}}) \leq \alpha \;\Rightarrow\; \boldsymbol{x} * \tilde{\boldsymbol{x}} \leq \cosh(\alpha) \; .$$

Since $\boldsymbol{x}, \tilde{\boldsymbol{x}}$ are both space-like, we further have $|\boldsymbol{x}|\,|\tilde{\boldsymbol{x}}| \leq \boldsymbol{x} * \tilde{\boldsymbol{x}}$. In summary, this gives

$$\textcircled{4}\quad |\boldsymbol{x}| \leq \frac{\cosh(\alpha)}{|\tilde{\boldsymbol{x}}|} \; .$$

Inserting $\textcircled{4}$ above, we get

$$\operatorname{asinh}\left( \frac{|\tilde{\boldsymbol{x}}|}{|\boldsymbol{x}|} \sinh(\gamma_H) \right) \overset{\textcircled{4}}{\geq} \operatorname{asinh}\left( \frac{|\tilde{\boldsymbol{x}}|^2}{\cosh(\alpha)} \sinh(\gamma_H) \right)$$
$$\overset{\ddagger}{=} \operatorname{asinh}\left( \frac{\sinh(\gamma_H)}{\cosh(\alpha)} \right) \; ,$$

where ($\ddagger$) follows from $|\tilde{\boldsymbol{x}}|^2 = \tilde{\boldsymbol{x}} * \tilde{\boldsymbol{x}} = 1$, since $\tilde{\boldsymbol{x}} \in \mathbb{L}^m$. Finally, the claim follows from

$$\operatorname{asinh}\left( \frac{\sinh(\gamma_H)}{\cosh(\alpha)} \right) \geq \frac{\gamma_H}{\cosh(\alpha)} \; .$$

$\square$

# F  Additional Experimental Results

## F.1  Hyperbolic perceptron

To validate the hyperbolic perceptron algorithm, we performed two simple classification experiments. For the two-class data set (ImageNet n09246464 and n07831146), we observe that hyperbolic perceptron can successfully classify the points into the two groups, i.e., it achieves zero test error. In a second experiment, we try hyperbolic perceptron on a linearly non-separable dataset. The algorithm was still able to classify reasonably well.

## F.2  Adversarial Gradient decent

### F.2.1  Choice of loss function

Following the large body of work on large-margin learning in Euclidean space, we tested our approach with the classic hinge (Eq. C.2) and least squares losses (Eq. C.3). While both algorithms work well in practise (see § 6 and Section F.2.2), they do not fulfill Ass. 1 on the whole domain. Therefore, our theoretical guarantees are not valid for those loss functions.

We derive theoretical results for the hyperbolic logistic loss (Eq. C.8) instead, which fulfills Ass. 1. Unfortunately, the hyperparameter $R_\alpha$ is difficult to determine in practice. We therefore decided to omit validation experiments with the hyperbolic logistic loss.

For choosing an *adversarial budget* $\alpha$ in practice, note that Assumption 1(2) imposes a norm constraint on the adversarial examples, relative to the maximal norm of the training points. Given the constant $R_x$, one can estimate an upper bound on $\alpha$. In addition, an upper bound on $\alpha$ depends on how separable the data set is, i.e., the maximal possible margin. Within these constraints, the choice of $\alpha$ is guided by a trade-off between better robustness and longer training time.

### F.2.2  Adversarial GD via least squares loss

Using the same data set as described in § 6, we also try classification in hyperbolic space with adversarial examples using the least squares losses (Eq. C.3). We use the same procedure to find adversarial examples. The results are plotted in Figure 8 with similar conclusions.

Figure 8: Performance of Adversarial GD using smoothed square loss (Eq. C.3). **Left:** Loss $L(\boldsymbol{w})$ on the original data. **Middle:** $\alpha$-robust loss $L_\alpha(\boldsymbol{w})$. **Right:** Hyperbolic margin $\gamma_H$. We vary the adversarial budget $\alpha$ over $\{0, 0.25, 0.5, 0.75\}$. The case $\alpha = 0$ corresponds to the setup in [6].

### F.3 Dimension-distortion trade-off

Euclidean embeddings computed using implementation in Nickel and Kiela [22] by Facebook Research[2].

| $d$ | Euclidean | Hyperbolic |
|----|-----------|------------|
| 4  | 0.54      | 0.51       |
| 8  | 0.53      | 1.00       |
| 16 | 0.68      | 1.00       |

Table 2: Classification performance (test error) in hyperbolic vs. Euclidean space of dimension $d$.

## Footnotes

[2]https://github.com/facebookresearch/poincare-embeddings