[Reviews · NeurIPS 2020]

Review 1

Summary and Contributions: The authors propose a large margin learning algorithm in hyperbolic space. They start with a perceptron-style algorithm and prove its convergence, and then extend it to a large-margin version using adversarial training. Empirical evaluations show the utility of the algorithm, especially on tree-like data which cannot be faithfully embedded into Euclidean spaces. The authors have addressed the questions raised by the reviewers sufficiently and I am keeping my score of acceptance.

Strengths: - The proposed algorithm is more principled with convergence proofs compared to previous work like [6]. Both the perceptron case and the large margin case are covered. - The main ideas are simple and clearly presented, with solid theoretical analysis. - Hyperbolic space embeddings is a very promising approach for tree-structured or network data, so directly having a learning algorithm that works in this space is an important contribution.

Weaknesses: - The experiments are more like a proof-of-concept for the algorithm. They can be strengthened by testing on more datasets to provide better evaluations on the utility of the algorithm. - Figure 3 is not completely clear. It looks like only training losses are shown (3a) and not the test loss. This is confusing. Also, why do the \alpha-robust loss stay relatively stable around iteration 200 in 3b while the margin keeps increasing in 3c? - Given a dataset, it is not clear how to choose the robustness parameter \alpha or how to estimate bounds for it for a search.

Correctness: The claims and empirical methodology are correct to the best of my knowledge.

Clarity: Yes, the paper is on the whole quite clear. It would be nice to have more background discussions on hyperbolic spaces and their use in machine learning, but not crucial due to space limit.

Relation to Prior Work: There are sufficient discussions on related works and how the current work differs.

Reproducibility: Yes

Additional Feedback:


Review 2

Summary and Contributions: Learning a large-margin classifier in hyperbolic rather than Euclidean space was first studied in the ICML 2019 paper "Hyunghoon Cho et al., Large-margin classification in hyperbolic space". Learning representations in hyperbolic spaces is especially beneficial for hierarchical data, requiring significantly fewer dimensions than standard Euclidean spaces. This paper presents the first theoretical guarantees for this kind of work, studying robust large-margin learning, adversarial learning, and dimension-distortion trade-offs in hyperbolic spaces. The theoretical derivation is sound, and the results are important. I appreciate this kind of work! But it may be better to provide some guidance on the real application of the theory, e.g., for what kinds of machine learning problem and data sets, the proposed theory has advantages over traditional theories? For the empirical evaluation, the authors only use one data set, which seems not convincing enough. I agree to accept the paper, but I will not change my original score. The reason is that I think more experiments are needed to verify the practical usefulness.

Strengths: The theoretical derivation is sound, and the results are important.

Weaknesses: For the empirical evaluation, the authors only use one data set, which seems not convincing enough. It may be better to provide some guidance on the real application of the theory, e.g., for what kinds of machine learning problem and data sets, the proposed theory has advantages over traditional theories?

Correctness: Yes

Clarity: Yes

Relation to Prior Work: Yes

Reproducibility: Yes

Additional Feedback: I agree to accept the paper, but I will not change my original score. The reason is that I think more experiments are needed to verify the practical usefulness.


Review 3

Summary and Contributions: The paper explores large margin linear classifier in the hyperbolic space from a theoretical point of view. The first contribution is an adaptation of the classical perceptron algorithm in the hyperbolic space quite similar in the definition to [6] but the authors focus on the analysis of the convergence (which is new). This analysis allows to state the difficulty to learn a large margin classifier in hyperbolic space wrt to the exponential number of iteration required. The second (and main) contribution focuses on this problem, how to learn a large margin classifier. The proposed solution is to consider a loss based on a hyperbolic logistic loss considering adversarial examples. The examples are generated by perturbing available data in order to maximize the logistic loss and such that they are still close to the original data in a distance lower than a given threshold. The proposed approach combining adversial examples and gradient descent allows to reduce to a polynomial time the number of iterations to learn a maximum margin classifier. Finally the authors propose an analysis of the effect of the distortion of the euclidean and hyperbolic embedding on the margin. This important discussion allows to state the superiority of hyperbolic embedding over euclidean when the data is suitable (hierarchical data).

Strengths: The paper is well organized, easy to read. It shows theoretically the benefits of hyperbolic classifiers for hierarchical data. The theoretical results are important and are a clear improvement over the current SOA in hyperbolic classification. *********** After feedback : Thanks to the authors for the clarification, the additional page seems a good idea.

Weaknesses: The experimental section analyses only from a theoretical point of view the results. The authors clearly state that their focus "is to understand the benefits of hyperbolic spaces" but it would be interesting to provide not only training loss and margin results but also usual test errors.

Correctness: To the best of my competence, yes (the supplementary material is very long and I did not have the time to review all the provided details).

Clarity: The paper is very clear, the supplementary material is quite complete (too long maybe ?) but helps to have a deeper comprehension.

Relation to Prior Work: The proposed method is a clear improvement of [7], the SOA is clearly stated and discussed. Minor comment : one recent reference is not discuss in the paper, the Hyperbolic Neural Network (Ganea et al.) which also provide in an other context theoretical results on hyperbolic machine learning.

Reproducibility: Yes

Additional Feedback: Minor comment : in Fig 3, concerning alpha-robust loss, the best result is not achieved for alpha=1 but alpha=0.5. Do you have any idea why ?


Review 4

Summary and Contributions: The paper provides theoretical guarantees for large-margin classifiers in hyperbolic spaces. Firstly, they show that learning a large margin classifier using a "standard" gradient-based algorithm over a margin loss may require an exponential number of iterations. To cope with this, the authors propose a method based on adversarial examples. In particular, the proposed approach enriches the training set with new examples that are perturbations of the input. A (bounded) perturbation x' of example x is chosen (via an optimization) in such a way of leading the classifier h to differently labeled the examples h(x') != h(x). Authors show that such an approach converges to a large margin classifier in polynomial time. Some experimental analyses have been conducted to support theoretical findings.

Strengths: - The paper is well written and clearly states the motivation and the impact of the findings; - The theoretical analysis seems sound; - The theoretical foundings are numerous and relevant to the community.

Weaknesses: - Empirical analyses are rather limited, e.g., only a dataset; - Sometimes the used notation is awkward, e.g., 2nd equation in 4.1; - Some sketch of the proofs would have been welcomed.

Correctness: The claims and methods seem correct. The empirical analysis is correct but barely sufficient.

Clarity: The paper is well structured and well written. The presentation is cured even though sometimes the notation is a bit awkward.

Relation to Prior Work: The authors clearly state that to the best of their knowledge they are the first to study adversarial learning in hyperbolic spaces. Authors also underline the differences w.r.t. [6], that is, a closely related work on support vector machine for hyperbolic spaces.

Reproducibility: Yes

Additional Feedback: Comments: - In the paper, authors often compare the Euclidean margin with the Hyperbolic one. However, it is not clear whether \gamma refers to the very same quantity in both spaces, or it is an 'overload' of the same notation. For example, this makes it hard to assess the impact of Remark 3.2. - Equation (4.2) is a bit rushed. The intuition is clear, but it is not clear from the text why optimizing this problem is equivalent to find the correct adversarial example. - Many of the proofs of the findings (almost all of them) are detailed described in the supplementary material. However, it would be highly appreciated to put in the paper at least a sketchy proof or a couple of lines providing an intuition. - In Section 5, the authors introduce the concept of embedding of hierarchical structures, referring the reader to other results about the embeddability of trees. However, since it represents one of the motivations for the use of hyperbolic spaces (HS), I would have appreciated some more details about the tree embedding in HS. - To add value to the paper, I would suggest including more data sets (e.g., the ones used in [6]) to the empirical evaluation. I have read the author feedback and I am partially satisfied by the response. I think that further experiments would be needed to demonstrate the practical usefulness of the proposed approach.

[Author Response · NeurIPS 2020]

We thank all reviewers for the detailed and encouraging comments.

**General comments**

• *Additional experiments:* We would like to emphasize that the focus of this work is theoretical. Therefore, large-scale
experiments and applications are beyond the scope of the paper. Our experiments serve as the first proof of concept,
which indeed successfully validate our theoretical findings. More extensive empirical testing is left for future work.

• *Additional background/ proof details:* We agree that it would be instructive to include more detailed proof sketches
and additional background on tree-embeddings and hyperbolic methods for ML in the main text. If accepted, we will
use the additional page to provide more details on proof ideas and extend the related work and background sections.

• *Training vs. test error in Figure 3:* Here, our aim is to study the learning/optimization aspect of the underlying
problem. Therefore, in Figure 3, we focus on the training objective and the margin realized on the training set. That
said, since the underlying dataset is well separable, in our experiments with adversarial GD, we do observe that the
obtained models achieve zero test error with higher $\alpha$ resulting in faster convergence to zero error. Similarly, we
briefly discuss in Appendix F.1 that the hyperbolic perceptron achieves zero test error on well separable data. We
will clarify this in the revised version.
On the other hand, while validating our dimension distortion trade-off analysis (line 319-329 and Appendix F.3),
where we are interested in the end-to-end performance based on the distortion introduced by the choice of the
embedding space, we do report test errors in Table 2 (Appendix F.3).

**R1**

• *Saturation of robust loss vs. margin loss:* Please note that the margin in Figure 3(c) shows a worst case quantity
(minimum margin over all data points), whereas the $\alpha$-robust loss in Figure 3(b) depicts an average quantity (mean of
the adversarial loss at all data points). We will add a clarifying remark in the revised version.

• *Choice of $\alpha$ in practice:* Assumption 1(2) imposes a norm constraint on the adversarial examples, relative to the
maximal norm of the training points. Given the constant $R_x$, one can estimate an upper bound on $\alpha$. In addition,
an upper bound on $\alpha$ depends on how separable the data set is, i.e., the maximal possible margin. Within these
constraints, the choice of $\alpha$ is guided by a trade-off between better robustness and longer training time. We will add a
clarifying remark in a revised version.

**R2**

• *When is a hyperbolic approach beneficial:* As discussed in the paper (e.g., Section 2.2 and Section 5), our approach
is suitable for truly hierarchical data, which embeds well into low-dimensional hyperbolic space, but requires a
high-dimensional Euclidean embedding space for accurate representation. Many data sets that we encounter in
practice are hierarchical (e.g., language, social networks, biological networks etc.), which suggests that hyperbolic
ML methods could be beneficial for applications in these areas. A growing body of literature studies this empirically,
see, e.g., (Monath et al., KDD'19), (Chami et al., NeurIPS'19), (Klimovskaia et al., Nature Communications'20).

**R3**

• *Related work:* Thank you for the suggestion. We will add a reference to Ganea et al. in the related work section.

• *Figure 3:* Please note that we characterize robust classifiers by the (worst case) margin they achieve. In Figure 3(c)
we see that $\alpha = 1$ indeed achieves the best margin.

**R4**

• *Hyperbolic vs. Euclidean margin:* Thank you for raising this issue. Yes, it is a notation overload. Up to Section 5, $\gamma$
denotes the hyperbolic margin. In Section 5 we compare hyperbolic and Euclidean margins, which are denoted as
$\gamma_H$ and $\gamma_E$ to avoid confusion. We will revise Table 1 accordingly and add a clarifying remark in a revised version.

• *Eq. (4.2):* Thank you for your comment. Currently, we explain the notation in Eq. (4.2) in the paragraph following
the equation, with a more detailed discussion of the robust loss deferred to the appendix. We will add a discussion
regarding our objective and the process that lead to Eq. (4.2) in the revised version.

• *Additional data sets:* As we discussed above, large-scale experiments are beyond the scope of the paper. For the
first proof of concept, we chose a data set that fulfills the assumptions in our theory and allows for validation of
the guarantees for both the adversarial approach and the dimension-distortion trade-off. Our chosen data set is
significantly larger than the data sets used in Cho et al. (1200 vs 50 data points).

[Meta-Review · NeurIPS 2020]

The paper presents new methods and, for the first time, theoretical guarantees for learning in hyperbolic spaces, which has benefits over Euclidean spaces especially learning with hierarchical data. The reviewers found the work sound. The narrow empirical evaluation was seen as a minor weakness.